# Pattern decorrelation in the mouse medial prefrontal cortex enables social preference and requires MeCP2

Pan Xu[1,2,7,8,10], Yuanlei Yue [1,2,10], Juntao Su[1], Xiaoqian Sun[3], Hongfei Du [4], Zhichao Liu [5,9], Rahul Simha[3], Jianhui Zhou[6], Chen Zeng [4] & Hui Lu [1,2✉]

Sociability is crucial for survival, whereas social avoidance is a feature of disorders such as Rett syndrome, which is caused by loss-of-function mutations in *MECP2*. To understand how a preference for social interactions is encoded, we used in vivo calcium imaging to compare medial prefrontal cortex (mPFC) activity in female wild-type and *Mecp2*-heterozygous mice during three-chamber tests. We found that mPFC pyramidal neurons in *Mecp2*-deficient mice are hypo-responsive to both social and nonsocial stimuli. Hypothesizing that this limited dynamic range restricts the circuit's ability to disambiguate coactivity patterns for different stimuli, we suppressed the mPFC in wild-type mice and found that this eliminated both pattern decorrelation and social preference. Conversely, stimulating the mPFC in MeCP2-deficient mice restored social preference, but only if it was sufficient to restore pattern decorrelation. A loss of social preference could thus indicate impaired pattern decorrelation rather than true social avoidance.

[1] The GW Institute for Neuroscience, The George Washington University, Washington, DC 20037, USA. [2] Department of Pharmacology and Physiology, School of Medicine and Health Sciences, The George Washington University, Washington, DC 20037, USA. [3] Department of Computer Science, School of Engineering and Applied Science, The George Washington University, Washington, DC 20037, USA. [4] Department of Statistics, Columbian College of Art and Sciences, The George Washington University, Washington, DC 20037, USA. [5] Department of Physics, Columbian College of Art and Sciences, The George Washington University, Washington, DC 20037, USA. [6] Department of Statistics, School of Arts and Sciences, University of Virginia, Charlottesville, VA 22904, USA. [7] Present address: Institute of Basic Science, Shandong Provincial Hospital Affiliated to Shandong First Medical University, Jinan, Shandong 250021, China. [8] Present address: Medical Science and Technology Innovation Center, Shandong First Medical University & Shandong Academy of Medical Sciences, Jinan, Shandong 250000, China. [9] Present address: School of Biological Information, Chongqing University of Posts and Telecommunications, Chongqing 400065, China. [10] These authors contributed equally: Pan Xu, Yuanlei Yue. ✉email: huilu@email.gwu.edu

Social interactions are among the most complex animal behaviors. Arising from a combination of emotional states, sensory input, social signals, and memories[1,2], social interactions involve many brain circuits but depend to a surprising degree on the prefrontal cortex (PFC) in both humans[3–5] and other species[6–9]. Excitatory pyramidal neurons within the PFC network carry social, spatial, and affective information such as eye contact and friendly gestures[10]. More specifically, neural activity in the part of the medial PFC (mPFC) known as the prelimbic cortex integrates spatial and social information, such that mice tend to return to locations where they have previously interacted with a fellow mouse[7]. When a mouse approaches or interacts with a conspecific[6,11,12], or senses socially relevant olfactory cues[13], mPFC neurons become more active. Excitatory activity within the mPFC is thus engaged in a variety of healthy, anxiety-free interactions.

Comparatively little is known, however, about what might be going awry in the mPFC circuit when the usual preference for social engagement seems to be lost, as appears to be the case in disorders such as Rett Syndrome. Girls with Rett, which is caused by loss of function of the X-linked gene *MECP2*, appear to develop normally for the first few years of life, then enter a regression phase during which they lose their acquired motor and cognitive skills[14]. They also develop signs of anxiety (such as respiratory dysrhythmias) and seem to lose interest in social interaction[14,15]. *Mecp2*-heterozygous female mice—which, like the humans, have mosaic expression of the mutant allele due to random X inactivation—replicate most of these features, including the social avoidance and anxiety-like behavior[16]. Although several approaches that mitigate MeCP2 deficiency in mice have been found to improve their social behavior—e.g., SUMOylation of MeCP2[17], allosteric modulation of mGlu7[18], or inhibition of pro-inflammatory P2X7 receptors[19]—to our knowledge there have been few studies that looked at mPFC function to understand social deficits in Rett syndrome. One electrophysiological study showed that the mPFC from adult male *Mecp2*-null mice shows functional hypoconnectivity[20]; a more recent electrophysiological study in *Mecp2*-heterozygous female mice found reduced excitatory postsynaptic currents with consequent disruption in excitatory/inhibitory (E/I) balance[21], which has been associated with social behavior deficits in mouse models of autism[22–24]. Valuable as these ex vivo studies are, however, they cannot answer questions about how information is encoded in the mPFC, as this encoding takes place at the level of coactivity within the population and not just at the level of individual neurons[25].

We therefore used in vivo $Ca^{2+}$ imaging via a head-mounted miniature microscope to monitor hundreds of mPFC neurons in both wild-type (*Mecp2*$^{+/+}$, WT) and heterozygous female *Mecp2*$^{+/-}$ mice, hereafter referred to as Rett mice, as they were presented choices of social and inanimate stimuli in a series of three-chamber tests. We found that the ability to show a preference for a social interaction over an inanimate object depends crucially on the mPFC's ability to decorrelate coactivity patterns in a manner similar to the way that the olfactory bulb distinguishes different odors.

## Results

### Rett mice lack a preference for social interactions and for social novelty.
The subject mouse was placed in the middle of three chambers, with each end chamber containing either another mouse or an object (Fig. 1a, Supplementary Fig. 1a). We tested Rett mice and their WT littermates at 5 months of age, when Rett mice show a clear lack of social preference. Before the test, the subject mouse was habituated for 10 min in the middle 45 cm chamber while the 10 cm end compartments were empty (Fig. 1a, Supplementary Fig. 1a). The subject remained in the middle chamber for three subsequent ten-minute testing sessions: in session 1 (S1), we placed a mouse (M1) and a centrifuge tube (i.e., an object, O) in opposite end chambers. In session 2 (S2), we switched the positions of M1 and O. In session 3 (S3), we replaced the object with a new mouse (M2), so the subject mouse had a choice of either a familiar mouse or a stranger to interact with. Our automated tracking software measured the amount of time the subject mouse spent interacting with the occupant in either end chamber, as defined by the mouse being oriented toward the stimulus and the mouse's nose being within the "sniffing zone," a 3 cm-wide zone near the barrier that separates an end chamber from the middle chamber.

WT mice were much more interested in their fellow mouse than in the object, but Rett mice spent equal time with both (Fig. 1b, Supplementary movies 1 and 2). The Rett mice also approached M1 more slowly than the object, perhaps indicating some anxiety (Fig. 1c, Supplementary Fig. 1b). The differences between genotypes narrowed in session 2, when M1 and O switched positions, because the WT subject mice seemed to be puzzled about the new position of the object in the chamber that had just contained M1 ("where's the mouse gone?"). In session 3, however, when the choice was between two mice, WT mice spent significantly more time exploring the new mouse (M2), while Rett mice spent about the same amount of time with M1 and M2 (Fig. 1b, Supplementary Fig. 1c, Supplementary movies 3 and 4). In fact, the Rett mice always spent 12–15% of their time interacting with whatever was in an end chamber, whereas WT mice spent about twice as much time interacting with M1 or M2 as they did with objects (~20% vs. ~10%).

The lack of preference for social interaction or social novelty was not due to differences in motor function, since Rett mice were as active as WT mice during the test (Supplementary movies 1–4). In fact, Rett mice were not hypoactive in the open field test at this age, though they clearly displayed an anxiety-like avoidance of the center of the arena (Supplementary Fig. 2a). Rett mice spent a similar amount of time in and crossing the middle zone across the three sessions (Supplementary Fig. 2b). Thus, Rett mice are less sociable than WT mice.

### Pyramidal neurons in the mPFC of Rett mice are hyporeactive to stimuli.
During the three-chamber test, we monitored the activity of excitatory pyramidal neurons in the prelimbic region of the mPFC using head-mounted miniature microscopes (Fig. 2a). For each mouse we identified, on average, 116 of these neurons (range 62–129) that expressed the $Ca^{2+}$ indicator, GCaMP6m (Fig. 2b), without bleaching effects in recording at 15 frames per second (Supplementary Fig. 3a, b). We synchronized the $Ca^{2+}$ imaging with the behavioral recordings, following the same neurons within each mouse across the different test sessions (see Methods). To determine whether population coding differs in response to different stimuli, we first estimated the $Ca^{2+}$ transient rate and amplitude (here defined as $\Delta F/F$, the change in GCaMP fluorescence intensity relative to baseline; it reflects the number of spikes per transient[26]). WT mice showed higher transient rates across the entire population of recorded mPFC neurons during their approaches to any stimulus, across all three sessions (Fig. 2c, d). This was consistent with changes in inter-event intervals in both groups of mice (Supplementary Fig. 3c). When Rett mice interacted with either M1 or O in S1, their overall transient rate was roughly 70–80% that of WT mice (Fig. 2d), but in the middle of the chamber, there was far less difference between the two genotypes (Fig. 2e). The hypoactivity of mPFC excitatory neurons in Rett mice was therefore context-

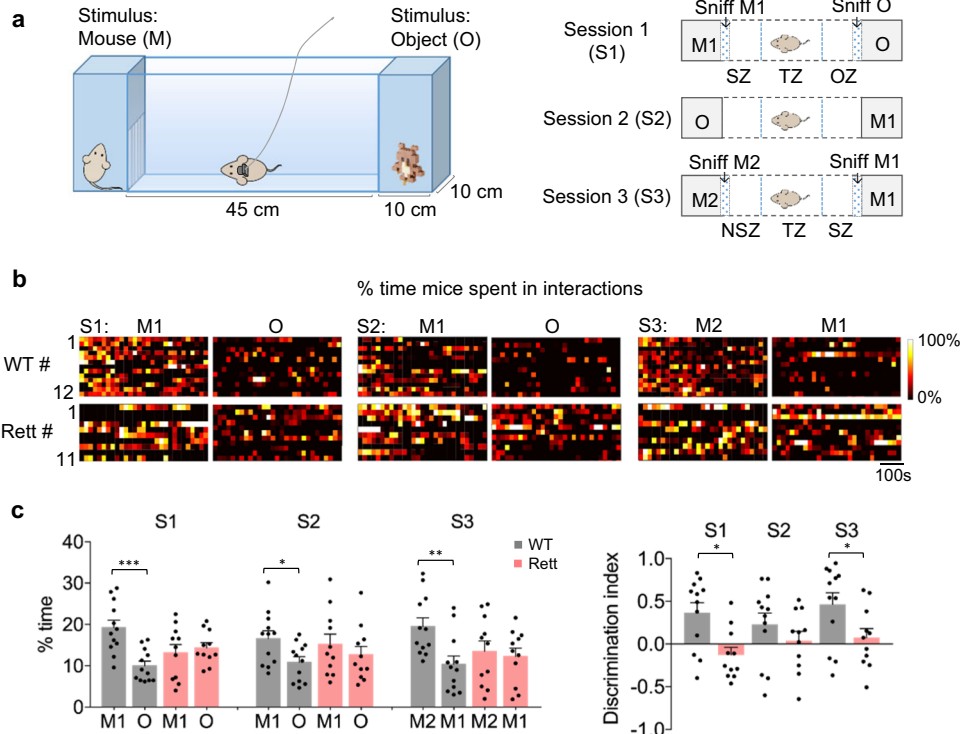

**Fig. 1 Female Rett mice display social deficits in social interaction test. a** Left: the apparatus for the social interaction test. The subject mouse is in the middle 40 cm-long chamber; the 10 cm end compartments contain different stimuli. Right: the three sessions of the social interaction test. In the first 10 minute test session (S1), a strange mouse (M1) and an object (O) were placed in the end chambers; session 2 (S2) used the same stimuli but swapped their positions; in session 3 (S3) a new mouse (M2) replaced O, so that the subject mouse must choose whether to interact with a familiar or strange mouse (M1 vs. M2). "Sniff" means the mouse was at the end of the chamber and interacting with a stimulus; social zone (SZ), object zone (OZ), and new social zone (NSZ) indicate the 10 cm regions of the central chamber that are nearest to the end chambers of M1, O, and M2, respectively; transition zone (TZ) means the middle area in the central chamber. **b** Raster plots of the percentage of time mice spent in interacting with different stimuli in 3–10 minute sessions. The time is divided into 20 second bins; the color ranges from deep red (0%) to yellow (100%). **c** Left: the percentage of time that WT ($Mecp2^{+/+}$; $n = 12$) and Rett ($Mecp2^{+/-}$; $n = 11$) mice spent interacting with stimuli in each session. Right: Discrimination lindices of WT and Rett mice indicating the difference in the amount of time the mice spent with each stimulus in a given session (M1 vs. O in S1 and S2; M2 vs. M1 in S3), relative to the total duration of stimulus interactions out of the whole session. Data are represented as mean ± SEM. *$P < 0.05$, **$P < 0.01$, ***$P < 0.001$, two-way RM ANOVA with Bonferroni–corrected post hoc comparisons. Source data are provided as a Source Data file.

dependent, being notable only during interactions with stimuli. Nevertheless, the overall dynamic range of the Rett circuit was less than half that of WT (Fig. 2e). Pyramidal neurons in the mPFC of both WT and Rett mice showed similar transient rates when not interacting with stimuli, and these transient rates were not influenced by spatial information (Fig. 2f) or the speed at which the mice moved (Fig. 2g).

When we subjected the mutant and WT mice to the open field test, which is not related to social behavior, we again saw stimulus-dependent hypoactivity: the transient rate of Rett mPFC excitatory neurons was significantly lower than that in WT controls only when the mice were in the anxiogenic center area (Supplementary Fig. 4a). Thus, the hypoactivity in the mPFC is most acute just when the animal is being stimulated, either by interaction or by something anxiety-provoking. This tendency for the MeCP2-deficient circuit to be hypo-responsive in the context of stimulation parallels our recent finding in the motor cortex, where excitatory pyramidal neurons in layer 2/3 were hypoactive in male Mecp2 null mice only in response to changing speeds on a running wheel[27]. Indeed, we recorded these socially-neutral L2/3 motor cortical neurons in Rett mice during the three-chamber test and found they were not hypoactive in the context of social behavior (Supplementary Fig. 4b). Therefore, the loss of MeCP2 seems to dampen the responsiveness of excitatory circuits to stimuli, in both the mPFC and the motor cortex. Because we were unable to determine the

genotypes of individually recorded neurons, we cannot say whether this is due to cell-autonomous or non-cell-autonomous effects of MeCP2 loss, but both are possible[21].

**Experience-dependent plasticity is limited in the Rett mPFC.** WT mice interacted most intensely with the mice in the end chambers at the beginning of the interaction; once they became familiar, the WT mice lost interest. This was reflected in an overall decrease in the $Ca^{2+}$ transient rate from sessions 1 to 3 (Fig. 3a). Moreover, in WT mice, the proportion of neurons responding specifically to social interaction with M1 (see Methods) diminished from one session to the next, as the subject mouse became familiar with M1 (Fig. 3b). The Rett mice did not show either of these trends, suggesting impaired experience-dependent plasticity.

To better understand the changes of activity patterns of the recorded neurons during social interactions across sessions, we used variational auto-encoding (VAE; see Methods) (Fig. 3c). This dimension-reduction method searches for features that best represent the spatiotemporal pattern of each imaging frame in two dimensions[28]. We examined social interactions from all three sessions and found that the relative radius of the feature distribution diminished across sessions in WT mice (Fig. 3d). This suggests that, as the mPFC circuit becomes familiar with a stimulus, it responds more consistently to that stimulus. The feature distribution for the Rett group, by contrast,

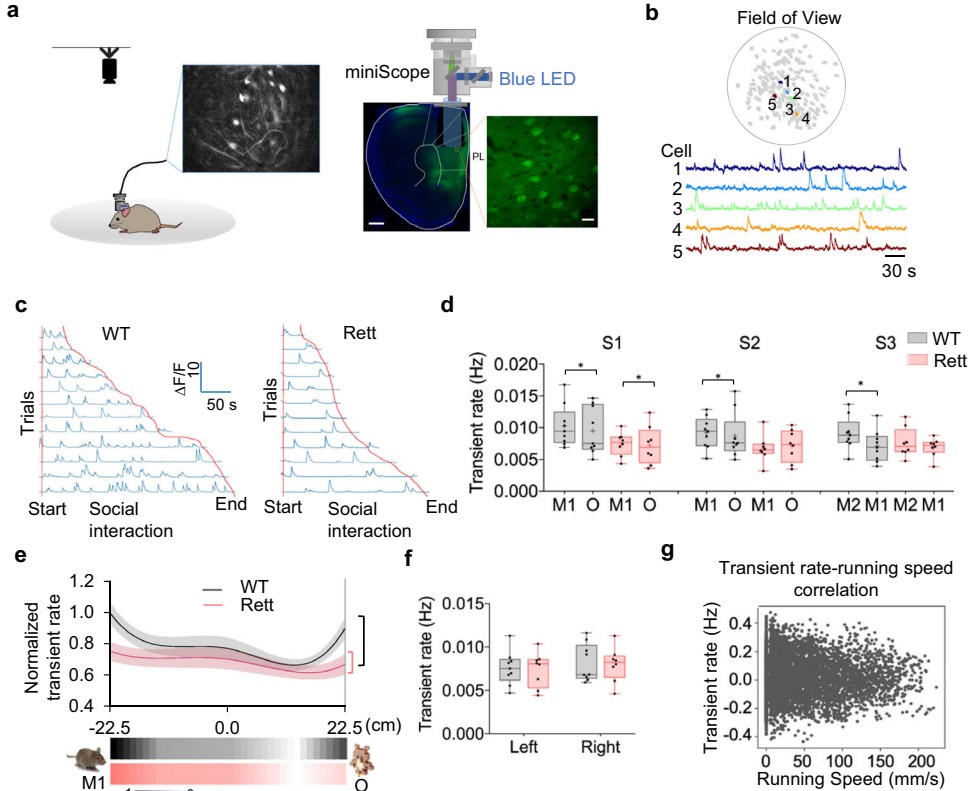

**Fig. 2 Reduced transient rate of mPFC neural circuit is related to the lack of interest in social interactions of Rett mice. a** Schematic of the experimental approach. Left: the calcium signal of the freely behaving mice in the chamber was imaged with a head-mounted miniature microscope, while behaviors were recorded by a high-speed camera. Middle: imaging setup and histology of the prelimbic (PL) region with GRIN lens implantation. Scale bar, 500 μm. Right: confocal image of PL neurons showing GCaMP6m expression. Scale bar, 10 μm. **b** Top: the field of view under a GRIN lens in one mouse with identified neurons numbered and colored. Bottom: fluorescence traces of example neurons marked in the above panel. **c** Responses of one representative neuron from a WT and a Rett mouse during social interactions in S1. Trials were organized by the length of interactions. **d** The averaged calcium transient rate when WT ($Mecp2^{+/+}$; $n = 9$) and Rett ($Mecp2^{+/-}$; $n = 8$) mice were engaged with different stimuli over the three sessions. Data are represented as mean ± SEM. *$P < 0.05$; **$P < 0.01$, two-way ANOVA with Bonferroni–corrected post hoc comparisons. **e** Top: the response field of the neural response. The normalized averge transient rates of WT ($n = 9$) and Rett ($n = 8$) mice for S1 are presented as solid lines, with standard error (SEM) conveyed by shaded regions (grey for WT, pink for Rett). Bottom: heatmap of the spatial field of the averaged response of WT and Rett neurons. The red and black brackets to the right mark the high and low points of the Rett and WT curves, to highlight the difference in their dynamic ranges. **f** Averaged transient rate of mPFC neurons when WT ($n = 9$) and Rett ($n = 8$) mice were in two 10 cm ends of the central chamber, which showed no difference between genotypes. **g** Relationship between the transient rate of the mPFC neurons and the running speed (per 10 second bin) of one representative mouse. There was no significant correlation. Box boundaries in (**d**), (**f**) are the 25th and 75th percentiles, the horizontal line across the box is the median, the cross "+" indicates the mean value, and the whiskers indicate the minimum and maximum values. Source data are provided as a Source Data file.

remained similar (Fig. 3d, bottom right), even though the range of responses was much wider and less consistent. The decrease of the feature distribution radius correlated with the shorter duration of interaction with M1 in individual WT mice, but no correlation was evident in the Rett group (Fig. 3e). To exclude the possibility that the difference in $Ca^{2+}$ dynamics simply reflected the fact that WT mice spent more time interacting with M1 in the first two sessions, we used the first 150 images collected in each interacting episode to perform the same analysis and found a similar distinction between WT and Rett groups (Supplementary Fig. 5a, b). Therefore, loss of MeCP2 impairs experience-dependent refinement of activity across the circuit, consistent with previous reports that MeCP2 is involved in experience-dependent plasticity[29,30].

We next asked whether the mPFC excitatory circuit is hypoactive in Rett mice because of too much inhibition from PV interneurons. This possibility was suggested by recent work showing that parvalbumin-expressing (PV) neurons (but not other inhibitory neuron types) mature early and are upregulated in the context of $Mecp2$ deficiency: whereas $Mecp2$ loss markedly

reduces dendritic arborization in pyramidal neurons, it actually enriches PV axonal and dendritic complexity[31]. We therefore injected AAV-flex-GCaMP6m into PV-Cre mice and measured PV neuron activity (Supplementary Fig. 6a, b). In control mice, the response (amplitude) of PV interneurons increased just before and during the interactions with stimuli, consistent with recent studies showing that PV neurons in the mPFC increase their discharge rates upon social interaction[32,33] (Supplementary Fig. 6c). The response of PV interneurons in Rett mice diminished prior to the onset of interaction with M1, though it increased for a few seconds after, albeit at a much lower level than in controls. Considering the fast-spiking property of PV interneurons and the limited temporal resolution of GCaMP6m, we calculated the area under the curve (AUC) of each neuron's fluorescence trace to better represent their discharge throughout the periods of interaction. Rett mice showed a significantly greater AUC per second than WT during M1 and M2 interactions (Supplementary Fig. 6d). Considering the activity of PV interneurons across the whole chamber, as we did for pyramidal neurons (Fig. 2e), we found that they were more active

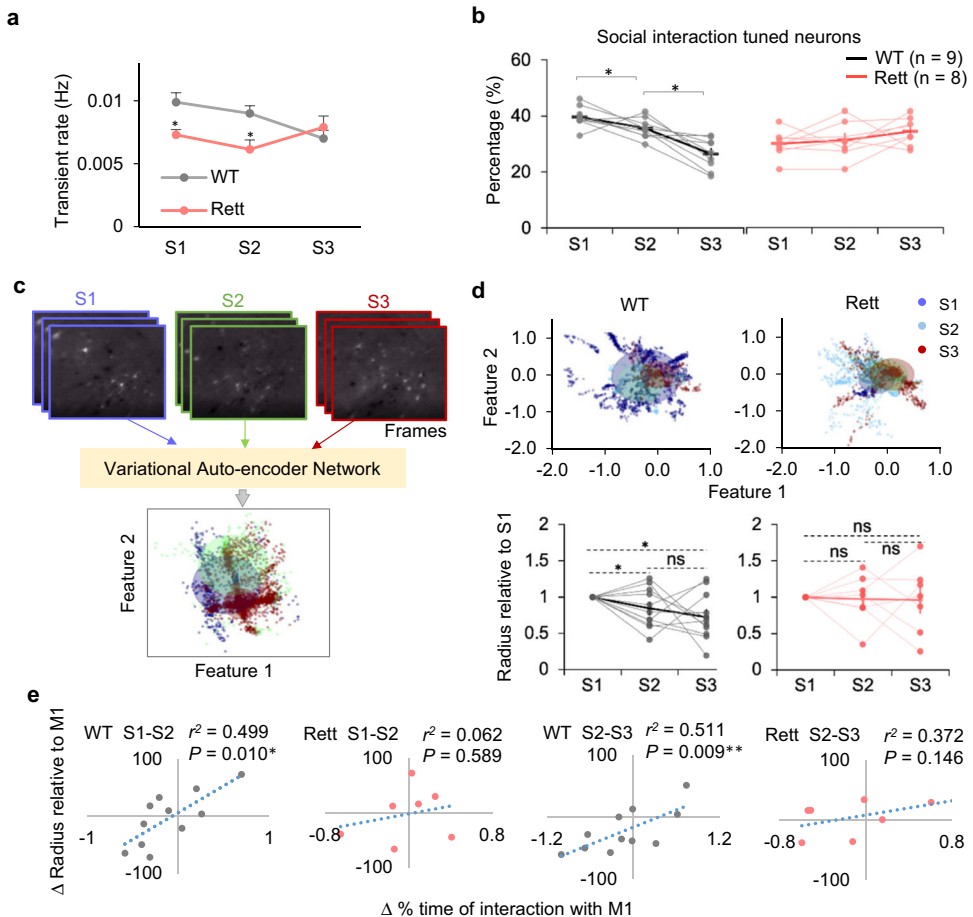

**Fig. 3 Limited responsiveness and plasticity in the Rett mPFC circuit. a** Transient rate changes of the mPFC neurons in the WT (*Mecp2*+/+; *n* = 9) and Rett (*Mecp2*+/−; *n* = 8) mice over three sessions of interacting with M1. Error bars indicate SEM. #*P* < 0.05, between genotypes, *\*P* < 0.05, WT vs. Rett in each session. Two-way RM ANOVA with Bonferroni–corrected post hoc comparisons. **b** Percentage of recorded mPFC neurons activated during social interactions in WT (*n* = 9) and Rett (*n* = 8) mice in each session. Each dot indicates one mouse. Lines connect data from one mouse; thicker solid lines connect the averaged value from all the mice in each group. *\*P* < 0.05, between sessions. Two-way RM ANOVA with Bonferroni–corrected post hoc comparisons. **c** The variational auto-encoder approach (VAE) transforms the coding information within each imaging frame into two features (1 and 2), represented by one dot in a two-dimensional plot. Frames recorded during social (M1) interaction periods over three sessions were transformed and their corresponding feature distributions were color-coded dark blue, light blue, and red, respectively. The circular shaded area represents the distribution radius (See Methods). **d** Top: Feature distributions of M1 interactions in S1, S2, and S3 from the representative WT and Rett mice. Bottom: changes of relative distribution radius (normalized to S1) responding to M1 interactions across three sessions of each WT (*n* = 11) and Rett (*n* = 7) mouse. Each dot indicates the normalized average value. Lines connect data from one mouse; thicker solid lines connect the averaged value from all the mice in each group. Error bars indicate SEM. *\*P* < 0.05, ns, no significance. Two-way RM ANOVA with Bonferroni–corrected post hoc comparisons. **e** Correlation between changes in the duration of social interaction and relative radius between sessions (S1-S2 and S2-S3). Pearson's correlations were calculated separately for WT (gray) and Rett (red) mice. Dots indicate individual mice. *\*P* < 0.05, *\*\*P* < 0.01, regression. Source data are provided as a Source Data file.

than WT before and after interactions, but similar to WT in the middle of the chamber (Supplementary Fig. 6e, f). Thus, their activity pattern is the inverse of what we observed with mPFC pyramidal neurons, and their dynamic range is less limited.

**Pattern decorrelation is impaired in the Rett mPFC circuit.** Thus far we have examined the activity of the mPFC averaged across all recorded neurons, but this misses out on the reason neural circuits have such enormous capacity to learn: their ability to encode information through their patterns of coactivity[25]. Pearson correlation coefficients can be used to detect functional correlations (coactivity) within the circuit, and pattern decorrelation disambiguates overlapping activity patterns. This is, for example, how the olfactory bulb distinguishes odorants that differ by only a slight change in molecular structure[34–36]. The limited plasticity of the Rett circuit led us to hypothesize that what looks like "social avoidance" in Rett mice might actually be an inability

to discriminate between M and O (or M1 and M2), i.e., impaired pattern decorrelation. The MeCP2-heterozygous mPFC may simply not have the dynamic range to form different patterns for different stimuli, and without distinct patterns, there can be no preference.

To test this hypothesis, we observed the temporal correlation of neural activities. In both WT and Rett mice, the pattern of mPFC neuron temporal correlation was specific to the social or object stimulus (Fig. 4a). The pairwise correlation coefficients for different stimuli across the recorded mPFC neurons from all mice were distributed around zero, but with various widths (Fig. 4b). In WT mice, the decorrelation was significant, with the full-width at half maximum (FWHM) of the correlation distribution being lower for the more-attractive stimuli in each session (Fig. 4c). This remained true when we focused on strongly correlated neuronal pairs (threshold = ±0.2) (Supplementary Fig. 7a). The decorrelation did not reach significance in the Rett

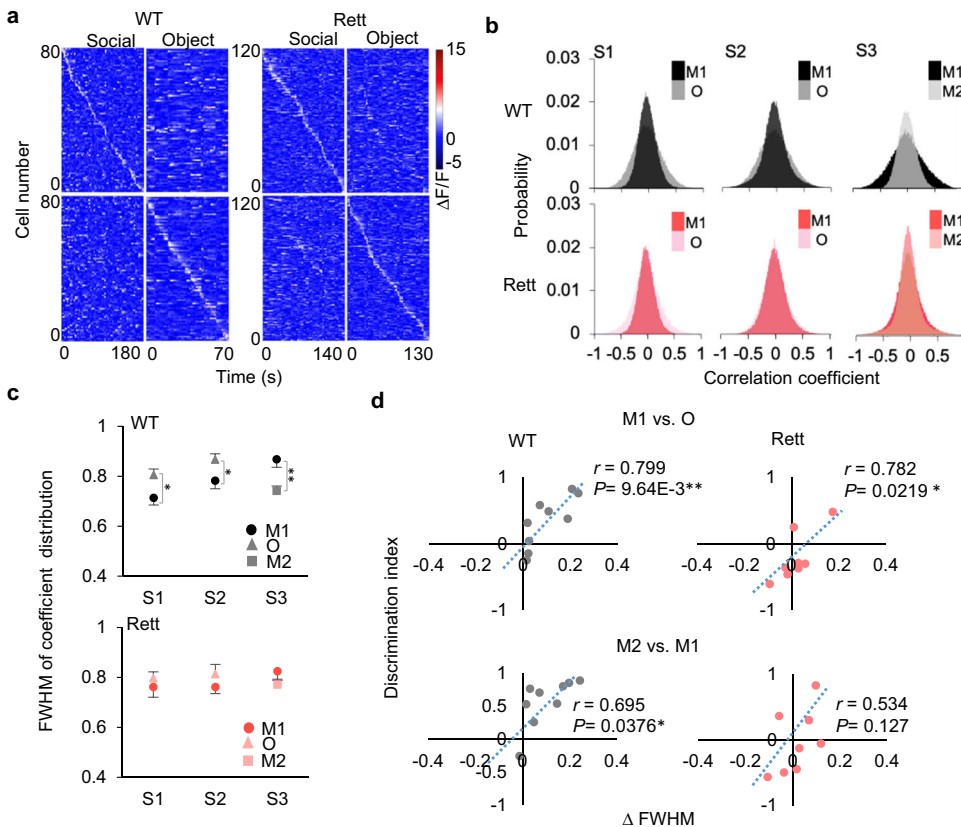

**Fig. 4 Pattern decorrelation enables social preference in WT mice but is impaired in Rett mice. a** Raster plots of calcium activity of individual mPFC neurons during interactions with different stimuli, from representative WT and Rett mice. Calcium transient traces are sorted by the peak activity time of social interaction (left top) and object interaction (right bottom), while the calcium traces of those sorted neurons in responding to object (top right) and social (bottom left) stimuli were also plotted in the same sequence of the sorted plot. **b** Distribution of pair-wise Pearson correlation coefficients among all recorded mPFC neurons of WT (*Mecp2*$^{+/+}$; $n = 9$) and Rett (*Mecp2*$^{+/-}$; $n = 8$) mice in responding to stimuli in each session. The black and red plots represent M1 interactions, while the lighter colors (grey and pink) represent the interaction with the other stimulus (O or M2). **c** The averaged full width at half maximum (FWHM) of correlation coefficient distribution (b) of individual WT mice ($n = 9$) and Rett mice ($n = 8$) during interactions with different stimuli. Data are represented as mean ± SEM. *$P < 0.05$, **$P < 0.01$, two-way RM ANOVA with Bonferroni-corrected post hoc comparisons. **d** The relationship between ΔFWHM and the discrimination index in each mouse, by genotype. In Rett, the ΔFWHM is centered around zero, meaning little or no decorrelation; the discrimination index is mostly below zero. $n = 9$ WT and 8 Rett mice. Each dot represents an individual mouse; Pearson's correlation coefficients were calculated across genotypes. **$P < 0.01$, ***$P < 0.001$, regression.

mice for any session. The relationship between ΔFWHM and the discrimination index (the difference in the amount of time spent with each stimulus in a given session) was also much stronger in the WT group than in the Rett mice (Fig. 4d, Supplementary Fig. 7b). We conclude that pattern decorrelation in the mPFC prelimbic circuit is impaired by loss of MeCP2, which renders Rett mice unable to disambiguate social from nonsocial (or novel social) stimuli.

**Suppressing hyperactive inhibitory neurons in the mPFC restores social behavior in Rett mice.** Based on the above results and the central role of PV interneurons in maintaining excitatory/inhibitory balance in the cortex[37], we asked whether suppressing these interneurons just enough to relieve excitatory hypoactivity would restore pattern decorrelation and rescue the social deficit of Rett mice. We expressed inhibitory halorhodopsin NpHR (EYFP control protein) in PV neurons and GCaMP6m in excitatory pyramidal neurons by injecting AAV-Ef1a-DIO-eNpHR 3.0-EYFP (AAV-Ef1a-DIO-EYFP as a control) and AAV-CaMKII-GCaMP6m into the prelimbic region of PV-Cre Rett mice (Supplementary Fig. 8a) (see Methods). This strategy enabled us to simultaneously manipulate PV neuron activity and monitor excitatory neuron activity (Fig. 5a). Optogenetic suppression of

PV neurons significantly raised the transient rate of mPFC excitatory neurons in mice expressing NpHR, compared to control mice expressing EYFP (Fig. 5b, top). The transient amplitude was not influenced by the manipulation (Fig. 5b, bottom). To ensure that light stimulation itself was not inducing functional changes, we measured neural activity before and after light stimulation and found no difference in either the NpHR or EYFP group (Supplementary Fig. 8b).

Because Rett mPFC neurons were less active than WT during interactions with both objects and mice, we had an opportunity to test our hypothesis that a limited dynamic range impairs pattern decorrelation: we manipulated the neurons only when the animal was in the social zone (or novel social zone in S3) to see if this would be sufficient to restore their social preference in Rett mice (Fig. 5c). (Pattern decorrelation disambiguates two or more stimuli; location-specific stimulation might increase the dynamic range and responsiveness to the social interaction but would not affect the object interaction, so we would predict it would not restore social preference to Rett mice.) Location-specific optogenetic manipulation significantly increased the mPFC neural transient rate (Fig. 5d, top), but did not separate activity patterns enough (Fig. 5e) to enable the NpHR mice to show social preference compared to Rett mice expressing EYFP (Fig. 5d,

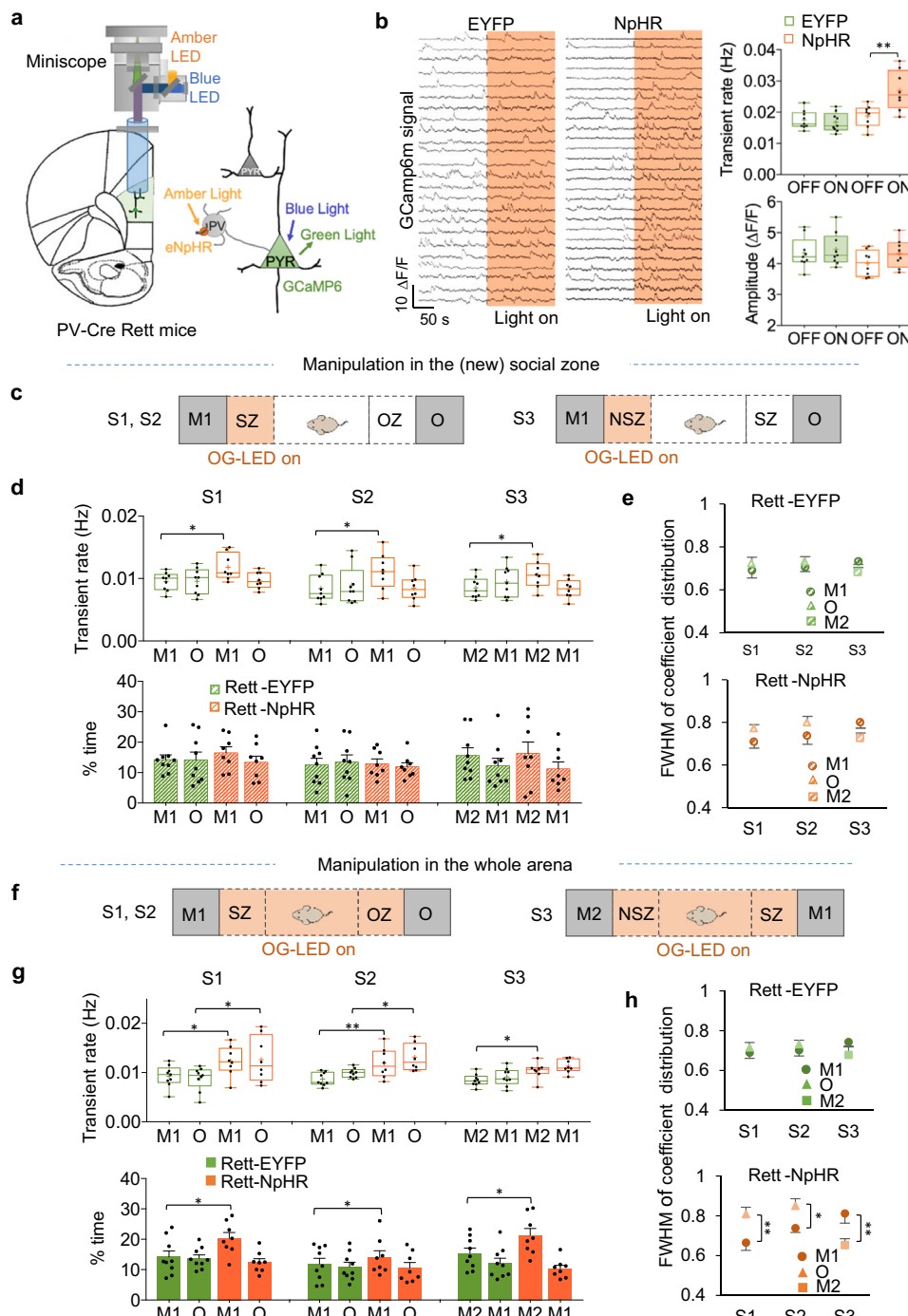

bottom; Supplementary Fig. 8c, d). To ensure that the failure of the social-zone-specific manipulation to restore social preference was not due to it inducing spatial aversion, we performed a place preference test with prelimbic optogenetic interference in one randomly chosen half of an empty chamber. The mice spent a similar amount of time exploring both sides of the chamber (Supplementary Fig. 8e). This verified that prelimbic manipulation did not have a spatial effect.

We then delivered light stimulation throughout the whole 10 min of the three-chamber test (Fig. 5f). This intervention significantly improved the social performance of Rett mice, increasing the time they spent in the vicinity of the social and novel social stimuli to WT levels (Fig. 5g, Supplementary Fig. 8f). PV neuron suppression significantly increased the transient rate

of mPFC pyramidal neurons in response to the interactions with both social and object stimuli (Fig. 5g). Lastly, the NpHR-expressing Rett mice, like WT, exhibited greater pattern decorrelation (separation along the y-axis), with both lower and higher FWHM values, than EYFP-expressing mice during social interactions and showed an experience-dependent correlation increase in responding to M1 and M2 across sessions (Fig. 5h, Supplementary Fig. 8g). Thus, suppressing inhibitory input from PV neurons to increase the activity of the mPFC pyramidal neurons in Rett mice by ~20%, to roughly WT levels, rescued social behavior in these Rett mice by restoring pattern decorrelation.

We mentioned earlier that anxiety might be a factor in the Rett mice's lack of sociability, in which case the problem would be

**Fig. 5 Elevating the activity of mPFC pyramidal neurons throughout the whole arena rescues social interactions of Rett mice by restoring pattern decorrelation. a** Schematic of recording configuration (left) and of light path (right) for the head-mounted microscope enabling optogenetic parvalbumin (PV) suppression and simultaneous imaging of pyramidal neurons (PNs) in the PL region of the *PV-Cre* mice. Inhibitory halorhodopsin NpHR (control protein, EYFP) and GCaMP6m were expressed by PV interneurons and excitatory PNs, respectively. **b** Sample calcium traces (left), averaged transient rate and amplitude (right) of PL pyramidal neurons in NpHR ($n = 9$) and EYFP ($n = 8$) mice during ordinary homecage behavior, with and without PL optogenetic manipulation. The shaded area represents the period with the amber light on. Values were plotted as mean ± SEM. *$P < 0.05$, amber light ON versus OFF, two-way RM ANOVA with Bonferroni–corrected post hoc comparisons. **c** Amber light was delivered only when the mouse was in the social zone (SZ) or new social zone (NSZ). **d** Optogenetic stimulation of the mPFC only in the social zone elevated the transient rate but did not affect the time spent in interacting with M1 or M2 (top). *$P < 0.05$, NpHR ($n = 9$) versus EYFP ($n = 8$), two-way ANOVA with Bonferroni–corrected post hoc comparisons. **e** Social zone-only stimulation did not rescue pattern decorrelation, as indicated by the overlapping FWHM of correlation distribution of individual EYFP ($n = 9$) and NpHR ($n = 8$) mice during interactions with different stimuli. Data are represented as mean ± SEM. Two-way RM ANOVA with Bonferroni–corrected post hoc comparisons. **f** The second experimental design involved delivering optogenetic light throughout the entire testing session. **g** The whole-arena stimulation of mPFC restored social preference (top) and the transient rate (bottom) of NpHR mice ($n = 9$), compared with EYFP mice ($n = 8$). *$P < 0.05$, **$P < 0.01$, two-way ANOVA with Bonferroni–corrected post hoc comparisons. **h** Whole-arena manipulation of mPFC restored pattern decorrelation, with the FWHM of NpHR mice showing a significant difference between stimuli in each session. NpHR: $n = 9$ mice; EYFP: $n = 8$ mice. Error bars indicate SEM. *$P < 0.05$, **$P < 0.01$, two-way RM ANOVA with Bonferroni–corrected post hoc comparisons. Box boundaries in (**b**), (**d**), and (**g**) are the 25th and 75th percentiles, the horizontal line across the box is the median, the cross "+" indicates the mean value, and the whiskers indicate the minimum and maximum values. Source data are provided as a Source Data file.

social avoidance, rather than lack of social preference. We, therefore, examined whether the location-specific stimulus, which had failed to restore social behavior, alleviated anxiety: if so, we would conclude that rescuing anxiety is not sufficient to restore social interaction in Rett mice. We performed location-specific stimulation during four consecutive 3 min periods (sessions 1–4) in the middle chamber, with simultaneous in vivo $Ca^{2+}$ imaging and optogenetic stimulation of the left prelimbic region. Stimulating amber light was delivered when mice were in the anxiogenic center area in S1 and S3, and the optogenetic suppression of eNpHR-expressing PV neurons effectively elevated the transient rate of mPFC excitatory neurons (Supplementary Fig. 9a, b). This manipulation reduced the anxiety-like behavior of NpHR-expressing Rett and WT mice in those two sessions, extending the time they spent in the center area (Supplemental Fig. 9c, d). This effect was even more pronounced during the second period of stimulation (S3) when both genotypes spent even more time in the center area.

The level of stimulation used in the location-specific social test was therefore sufficient to relieve anxiety. If anxiety were the primary reason Rett mice are unsociable, then rescuing circuit hypoactivity in the social zone should have prolonged the time that the Rett mice spent in the social zone, but this was not the case. These results indicate that Rett mice do not suffer social avoidance due to anxiety but rather that they lack the ability to express social preference due to insufficient pattern decorrelation.

**Manipulating mPFC pyramidal neuron activity in WT mice causes loss of social preference**. These findings support a causal relationship between the loss of dynamic range, impaired pattern decorrelation in the mPFC prelimbic excitatory circuit, and the social deficit of Rett mice. To further test whether pattern decorrelation depends on normal E/I balance and is necessary for generating normal social behavior, we tested whether inhibiting mPFC excitatory activity in WT mice would reproduce the social deficits observed in Rett mice. We expressed AAV-Ef1a-DIO-eNpHR 3.0-EYFP (AAV-Ef1a-DIO-EYFP as a control) and AAV-ef1a-Flex-GCaMP6m in the prelimbic region of CaMKII-Cre WT mice (Fig. 6a). The amber light effectively decreased the transient rate of pyramidal neurons (Fig. 6b). We then performed behavioral tests similar to those in Fig. 5. Each mouse received two types of optogenetic manipulation in a random sequence. WT mice expressing NpHR in mPFC excitatory pyramidal neurons showed social deficits resembling those of Rett mice. Both location-specific optogenetic manipulation and whole-arena

manipulation effectively reduced the activity of the mPFC excitatory neurons in response to stimulus interactions (Fig. 6c, e) and impaired pattern decorrelation (Fig. 6d, f).

To test this causal relationship from a different direction, we optogenetically suppressed inhibitory neuron activity in WT mice expressing NpHR in the mPFC PV population (Supplementary Fig. 10a, c). In both location-specific and whole-arena manipulation modes, enhanced excitation in the mPFC impaired social preference in WT animals (Supplementary Fig. 10a, c). Pattern decorrelation shrank to insignificance during social interaction in NpHR-expressing mice (Supplementary Fig. 10b, d).

These observations indicate that skewing E/I balance in either direction leads to failure of pattern decorrelation and loss of social preference (Fig. 7). In Rett mice, elevating the activity of the hypoactive pyramidal neurons improves social behavior, but only to the extent that these measures restore E/I balance and pattern decorrelation. We conclude that pattern decorrelation in the prelimbic excitatory circuit underlies social discrimination in mice.

## Discussion

Pattern decorrelation is perhaps best known for enabling discrimination among odors[36,38], but it has also been observed in the dentate gyrus of the hippocampus[39,40], which pre-processes information for storage and classification in area CA3[41]. If the activity patterns in response to different stimuli fail to diverge, the outputs will be indistinguishable to downstream target regions. Recovering the capacity for prelimbic pattern decorrelation rescues the social deficit in Rett mice, despite the fact that the mice remain MeCP2-deficient. This is rather remarkable, but it appears that restoring function to specific brain regions can exert broad benefits in MeCP2 mutant mice: stimulating the motor circuit in a forced exercise paradigm not only improves motor coordination but reduces anxiety and extends the lifespan of male Mecp2-null mice[27], and restoring activity in area CA1 improves social behavior in a mouse model of MeCP2 duplication syndrome[42].

Normal mPFC activity is crucial for the expression of sociability, and others have reported social behavior impaired by dysfunction of the mPFC circuit or its afferent and efferent signaling[23,43–45]. Disruptions in mPFC E/I balance in either direction ($E > I$ or $E < I$) have been linked to multiple abnormal social behaviors[22–24,46]. Adult mice lacking the autism-associated gene Cntnap2 have social deficits arising from a hyperactive mPFC, whose normalization rescues the phenotype[33]. Our study may help explain why Cntnap-deficient mice are deficient in both

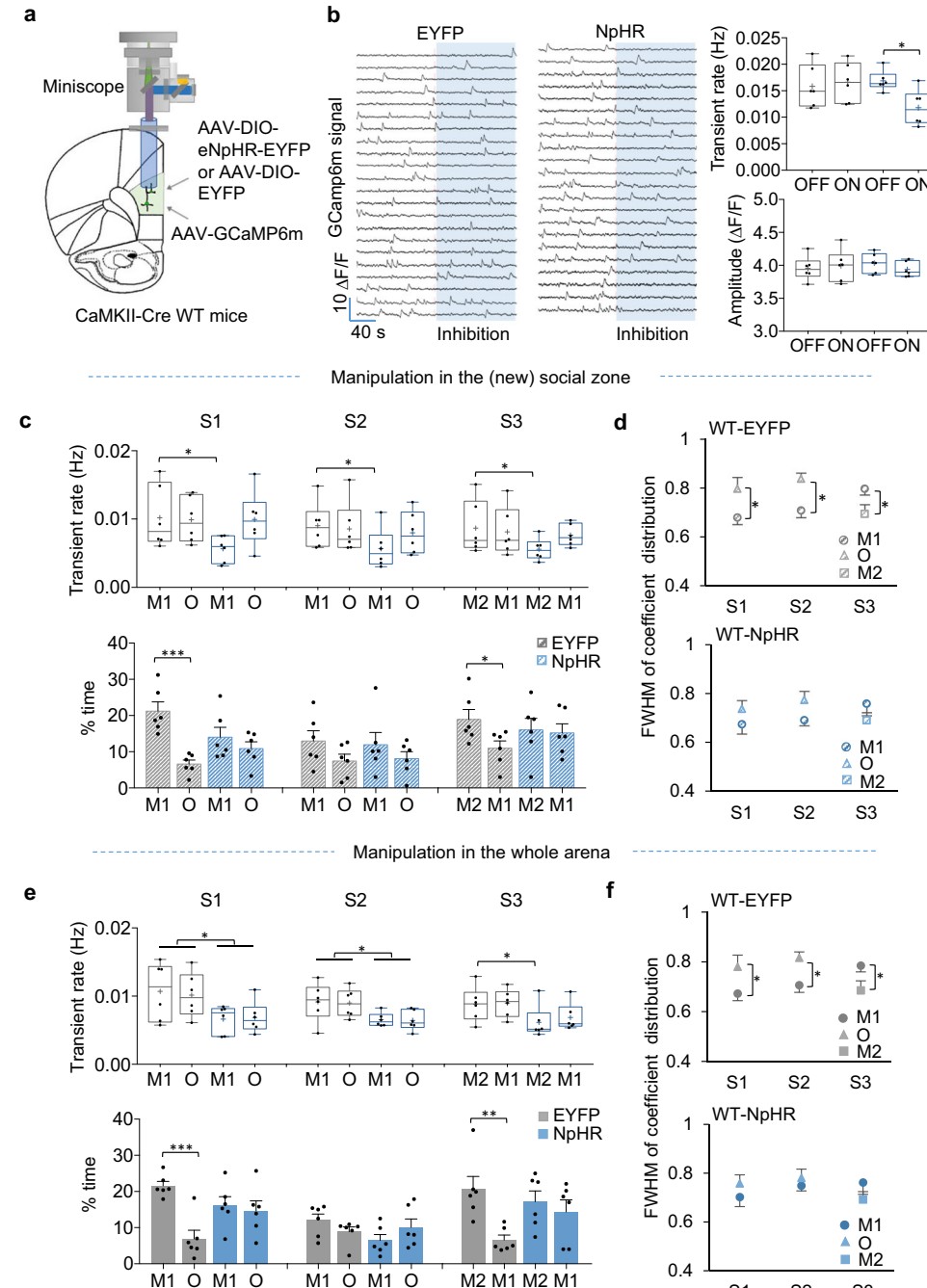

**Fig. 6 Inhibiting mPFC pyramidal neuron activity impairs social preference in WT mice. a** Diagram showing virus injection and lens implantation in the PL region for microscope enabling simultaneous optogenetic inhibition and imaging of pyramidal neurons of the *CaMKII-Cre* WT (*Mecp2+/+*) mice. **b** Sample calcium traces (left), transient rate, and amplitude (right) of PL pyramidal neurons in NpHR or EYFP mice recorded in their home cage with and without optogenetic inhibition. The shaded area represents the period with the optogenetic light on. NpHR: $n = 6$ mice; EYFP: $n = 6$ mice. **c** Social-zone manipulation of mPFC reduced preference for the social novelty of NpHR WT mice (top) and inhibited their transient rate (bottom) in the stimulated zone. *$P < 0.05$, ***$P < 0.001$, NpHR ($n = 6$) versus EYFP ($n = 6$), two-way ANOVA with post hoc Bonferroni correction. **d** The averaged FWHM of correlation distribution of EYFP ($n = 6$) and NpHR ($n = 6$) WT mice during interactions with different stimuli. Data are represented as mean ± SEM. *$P < 0.05$, two-way RM ANOVA with Bonferroni–corrected post hoc comparisons. **e** The whole-arena stimulation of mPFC more severely impaired the sociability and social novelty preference (top) and decreased the transient rate (bottom) of NpHR mice ($n = 9$), compared with EYFP mice ($n = 8$). Error bars indicate SEM. *$P < 0.05$, **$P < 0.01$, ***$P < 0.001$, NpHR ($n = 6$) versus EYFP ($n = 6$), two-way ANOVA with post hoc Bonferroni correction. **f** Whole-arena manipulation of the mPFC impaired pattern decorrelation between two stimuli in NpHR WT mice. NpHR: $n = 6$ mice; EYFP: $n = 6$ mice. Error bars indicate SEM. *$P < 0.05$, two-way RM ANOVA with Bonferroni–corrected post hoc comparisons. Box boundaries in (**b**), (**c**), and (**e**) are the 25th and 75th percentiles, the horizontal line across the box is the median, the cross "+" indicates the mean value, and the whiskers indicate the minimum and maximum values. Source data are provided as a Source Data file.

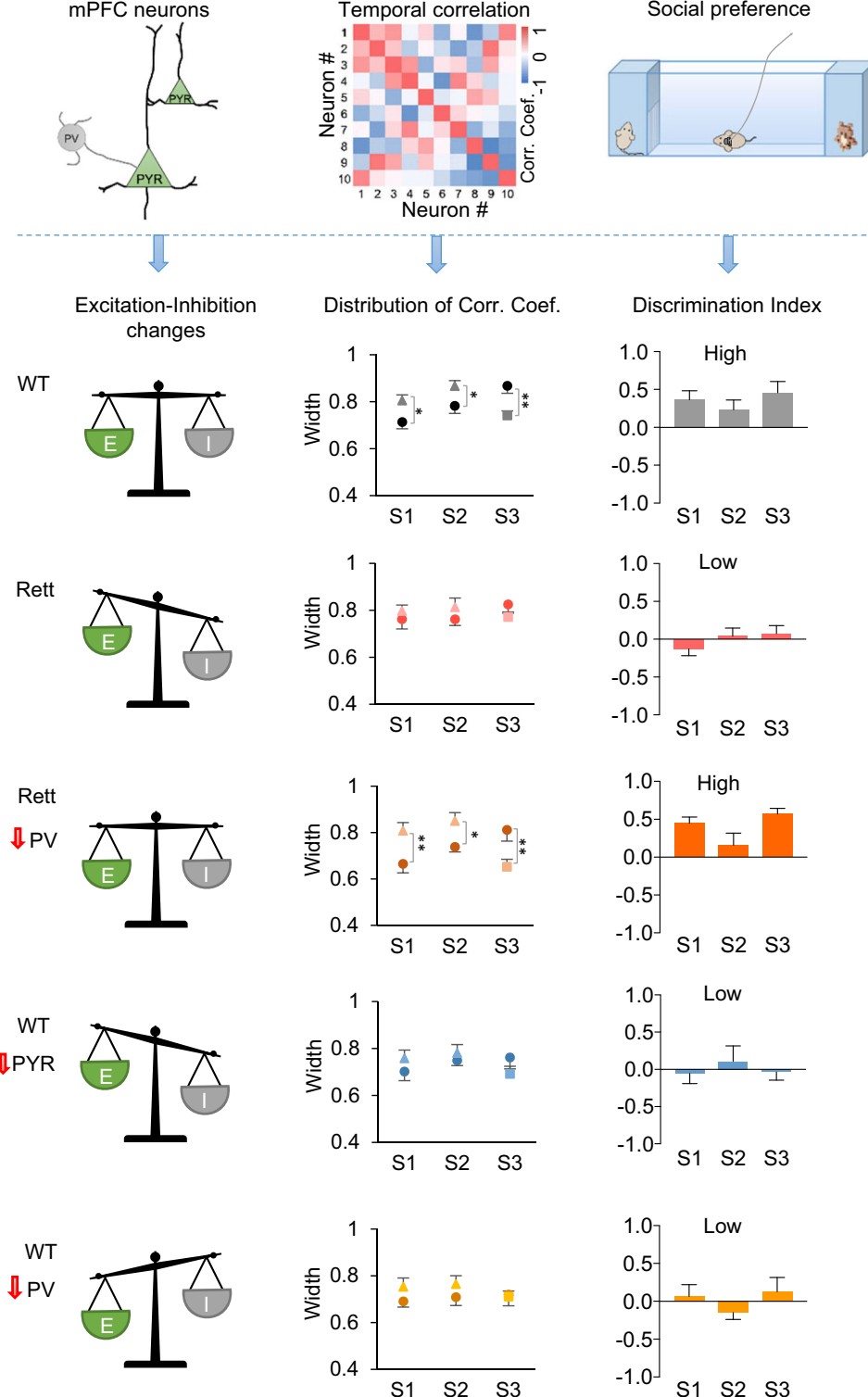

**Fig. 7 The relationship of changes in E/I balance at the cellular, circuit, and behavioral levels.** The cell-level E/I imbalance (left) in the mPFC impairs circuit-level pattern decorrelation, leading to the behavioral loss of social preference. The correction of cell-level E/I imbalance (left) in the mPFC of *Mecp2*-deficient Rett mice restores circuit-level pattern decorrelation and restores social preference. Source data are provided as a Source Data file.

the categorization of sensory stimuli and the refinement of social representations (provided by odor) in the mPFC[13].

In WT mice, there was less variation in neural correlations during the initial interaction with the social stimulus (M1 in S1 and M2 in S3) than during subsequent exposures (M1 in S2 and S3). This may be because more attention is devoted to a new

stimulus, and a more consistent inter-neuronal correlation likely improves the signal-to-noise ratio[34,35]. Over multiple interactions, the tail of the correlation distributions lengthened, as the circuit patterns for social representation became more stable in WT mice. This experience-dependent refinement of the mPFC circuit was absent in Rett mice. We also found that the Rett

mPFC was not able to streamline its information encoding across experimental sessions, indicating that the circuit was impaired in storing learned information. This is likely due to impaired synaptic plasticity; a previous ex vivo study demonstrated that loss of MeCP2 reduces excitatory postsynaptic currents, the ratio of NMDA to AMPA currents, and evoked population activity in the mPFC[20]. This electrophysiological data lend support to our observation of neuronal hypoactivity in the mPFC via miniscope calcium imaging, which can underestimate neural activity; future studies will be needed to confirm our results with in vivo electrophysiology, which would increase the temporal resolution of neuronal recordings and enable better understanding of how stimulus-specific patterns develop and differentiate through decorrelation[35].

Given that heterozygous Rett mice have mosaic expression of MeCP2 throughout the brain, and that MeCP2-negative neurons affect the development of neighboring MeCP2-positive neurons[47], it is possible that both cell-autonomous and nonautonomous effects are at play. We were unable to determine the genotype of specific neurons within the mPFC in our study, but one study using ex vivo physiology showed that MeCP2-deficient neurons in the female Rett mouse mPFC have increased inhibitory input, while the MeCP2-expressing neurons are similar to those in WT animals[21]. Even without knowing the genotype of individual neurons, however, it is clear that the MeCP2-deficient excitatory circuit as a whole is dampened in its responsiveness to stimulation. This limited range of responsiveness is evident in the motor circuit as well[27]. In addition, although we did not identify neural ensembles (groups of cells that fire together in the same circumstances over time) as Liang et al.[48] did when studying exploratory behavior in the mPFC, by studying the whole population of recorded neurons we were able to determine that pattern decorrelation is necessary for the expression of social preference. The failure of pattern decorrelation in the Rett mice suggests that their ability to form such ensembles is impaired.

In summary, our results demonstrated that prelimbic hypoactivity-induced failure of stimulus classification plays a causal role in the social deficit of female Mecp2 heterozygous mice, a physiologically relevant model of Rett syndrome. This study suggests that in Rett syndrome, if not in other autism spectrum disorders, the social behavioral deficits are not due to anxiety or avoidance, but to an inability to discriminate between social and nonsocial cues. Although multiple other brain regions besides the mPFC are involved in social interactivity[1], restoring mPFC function alone was sufficient to rescue the social abnormality of Rett mice. Given that a previous study found that activating the mPFC of Rett mice reverses respiratory symptoms[49], stimulating the mPFC could be a useful strategy to mitigate several core symptoms of Rett syndrome patients. Modulating the PFC might prove a strategy for treating other autism-associated disorders as well[50,51].

## Methods

Mouse maintenance and use were in accordance with NIH Guidelines and with the approval of the Institutional Animal Care and Use Committee of George Washington University.

**Animals**. Rett syndrome is caused by mutations in the X-linked gene MECP2. Because of X chromosome inactivation, both humans with Rett syndrome and female mouse models of Rett have mosaic expression of MeCP2 protein. *Rett* female mice on the 129S6SvEvTac strain were obtained from Dr. Huda Zoghbi's lab at Baylor College of Medicine, while Camk2-Cre and PV-Cre mice of pure C57BL/6 background were purchased from Jackson Lab (JAX#005359). Female Camk2-Cre, *Rett* mice and Camk2-Cre, *Mecp2+/+* (WT) mice were obtained by breeding male Camk2-Cre mice and female *Rett* mice; the wild-type and mutant offspring (the latter we call Rett mice) were used for excitatory neuron imaging. Similarly, male PV-Cre mice were mated with female *Rett* mice, and the resulting offspring were used for optogenetic manipulation experiments. We performed surgeries on WT and Rett mice at 3.5 to 4 months of age, then behavioral experiments at 5 months of age when the social abnormalities are apparent in Rett mice. Animals were given ad libitum access to standard mouse chow and water, housed 4 to 5 per cage in a temperature of ($23 \pm 1°C$) and humidity ($50 \pm 10\%$) controlled room with a 12 h light-dark cycle.

**Virus injection and GRIN lens implantation**. For imaging excitatory neurons, AAV-Efla-Flex-GCaMP6m (Baylor College of Medicine) was stereotaxically injected as previously described[52]. Briefly, 3.5–4-month-old Camk2-Cre mice (Rett and WT) were anaesthetized and placed in a stereotaxic frame (Neurostar, Tübingen, Germany), then a 1.1 mm-diameter craniotomy (AP: + 1.95 mm, ML: −0.5 mm) was made with a high-speed rotary stereotaxic drill (Model 1474, AgnTho's AB, Lidingö, Sweden). The virus was injected unilaterally (Nanojector II, Drummond Scientific) into the left region of mouse prelimbic cortex, with the stereotaxic coordinates from the bregma: +1.85 ~+2.00 mm anterior-posterior (AP), −0.35 mm medial-lateral (ML), −2.3~−2.5 dorsal-ventral (DV) following a high-resolution atlas[53]. A total of 600 nL virus (diluted with 600 nL PBS) were injected at a rate of 30 nL/min, and the needle was left in place for an additional 5 min after injection.

For optogenetic manipulation of PV neurons with simultaneous calcium imaging of pyramidal neurons, we injected 600 nL halorhodopsin AAV-Ef1a-DIO-eNpHR 3.0-EYFP (AAV-EF1a-DIO-EYFP as control virus, Addgene, Watertown, MA) bilaterally into the prelimbic region (PL, AP: + 1.85 ~ +2.00 mm; ML: ±0.35 mm; DV: −2.3~−2.5 mm) of the PV-Cre *Rett* and *Mecp2+/+* mice. Then, 600 nL of CaMKII-dependent virus AAV-CaMKII-GCaMP6m (obtained from the Neuroconnectivity Core at Baylor College of Medicine, Houston, TX) was injected into the left PL following the surgical protocol as described above.

For simultaneous optogenetic inhibition and calcium imaging of pyramidal neurons, the PL (AP: + 1.85 ~+2.00 mm; ML: ±0.35 mm; DV: −2.3~−2.5 mm) of the CaMKII-Cre WT mice were first injected with 600 nL AAV-Ef1a-DIO-eNpHR 3.0-EYFP bilaterally (AAV-EF1a-DIO-EYFP as control virus, Addgene, Watertown, MA), then with 600 nL AAV-ef1a-Flex- GCaMP6m (Addgene, Watertown, MA) in the left region. The injection was conducted complying with the protocol above.

For observation of PV neuron activity, 600 nL halorhodopsin AAV-ef1a-Flex-GCaMP6m (Addgene, Watertown, MA) was injected to the left PL (AP: + 1.85 ~ +2.00 mm; ML: −0.35 mm; DV: −2.3~−2.5 mm) of PV-Cre Rett and WT mice, following the above surgical protocol.

Then, a 1 mm diameter, 4 mm length GRIN lens (Inscopix, Palo Alto, CA) was implanted into the left prelimbic region (AP: + 1.95 mm; ML: ± 0.35 mm; DV: −2.1~−2.3 mm), 0.2 mm above the virus injection site without tissue aspiration. It was lowered at a speed of 50 µm/min, and then cemented in place (Metabond S380, Parkell). Mice were kept warm on a heating pad during recovery from surgery, and we monitored them closely for 7 days afterward, during which they received injections of analgesics.

For imaging excitatory neurons in the primary motor cortex, AAV-Efla-Flex-GCaMP6m (Baylor College of Medicine) was stereotaxically injected into the forelimb area of the right motor cortex (AP: + 1.5 mm, ML: + 1.5 mm) at depths around 200−300 µm to reach L2/3 neurons. 300 nL of virus (diluted with PBS) was injected at a rate of 30 nL/min, and the needle was left in place for an additional 5 min after injection. After gently removing the dura and rinsing the brain window clean, we implanted the GRIN lens on the surface of the right motor cortex and cemented it in place. Mice were kept warm during and after surgery; we monitored the mice closely for 7 days afterward and gave them injections of analgesics until they were completely recovered.

**Baseplate attachment**. Three to four weeks after surgery, we checked viral expression in the anesthetized mouse with a miniaturized microscope (Inscopix, Palo Alto, CA). If GCaMP+ neurons were visible and clear, the microscope attached with a baseplate would be positioned on the mouse's skull window and adjusted to assess the optimal focal plane. Then the microscope was detached and the baseplate was dental-cemented onto the skull and capped with a cover. Before the behavioral test, mice were habituated to the environment of the test room with the dummy microscope mounted and handled for 5–7 days, 40 min each day.

**Selection of animals for imaging**. Mice were selected based on the following criteria: (1) for combined imaging and behavioral experiments, mice were posthoc excluded if the lens was placed outside of the prelimbic cortex, or the imaging plane was unclear or occluded by blood or debris. (2) For combined imaging and optogenetic manipulation experiments, mice were excluded if they did not exhibit NpHR or EYFP expression in prelimbic regions, or major virus expression was detected outside of the prelimbic region, or the lens was not correctly placed.

**Determination of sample size**. The sample sizes for behavioral experiments were determined by the current standard used for mice in behavioral neuroscience experiments, based on the minimal number of mice required to detect significance

with an α rate set to 0.05 in a standard-powered experiment with a statistical power of 80% or better.

The sample size for the number of neurons was 62-129 neurons expressing GCaMP6m for each individual mouse (mean = 116), and was affected by the volume of viral GCaMP6m injected, level of expression, and the efficiency of Cre-mediated recombination. After the processing of the imaging data (after neuron identification by principal component and independent component analyses (PCA-ICA)), 1~9 % of the identified components were discarded as artefacts and the remainings were treated as neurons and used for further analysis.

**Blinding and randomization.** Experimenters were blind to mouse genotype during experimental sessions. Analysis of behavioral and neural data was performed by an experimenter blinded to the group assignment of the animal. Computer-based analyses ensured unbiased data collection and analysis in most cases.

In combined imaging and behavioral experiment, CamKII-Cre mice or PV-Cre mice were assigned to experimental groups based on their genotypes. In optogenetic experiments, PV-Cre Rett and WT mice were randomly assigned to the EYFP and NpHR groups, and two kinds of optogenetic manipulation-involved social tests were interleaved between two groups.

**Behavioral tests.** Behavioral tests were conducted after habituation when mice were 5 months old. Each day only one behavioral test was conducted, and the chamber was cleaned with 70% ethanol between trials for each test. The Topscan behavior analysis system (Clever Sys, VA) was used to monitor animal behaviors, which would send a TTL signal simultaneously to trigger the microscope to record neural activity at the beginning of each test.

*Social interaction test.* This test was carried out as previously described[6] but with some refinements. To facilitate in vivo imaging, the conventional three-chamber apparatus was modified to one open box (45 × 10 × 20 cm) with two small removable lateral chambers (10 × 10 × 20 cm), and chambers were separated by 1 cm-spaced thin metal wires, allowing mice to interact with stimuli. Each test consisted of three 10-min sessions, which were conducted following 10 min habituation in the center. At the beginning of each session, the testing mouse was placed in the center of the open box and allowed to explore freely. In session 1 (S1), an age- and weight-matched same-sex conspecific (the first social stimulus, M1) and a centrifuge tube (object, O) were separately placed in the end-chambers randomly. In session 2 (S2), the positions of those two stimuli were switched, which was designed to diminish the spatial influence in stimulus-induced neural activities. A healthy mouse prefers social over inanimate stimuli. In session 3 (S3), a new age- and weight-matched same-sex conspecific (new social stimulus, M2) replaced the object in the lateral chamber to evaluate social novelty preference; wild-type mice spend more time with a new conspecific than with a familiar one. The time spent involved in social interaction, object interaction, social zone, object zone, and middle zone were measured.

*Open field test (OFT).* This test was conducted in a square box (dimensions: 50 × 50 × 50 cm). The mouse was gently placed in the central field and allowed to explore freely during the 10 min testing session. Locomotor activity was recorded by the camera. The center is defined as the central 25 cm × 25 cm square area, while the corner sectors are the areas within a 12.5 cm radius from each corner. Total distance traveled and time spent in the center or corner of the box were recorded.

*Place preference paradigm.* In this assay, the open-field square box was separated into stimulation and non-stimulation zones by a wall, and a door in the middle of the wall allowed mice to move freely between the zones. The locations of the two zones were counter-balanced. Optogenetic LED light was delivered automatically to mice only when they entered into the stimulation zone, whereas the excitation LED was kept on during the 10-min testing session. The movement and duration of mice in different zones were recorded and compared.

**Calcium imaging with miniature microscope.** We use a head-mounted miniaturized microscope (nVista 2.0 and nVoke 1.0, Inscopix, Palo Alto, CA) to record the GCaMP6m fluorescent signals from mPFC neurons. The increase of GCaMP6m signal reflects a burst of spikes/action potentials[54]; the number of spikes has been experimentally determined for certain cell types[26], though not, as yet, for mPFC neurons. The microscope was mounted onto the mouse's head right before imaging and was triggered by a TTL pulse from the Topscan system to simultaneously acquire fluorescent signal and behavioral video. The imaging data were acquired at a frame rate of 15 Hz and at 1024 × 1024 pixels. The LED power was set to 0.3–1 mW and the gain was 1 to 2 depending on fluorescence intensity.

**Optogenetic manipulation during social interaction test.** There were two strategies of optogenetic manipulation modes: in the spatial-specific mode, the optogenetic light was delivered once mice entered into the social zone (new social zone in S3), which was triggered automatically by a signal from Topscan behavior

acquisition system. In the whole-arena mode, the optogenetic intervention was given no matter where the mice were, throughout the entire 10 min session. These two strategies of optogenetic manipulation were applied to the same group of mice with a one-month interval between the two tests, and they were counterbalanced between mice groups.

The nVoke imaging system (2.0, Inscopix, Palo Alto, CA) was used for combined optogenetic and imaging experiments, with two LED lights transmitted through GRIN lens into the prelimbic region. The excitation LED of blue light (448 nm) was set to 0.5–1.0 mW for GCaMP imaging, with an analog gain of 1.0–2.0, whereas the optogenetic LED of amber light (590 nm, 3.5 mW, continuous) was used to inhibit the activity of neurons expressing NpHR.

Note that these stimulation levels are low compared to many studies, as we sought to elevate excitatory neuron activity only by ~20%, in order to reach wild-type levels. In this study, we evaluated the effect of optogenetic suppression of PV neurons by observing the activity of excitatory neurons, rather than exploring how different optogenetic light intensities might recruit PV neurons to different degrees. Nonetheless, to ensure prolonged light exposure did not exert functional effects, we investigated mPFC activity properties of 3 min periods (without manipulation), both before and after the 10-minute social interaction test, combined with optogenetic manipulation in both EYFP control and NpHR-expressing mice. We found no significant difference in the transient rate or amplitude between EYFP and NpHR mice, either before or after recordings (Supplementary Fig. 8b). This indicates that the neural activity of NpHR mice recovers to normal levels after ten minutes of optogenetic manipulation, without functional damage. Moreover, since we compared the NpHR mice to control EYFP mice, which both received the same amount of light stimulation, any behavioral or neural differences cannot be due to light exposure.

**Histology.** Recording sites were verified by histological examination of lesions induced by lens implantation. Mice were anesthetized (i.p.) with an overdose of ketamine (400 mg/kg) and xylazine (20 mg/kg), then perfused transcardially with phosphate buffer solution (PBS) followed by 4% paraformaldehyde (PFA). Mouse brains (with skulls and baseplate) were post-fixed with 4% PFA for 3 days. Then, brains were removed and sliced into 50-100 μm sections using a Vibratome Series 1000 (St. Louis, MO) and mounted on slides. Slides were incubated and stored in 1:1000 Hoechst in 1× PBS (Invitrogen, Carlsbad, CA) to label cell nuclei. Slides were observed and imaged on a fluorescence microscope (Leica, DM6000, Buffalo Grove, United State) or a confocal microscope (Zeiss LSM 710, Oberkochen, Germany) and fluorescence of viral expressions and location of GRIN lenses could be recorded.

**Data analysis**

*Behavior.* Behavioral data were automatically tracked by top-down movies using Topscan behavioral data acquisition software (Clever Sys, Reston, VA). The software tracked the 2D locations of mice in throughout all five zones of the chamber (the anxiogenic central zone, the social zone, object zone, social sniffing zone, and object sniffing zone; see diagram in Fig. 1a). The orientation of the mouse's nose toward either object or fellow mouse while in the "sniffing zones" was also recorded. The software recorded the type and duration of behavior involved in different tests, including social or object interactions (mice sniffing stimuli with their nose within the sniffing area), grooming in a social zone, and approaching M1, M2, or O.

*Calcium image processing.* All calcium imaging movies were processed using the Inscopix Data Processing Software (v1.2.1)[55]. The video files collected within one experimental day were concatenated and processed as a single video (15 Hz, Frame size 1280 × 800 pixels, 1050 × 650 μm, 40 mins). We spatially cropped the video to match active ROIs and used the 3 × 3 pixel median filter to fix defective pixels. Then we temporally downsampled the videos (5 hz) to reduce the file size and used spatial bandpass filtering to remove low- and smoothen high-spatial frequency components from the movies. Next, we motion-corrected the movies using mean projection images as references. We calculated the ΔF/F value of a pixel at a given time by the following formula:

$$Fpix'(x, y, t) = \frac{F(x, y, t) - F\text{baseline}(x, y)}{F\text{baseline}(x, y)}(x, y, t) \qquad (1)$$

where F($x$, $y$, $t$) is the value at pixel coordinates ($x$, $y$) of frame t of the movie. The baseline fluorescence value for each pixel $F_{baseline}$($x$, $y$) was based on the minimum pixel fluorescence across all time frames.

We identified regions of interest (ROI) by combining Principal Component Analysis and Independent Component Analysis for spatial/temporal unmixing (PCA/ICA). We estimated the number of independent components as 15% over the number of cells, through visual inspection of the movie maximum projection images in ImageJ. We refined the ROI by manual inspection of raw movie and cellular trace quality and excluded: (1) ROI less than 4 pixels or without the neuronal soma shape; (2) traces that contained >2 components or a signal-to-noise ratio (SNR) of <3; (3) one of two neighboring neurons if the distance between centroids was <5 pixels. On average, we identified 116 neurons (range 62–129) that expressed GCaMP6m in each mouse after excluding ~30–40% of detected ROI.

Time-stamped traces of neurons were exported to Python (v3.8) for analysis onwards. $Ca^{2+}$ transients were identified using a peak-finder algorithm for each cell, then frequency (transients/min) and amplitude ($\Delta F/F$) data were processed using custom-written scripts. To determine the transient frequency we aligned the frames of the image with the behavioral data so that we could count the number of transients that took place when the mouse was in each of the 5 zones (social/object sniffing zones, social/object zones, and the middle zone—see diagram in Fig. 1a). For each recorded neuron, we divided the number of events counted in each zone by the time the mouse spent in that zone; this yielded 5 averaged data points for each recorded neuron. We then averaged the transient rates for all recorded neurons of each subject mouse in a given genotype by zone, so that we had five average transient rates for the WT and Rett groups. We then normalized these five averaged transient rates to the maximal value of the average transient rates in the WT group. We then produced a smoothed curve from these five averaged rates using BSpline and make_interp_spline in the spicy package of python (Fig. 2e).

*Identification of neurons that showed significant responses during social interaction.* To identify the groups of neurons responsive to social interaction in each session, we evaluated the response preference of each neuron to one specific stimulus interaction by comparing this activity level with the neuron's own chance level of activity[48,56]. In brief, we calculated the actual similarity (Sa) between vectors of calcium trace (ck) and behavior interaction (b) using the formula: $2b \cdot c_k/(|b|^2 + |c_k|^2)$[48]. Then, the behavior vector was randomly shuffled to calculate a new similarity (Ss) with a neural trace for a neuron, which was repeated 5000 times in order to generate the Ss distribution histogram. The neuron was classified as a social interaction-responsive neuron if its Sa was greater than 99.95% of the Ss distribution; the proportion of socially-responsive neurons in each session was calculated for each mouse.

*Neural correlation.* To evaluate the functional connectivity between neurons, we calculated the average of pairwise Pearson correlation coefficients of calcium traces in the neural population that corresponded to the duration of a specific behavior. To observe the correlation between highly correlated neurons, we used only the top 5% of correlation coefficient values.

*Variational autoencoder (VAE) analysis of neuron images.* To characterize the dynamics of neural activity within the image frames, we used a variational autoencoder (VAE), an unsupervised machine learning algorithm, to extract the low-dimensional latent features of the images. VAE sets up a neural network to search for putative Gaussian-distributed features that can best reconstruct the input images in the learning process. To do this, the learning process optimizes two loss functions simultaneously: the loss function of KL divergence to enforce Gaussian-distributed latent features and the mean-square-error (MSE) loss function to minimize the difference between the original input images and the reconstructed images[28,57]. In this study, we assumed only two latent features for 2D projection of these images. To reduce background noise, the images were preprocessed before VAE was applied by keeping the neural activity value only for neuron pixels in the images and setting the value to zero for all other pixels (such as those belonging to neuropils). We further scaled down the image size to reduce computational cost.

The VAE model was implemented using Keras (https://keras.io/examples/variational_autoencoder/). A typical VAE architecture was adopted following that used for the MNIST database of handwritten digits (http://yann.lecun.com/exdb/mnist/). The encoder part of the VAE model was set with two convolutional layers with 32 and 64 filters of size $3 \times 3$ and stride 2, respectively, followed by a fully connected layer of 16 neurons and the latent embedding layer of 2 dimensions (i.e., 4 neurons for two means and two variances). The decoder was the reverse of the encoder. During model training, i.e., optimization of two-loss functions of MSE error and KL divergence, we used the optimizer RMSProp[58]. We chose a batch size of 128 frames and repeated the training 200 times, which was sufficient for the loss for all samples to converge with little fluctuation. Other parameters follow the default setting of Keras. The frame projections in the 2D latent space were used for downstream analysis. In the 2D projection of frames in each session, the center location of the frames in the 2D space was calculated, and the distribution range (radius) was defined as the average distance of all frames to the center in the 2D latent space.

**Statistics**. All statistical analyses were performed using SPSS Statistics (version 24, IBM, Armonk, NY), Excel (Microsoft, Redmond, WA), and Python custom scripts. Since all the data could pass the normal distribution test (D'Agostino and Pearson), a two-tailed paired sample or unpaired *t*-test was applied in comparison. For multiple-factors comparison, two-way RM ANOVA was used, followed by Bonferroni–corrected posthoc comparisons. Statistical significance was taken as */#$P < 0.05$, **/##$P < 0.01$, ***/###$P < 0.001$. All data are represented as mean ± SEM unless otherwise specified. The linear curve was fitted to indicate transient rate-behavior correlation, and the correlation was tested by regression analysis. The supplemental table includes the full statistical methods and values for each related figure panel.

**Reporting summary**. Further information on research design is available in the Nature Research Reporting Summary linked to this article.

## Data availability
The data generated in this study are provided as a Source Data file with this manuscript, and video files are publicly available at https://github.com/Lulab-GWU/Social-MS. Source data are provided with this paper.

## Code availability
The codes used for analysis in this manuscript are publicly available at https://github.com/Lulab-GWU/Social-MS.

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

## Acknowledgements

We would like to thank Dr. Huda Zoghbi at Baylor College of Medicine for providing the Mecp2-deficient mice. We also thank Ethan Jin for significant input in data analysis and V.L. Brandt for probing discussions and critical comments on the manuscript. This work was supported by funding from NIH grant 5R00NS089824 (H.L.), the Brain & Behavior Research Foundation 2017 NARSAD Young Investigator Grant (H.L.), and The George Washington University 2018 Cross-Disciplinary Research Fund (H. L and C.Z.).

## Author contributions

P.X., Y.Y., and H.L. designed and performed the experiments. P.X., J.S., X.S., H.D., Z.L., and J.Z. analyzed data. P.X., Y.Y., J.S., X.S., H.D., Z.L., R.S., J.Z., C.Z., and H.L. reviewed and interpreted data. P.X., J.S., X.S., H.D. and Z.L., and H.L. wrote the manuscript with critical input from Y.Y., R.S., and C.Z.

## Competing interests

The authors declare no competing interests.
