## [Peer Review File · Nature Communications]

Reviewers' Comments:

Reviewer #1:

Remarks to the Author:

The manuscript by Xu et al studied the prelimbic neural activity during social behaviors of the MeCP2-deficient female mice, using *in vivo* calcium imaging recording and optogenetic manipulation. They concluded that in RTT mice (a mouse model of autism), mPFC hypoactivity impairs the decorrelate patterns for social and non-social stimuli, and optogenetically silencing PV inhibitory neurons can reverse behavioral deficits.

Major points:

1. Some data presented in this manuscript are not well organized, and even somewhat confusing. 1a) One of the main conclusions --"Pyramidal neurons in the mPFC of RTT mice are hypoactive during interactions" is subtitled on page 6. The authors actually started to present hypoactivity data before this subtitle. For instance, "RTT ensembles, however, especially social-ON and new social-ON ensembles, consistently showed a lower amplitude of response" (Page 5, line 10). What is confusing is that, in the supplementary figure 6a, the amplitudes of Social-ON ensembles in WT and RTT mice appear very similar.

1b) "Actually, when we considered only the immediate vicinity of one stimulus (where interactive sniffing happened), the corresponding stimulus-tuned ON ensemble displayed a much lower transient rate in RTT mice than WT" (page 5, third paragraph, line 5-7). Again, this data was presented before the "hypoactivity" subtitle. In addition, in Figure 3a, it looks like ALL neurons in RTT mice display reduced transient rates during interactions than that of WT mice. Is the hypoactivity a general feature for all PFC neurons in RTT mice? Or is the hypoactivity specific for particular neuronal ensembles? What if comparing the firing rate/amplitude of WT and RTT mice during other behavioral tests such as open field test? Another issue is that the authors did not explain in methods how they did the normalization for transient rates.

2. Optogenetic rescue only works when continuously silencing PV inhibitory neurons in RTT mice throughout the entire 10 minute recording. This observation seems suggesting that the deficits in RTT mice may not be behaviorally specific, but rather, for instance, due to generally increased inhibitory neural activity in the PFC of RTT mice. An important control is to perform the same optogenetic approach to testing the behavioral consequence of silencing PV neurons in WT mice.

3. The authors cited the previous work from Levy et al about impairments in decorrelation of social vs non-social stimuli from another autism mouse model, CNTNAP2 KO mice. However, in this model, instead of silencing PV neurons, previous optogenetics works from Selimbeyoglu et al, demonstrated that enhancing PV neural activity in the mPFC of CNTNAP2 KO mice rescued their social behavior deficits (Sci. Transl. Med. 9, eaah6733 (2017)), while optogenetically enhancing PV neural activity in the mPFC of WT mice did not change the social behavior (Yizhar et al, Nature 477, 171-178 (2011)). The authors should discuss why opposing regulation of PV neurons both achieve rescue effects for autism mouse models.

4. Some figures are missing and many figure panels do not match the figure legends

4a) Supplementary Fig 2:

The entire figure legends do not match the figure panels at all.

4b) Figure 2C:

Figure legend states: "Vertical pink bars indicate discrete episodes of social interaction." However, there is no indication (pink bars) for the social behavior episodes.

Minor points:

1) Page 4, first subtitle: RTT mice lack a preference for social novelty

From the results, it looks like RTT mice lack not only the preference for social novelty, but also sociability.

2) Figure 1b:

Raster plots should separately display both the time spent in M1 and O during S1 and S2, as well as, in M2 and M1 during S3 sessions, for WT and RTT mice, respectively.

Bar graphs should clearly specify the statistical significance between which groups (bars). In order to make points for deficits in sociability and social novelty in RTT mice, the comparison should be between M1 and O during S1 and S2 session; and M1 and M2 during S3 session for RTT mice.

Reviewer #2:

Remarks to the Author:

In this study, Xu et al. set out to examine the relationship between neuronal ensemble activity in the medial prefrontal cortex (mPFC) and behavioral impairments in a social interaction task of MeCP2-het mice. The investigators used calcium imaging with Inscopix head-mounted miniscopes to record from ensembles of neurons while freely moving mice chose to interact with an object, a familiar mouse, or a novel mouse. They found that excitatory (exc) neurons in mPFC of RTT mice are hypoactive compared to those of wild type controls. By modestly raising the activity of exc neurons using in vivo optogenetics, they restored normal behavior in RTT mice. These are challenging experiments for sure. Some of the data is quite interesting, in particular the finding that the rate of calcium transients in mPFC correlates with the % of time that mice spend visiting the familiar mouse (M1), but not an inanimate object (Fig. 3d) [although it would be nice to know if the correlations persist for either genotype alone]. Overall, there are many strengths. The study is timely and significant because very few others have recorded network activity in the brain during behavior in models of autism, and therefore little is known about the role of the mPFC in social preference. The authors use sophisticated analyses of network dynamics that provide clues as to how exc neurons in mPFC encode social exploration. On the other hand, I have concerns about the statistics, about missing controls for some experiments, and about the interpretation of results. If the authors can address the issues I raise below, I would be enthusiastic about publication.

1. Regarding the significance, the finding of hypoactivity in exc neurons in RTT mice is not novel (and they acknowledge this). One point the authors emphasize is that the activity of exc neurons is not just generally reduced, but that the problem is specifically that they are not modulated by specific object/mouse explorations. I'm not convinced about this interpretation. In Fig. 3a it looks like activity is depressed throughout the task. Moreover, the optogenetic rescue strategy was only successful when light was applied throughout the entire session. An important control would be to show mPFC activity during open field exploration to demonstrate that mPFC hypoactivity is specific to social interaction. A second important control would be to record from a neutral brain region that has little to do with social interaction (eg. visual cortex) during the same social assay, to rule out that hypoactivity is not just a global phenomenon in RTT mice.
2. We also don't know why exc neurons are hypoactive. Some mechanistic investigation of the cause of this network phenotype would have gone a long way as far as raising the significance. Is it because of hyperactive inhib interneurons?
3. The authors conduct further analyses on the calcium data to demonstrate pattern decorrelation when mice are interacting with a more 'attractive' social stimulus. This is interesting, but to show causality, the authors would need to interfere with decorrelation in WT mice and show that it is sufficient to replicate deficits seen in RTT mice.
4. Ultimately the bulk of their conclusions rests on a single measure of activity, miniscope calcium dynamics, which suffers from serious limitations. Confirming some of the main findings with a different technique (eg. silicon probes) would help a lot.
5. I have concerns about the statistics. In Fig. 2f and 2g (and in Fig. 4f/4h), they appropriately use an ANOVA test for the multiple comparisons, but in Fig. 4c-d they use a student t-test for a similar comparison (they should use ANOVAs throughout). Also, what statistical test was used in Fig. 3e (again, it should be an ANOVA). In the case of Fig. 3c, it does not seem appropriate to use a t-test, since these are multiple comparisons across WT and RTT mice. It's also not clear whether the data are normally distributed. If not, they must use non-parametric test instead of ANOVA. In Suppl. Fig. 6, it seems they are using $n = \#$ neurons (rather than $\#$ of mice) as the sample size for statistics which does not seem appropriate. Elsewhere in the paper it seems they appropriately

compared individual mice to each other and did not lump all the neurons. Or did they? Can the authors compare that stats used $n = \#$ mice for comparisons in Figs. 2f, 3a, 3c, 3e, 3h, 4c-d, 4f, 5d, 5i, and 5k-l.

6. There have been recent concerns about thermal impact of light stimulation for optogenetic manipulations (Owen et al., Nat Neurosci 2019). This would be particularly worrisome for prolonged light exposure as was done in this study. To diminish this concern, it would be important to show before-and-after transient rates for EGFP and NpHR mice separately. Control experiments in brain slices with ephys of PV neurons after 1 month of NpHR expression and exposed to 10 min long light stimulation seems necessary.

7. Here are other concerns related to individual figures:

For Fig. 1, we need to know the number of seconds that mice spent in each zone. It is possible that RTT spend more time in the middle, or alternatively that they move from side to side very quickly. It is possible that the authors presented this data in Suppl. Fig. 2 but unfortunately, they reproduced Suppl. Fig 1 twice instead. It is not sufficient to only represent the data as a percentage of time. It would be very helpful to show videos of examples of how wild type and RTT mice behave in this task. Aren't MeCP2het mice known to have reduced locomotion?

For Fig. 2, what was the average number of neurons that they successfully recorded from in mice (they say 60-120 neurons per animal, but what was the exact range, and the mean)? Also, for many of the selected ROIs labeled as "other" were they active or inactive, and on average what % of ROIs were in this 'other' category? In Fig. 2b the traces need a scale bar for time. In Fig. 2c I could not see vertical bars to indicate the epochs of social interaction. It was not possible for me to interpret this figure. When they say "...by S3, RTT mice showed a greater % of social-ON cells..." this was not significant, so they should make that clear. I did not understand panel e. Why is activity so high before the onset of interaction in the social OFF panel? Are shaded bars in panel f the s.e.m. of all neurons or all mice?

In Fig. 5 it is surprising that complete silencing of PV neurons leads to only a very modest increase (20%) in the frequency of calcium transients for exc neurons and no change in amplitude of transients. It's also surprising that the effect is not global, and only occurs with certain interactions (M1, M2). In Fig. 5h, it sounds like blue LEDs were turned on for the entire 10 min and this should be explicitly stated in the Results section too (not just Methods). [In general, the Methods section was missing important details.] It seems critical that the authors record from excitatory neurons in vivo (silicon probes) to confirm that activity was modulated by such prolonged optogenetic manipulation (is it possible that miniscopes underestimate the effect?) Were there any adverse effects of silencing PV neurons in mPFC, seizures perhaps, hyper-locomotion?

Suppl. Fig. 3a does not make sense to me. Is there really an extremely brief increase in DF/F in all 1600 neurons? One would have expected to find a sequence of cell activations before, during and after the social interaction, but some cells would have fired for longer than others. Also, why is the hypoactivity of mPFC neurons in RTT mice not obvious from these pseudocolor raster plots? It would be important to show similar raster plots for social interaction vs. object interactions to see the differences

In Suppl. Fig. 4a it seems several of the differences the authors point to are not significant, like social ON in WT (34.83) vs. RTT (29.01) or S2 vs. S3 within RTT mice (29.01 vs. 34.56). This should be stated in the text.

Typos and other small things:

- Abstract: I think some people may object to the notion that their results relate to social avoidance. They are studying social preference and discrimination.
- Why did they select 5 months of age for the mice (as opposed to 2-3 months)? Is it because the deficits in social behavior on their task are not present earlier? In the original 3-chamber assay, older mice (~6 months) tended not to participate as much. Also, the Methods section says they were tested at 3.5-4 months.
- Page 5, the sentence "Whereas WT mice had significant changes in from one session..." needs to

be rewritten.

- They refer to “firing rates” (e.g., in page 6), but this is not appropriate since they only recorded frequency rates of calcium transients.
- Page 8: “dimension-deduction” should be ‘reduction’
- Page 9, “PV-neuron suppression increased the transient rate markedly” – not sure about the word ‘markedly’; Perhaps just ‘significantly’ is more appropriate (it’s a 30% increase at most in Fig. 5i)
- Page 10, “weakened, homogenized responses”; do they mean that the rate of transients is reduced throughout the various interactions (object and mouse; well, pretty much throughout the entire assay really)? If so, homogenized is not the right word. Maybe change to ‘homogenously weakened’, or “weakened, non-modulated’ responses?
- Page 14: The dimensions of the “square box” and the removable lateral chambers don’t make sense. 45x10x20 cm is not a square and the dimensions of the side boxes don’t fit with the main box. I suggest they draw the box to scale in Fig. 1a and Fig. 5c (and label the dimensions of each side)
- Page 15: The paragraph entitled “Social preference paradigm” should be rewritten; it is not very clear. How was LED light delivered only when mice entered the stim zone, was it done manually under direct supervision, or was it automated somehow?
- I was surprised that data in Fig. 5 was all from the same cohort of mice, which received the two patterns of stimulation in different sessions 1 month apart. It would be helpful if the authors labeled which mice underwent full-session stimulation first vs. last in different colors in Fig. 5.
- Perhaps they should cite and comment on a different paper using miniscopes to record activity in hippocampal CA1 in social behavior in the mouse model of MeCP2dup syndrome (Sun et al., Science Bulletin 2020, 65:1192-1202), if only to mention a similar study using miniscopes in PFC, in a RTT model, is a social task
- In the Discussion they should comment on the fact that, in several other models of autism, there is hypoactivity of PV interneurons, not exc neurons (PMID: 30679017, 30250263, 26830140), which is the opposite of what they find. Does their data help us understand symptoms of RTT but not autism?
- They should probably cite Sacai et al. Nat Comm (2020) PMID: 33046712
- Figure 5k has mislabeled RTT and WT when it should be NpHR and EYFP.

Reviewer #3:

Remarks to the Author:

NCOMMS-20-46394

Xu et al.

This manuscript describes excitatory neuron ensembles in the mouse PL-mPFC (identified by in vivo Ca²⁺ imaging) that were either active or inactive during interactions with either mice or objects in a modified 3-chamber arena, and their altered features in female *Mecp2* heterozygous mice. In addition, the authors show how optogenetic inhibition of PV interneurons in the PL-mPFC increases social interaction time in female *Mecp2* heterozygous mice. The manuscript is well written (although it lacks precision about cited work and the meaning of the observed variables), and the data seem to be of sufficient quality to support the authors’ interpretation. However, several issues should be addressed experimentally to better support the authors’ conclusions (which are sometimes a bit exaggerated in their extent and meaning).

Major questions (require additional experiments)

1. The brain of female *Mecp2* heterozygous mice shows a mosaic expression pattern of MeCP2 in all cells based on their random inactivation of the X chromosome, but the AAVs used to deliver GCaMP6m and eNpHR will transduce all cells equally. Therefore, the grouping of cells into ON ensembles using Ca²⁺ imaging traces in female *Mecp2* heterozygous mice lacks information about the cellular genotype of the imaged neurons, i.e. are they MeCP2-expressing WT neurons or MeCP2-lacking mutant neurons? This is critical information because the recruitment of MeCP2-expressing WT neurons and MeCP2-lacking mutant neurons into different ensembles could be different or not, and cannot not be assumed to be one way or the other. In addition, the intrinsic (passive and active) and synaptic properties of MeCP2-expressing WT neurons and MeCP2-lacking mutant neurons cannot be assumed to be the same (because they do differ, in different ways, in

different brain regions), and all those features have a significant role in the recruitment of neurons into ON ensembles and their drop-off in OFF ensembles. There are several studies using female *Mecp2* heterozygous mice that revealed unexpected cell autonomous and non-cell autonomous consequences of *Mecp2* loss in either neurons or glial cells. The authors need to find a way to know the cellular genotype of the imaged neurons, otherwise, the data from their "RTT mice" doesn't have much information.

2. The authors need to provide specific information about how many action potentials are reflected in each GCaMP6m transient ("spike" in their words), taking into account the frame rate of image acquisition and the response kinetics of the Ca²⁺ sensor. Without this information from WT mice and the two cellular genotypes present in *Mecp2* heterozygous mice, any reference to firing/spiking frequency does not carry much meaning, because it is unknown if all those parameters remain constant and similar in neurons from WT mice, as well as in MeCP2-expressing WT neurons and MeCP2-lacking mutant neurons in *Mecp2* heterozygous mice.

3. The authors need to provide direct evidence of what underlies the frequency of GCaMP6m transients, which they assume is different from what underlies the amplitude of each GCaMP6m transient (they seem to interpret that those two variables provide different information).

4. The authors need to perform optogenetic inhibition of PV cells in both WT and female *Mecp2* heterozygous mice, not only in WT mice.

5. Optogenetic inhibition of PV cells should be confirmed by optogenetically exciting pyramidal neurons (same end result, as they interpret their data and based on their model of mPFC hypoactivity). In addition, it should be complemented with optogenetic inhibition of pyramidal neurons, as well as optogenetic excitation of PV cells. This should be done in both genotypes (i.e. "improve the RTT mice, mimic RTT in WT mice"). Only then their model and conclusions would be experimentally supported.

6. The authors' interpretation of the difference observed when inhibiting PV cells either in the interaction zone or the entire arena ignores the complex dynamics that result from hyperpolarizing GABAergic interneurons that are widely connected by electrical synapses (gap junctions). The authors need to consider other potential consequences of inhibiting PV cells, including their rebound excitation, and the spread of the hyperpolarization itself and of the rebound excitation through gap junctions.

7. What was the statistical Power yielded by the sample numbers used (post-hoc)? Also, the authors should explicitly state if they were blind to genotypes and AAV injection conditions (for eNpHR), if mice were randomized to AAV injections (for eNpHR) and behavioral testings, and if they used specific criteria for data inclusion and exclusion. Sample numbers should include all mice used, not just the number of successful experiments that reached the analysis stage (which can be reported separately).

Minor questions (may not require additional experiments)

8. The genetic background of the *Mecp2* mutant mice (S129) is different than the one of CaMKII-Cre and PV-Cre mice (C57/Bl6). The authors need to address the potential contribution of the genetic background to the observed differences in ensemble dynamics and mouse behaviors.

9. All bar graphs should include all data points, or be replaced by scatter plots.

10. The authors need to be explicit about the genotype of the mice in the studies cited, because the majority of the cited work used male *Mecp2* KO mice and this is a major difference with the current study.

11. The text should be explicit in that the ensembles studied are made up of CaMKII-expressing cells, not all neurons in the mPFC (as implied in several parts of the manuscript).

12. The authors should be explicit about the limitations of extracting "neuronal activity" and "firing rates" from Ca²⁺ imaging data, which only provides a rough estimate by reflecting an envelope of

activity due to the combination of Ca sensor kinetics and imaging frame rate. They also need to be explicit about the meaning of reductions in GCaMP6m $\Delta F/F$ (e.g. Fig. 2f): does intracellular Ca^{2+} go below resting/baseline levels during that prolonged time?

13. The occurrence of GCaMP6 transients should be expressed as inter-event intervals (not Hz) and plotted as cumulative distribution probabilities because the mathematical average (i.e. mean) does not accurately reflect the intrinsic variability of the data set.

14. The reference to Phillips et al. 2019 is inaccurate: its title itself says exactly the opposite of what the authors wrote in the Introduction (page 5). The authors are encouraged to read carefully and completely all the papers they cite. In this case, they misrepresented the title itself; what other mistakes could be in other citations?

15. The authors need to explain why did they use Camkii-Cre mice if they injected AAV1.GCaMP6m, which does not require Cre recombinase for its expression (like the DIO construct they used for eNpHR expression in PV cells from PV-Cre mice).

16. It is unclear by there are eNpHR-expressing neurons in the mPFC that is contralateral to the side where the GRIN lens was implanted, as shown in Suppl. Fig. 11. The authors state that all their injections were on a single hemisphere and right under the GRIN lens.

The reviewers all provided very constructive comments, which we have been happy to address. For the sake of clarity, we present the reviewers' comments in *italics* and our responses in regular font. In-text citations are listed in full, alphabetically, at the end of this document.

Reviewer #1

The manuscript by Xu et al studied the prelimbic neural activity during social behaviors of the MeCP2-deficient female mice, using in vivo calcium imaging recording and optogenetic manipulation. They concluded that in RTT mice (a mouse model of autism), mPFC hypoactivity impairs the decorrelate patterns for social and non-social stimuli, and optogenetically silencing PV inhibitory neurons can reverse behavioral deficits.

Major points:

1. Some data presented in this manuscript are not well organized, and even somewhat confusing.

1a) One of the main conclusions -- "Pyramidal neurons in the mPFC of RTT mice are hypoactive during interactions" is subtitled on page 6. The authors actually started to present hypoactivity data before this subtitle. For instance, "RTT ensembles, however, especially social-ON and new social-ON ensembles, consistently showed a lower amplitude of response" (Page 5, line 10). What is confusing is that, in the supplementary figure 6a, the amplitudes of Social-ON ensembles in WT and RTT mice appear very similar.

1b) "Actually, when we considered only the immediate vicinity of one stimulus (where interactive sniffing happened), the corresponding stimulus-tuned ON ensemble displayed a much lower transient rate in RTT mice than WT" (page 5, third paragraph, line 5-7). Again, this data was presented before the "hypoactivity" subtitle.

We apologize for the unclear organization (which we have revised) and for inadvertently "mis-writing" in the quoted sentence: as the reviewer notes below, it's not the amplitude but the transient rate that is affected by the loss of MeCP2.

In addition, in Figure 3a, it looks like ALL neurons in RTT mice display reduced transient rates during interactions than that of WT mice. Is the hypoactivity a general feature for all PFC neurons in RTT mice? Or is the hypoactivity specific for particular neuronal ensembles? What if comparing the firing rate/amplitude of WT and RTT mice during other behavioral tests such as open field test?

This was an excellent question/suggestion from both Reviewers 1 and 2. We performed this experiment, monitoring the somatic Ca²⁺ activity of mPFC pyramidal neurons of WT and RTT mice as they underwent the open field test, and found that the hypoactivity of the mPFC depends on the stimulus and the location of the mouse in the chamber (**Supplementary Fig. 5b**). In WT mPFC, the average transient rate and amplitude were similar in the center and corners of the arena, but in the RTT mPFC, the transient rate was significantly lower than WT when the mouse was in the center area (but not corners). This result is included in the **new Supplementary Fig. 4e**.

Another issue is that the authors did not explain in methods how they did the normalization for transient rates.

We apologize for inadvertently omitting this information. We used a min-max normalization method to calculate neuronal activity between 0 and 1 and have added this to the Methods.

2. Optogenetic rescue only works when continuously silencing PV inhibitory neurons in RTT mice throughout the entire 10 minute recording. This observation seems suggesting that the deficits in RTT

mice may not be behaviorally specific, but rather, for instance, due to generally increased inhibitory neural activity in the PFC of RTT mice. An important control is to perform the same optogenetic approach to testing the behavioral consequence of silencing PV neurons in WT mice.

All three reviewers made this important request. We have now performed this experiment and found that suppressing (not silencing) PV inhibitory neurons of WT mice with NpHR (vs. EYFP as a control; **new Supplementary Fig. 12**) basically abolished the normal preference for interacting with another mouse, whether in the social zone or the whole area (**Supplementary Fig. 12a, d**). At the same time, pattern decorrelation between mPFC excitatory activity and social interaction also disappeared in NpHR mice, as the two stimuli (mouse and object) elicited similar levels of correlation while mice displayed no social preference (**Supplementary Fig. 12b-c, 12e-f**). Combined with optogenetic manipulation of the E/I balance in WT and RTT mice with different strategies, we conclude that the impaired social discrimination of RTT mice is related to a failure of pattern decorrelation in the mPFC.

Interestingly, both optogenetic inhibition of PV neurons (in female WT and RTT mice) and optogenetic inhibition of mPFC pyramidal neurons (in female WT mice) disrupts E/I balance, though in different directions. Both manipulations abolish pattern decorrelation of mPFC excitatory circuit activity, leading to an inability to discriminate social from nonsocial cues. The results of these experiments, summarized in the new **Fig. 7**, support the idea that there is a causal relationship between pattern decorrelation and social performance.

3. The authors cited the previous work from Levy et al about impairments in decorrelation of social vs non-social stimuli from another autism mouse model, CNTNAP2 KO mice. However, in this model, instead of silencing PV neurons, previous optogenetics works from Selimbeyoglu et al, demonstrated that enhancing PV neural activity in the mPFC of CNTNAP2 KO mice rescued their social behavior deficits (Sci. Transl. Med. 9, eaah6733 (2017)), while optogenetically enhancing PV neural activity in the mPFC of WT mice did not change the social behavior (Yizhar et al, Nature 477, 171–178 (2011)). The authors should discuss why opposing regulation of PV neurons both achieve rescue effects for autism mouse models.

This is indeed an important point to discuss. The Rett field has already learned that molecular/genetic defects that operate quite differently at the level of individual neurons can produce the very same functional defects at the circuit level. For example, Lu et al. (*Neuron* 2016) showed that both loss and gain of MeCP2 function (in RTT and *MECP2* duplication syndrome (MDS), respectively), cause the same hypersynchrony in the hippocampal circuit, which is rescued in both models by deep brain stimulation. This is despite the fact that the transcriptional profiles of RTT and MDS brains show opposite patterns of dysregulation of thousands of genes (Chahrour et al., *Science*), and RTT and MDS cause opposite patterns of mRNA expression, synaptic plasticity, synapse density, and dendritic arbor complexity. This shared circuit dysfunction could help explain why there is overlap between RTT and MDS at the behavioral level as well.

In the case of RTT mice and CNTNAP2 KO mice, it appears that opposite abnormalities in the mPFC circuit (hyperactivity vs. hypoactivity) can produce similar behavioral outcomes, in this case, altered social preference behavior. We and others have noted that the pattern of behavior and neuronal changes seen in the CNTNAP2 KO mouse is closer to that of MeCP2 duplication mice, whose mPFC has also been reported to be hyperactive. It seems that there is a "goldilocks" principle at play, where the proper functioning of the mPFC circuit needs to be brought within normal range in order to support normal social behavior. We now discuss this in the revised manuscript.

4. Some figures are missing and many figure panels do not match the figure legends

4a) Supplementary Fig 2: The entire figure legends do not match the figure panels at all.

4b) Figure 2C: Figure legend states: "Vertical pink bars indicate discrete episodes of social interaction." However, there is no indication (pink bars) for the social behavior episodes.

We apologize for these mistakes, which we have now corrected.

Minor points:

1) Page 4, first subtitle: *RTT mice lack a preference for social novelty*

From the results, it looks like RTT mice lack not only the preference for social novelty, but also sociability.

Agreed; we have changed the subtitle.

2) Figure 1b: Raster plots should separately display both the time spent in M1 and O during S1 and S2, as well as, in M2 and M1 during S3 sessions, for WT and RTT mice, respectively.

Bar graphs should clearly specify the statistical significance between which groups (bars). In order to make points for deficits in sociability and social novelty in RTT mice, the comparison should be between M1 and O during S1 and S2 session; and M1 and M2 during S3 session for RTT mice.

Agreed; we have revised accordingly.

Reviewer #2

In this study, Xu et al. set out to examine the relationship between neuronal ensemble activity in the medial prefrontal cortex (mPFC) and behavioral impairments in a social interaction task of MeCP2-het mice. The investigators used calcium imaging with Inscopix head-mounted miniscopes to record from ensembles of neurons while freely moving mice chose to interact with an object, a familiar mouse, or a novel mouse. They found that excitatory (exc) neurons in mPFC of RTT mice are hypoactive compared to those of wild type controls. By modestly raising the activity of exc neurons using in vivo optogenetics, they restored normal behavior in RTT mice. These are challenging experiments for sure. Some of the data is quite interesting, in particular the finding that the rate of calcium transients in mPFC correlates with the % of time that mice spend visiting the familiar mouse (M1), but not an inanimate object (Fig. 3d) [although it would be nice to know if the correlations persist for either genotype alone]. Overall, there are many strengths. The study is timely and significant because very few others have recorded network activity in the brain during behavior in models of autism, and therefore little is known about the role of the mPFC in social preference. The authors use sophisticated analyses of network dynamics that provide clues as to how exc neurons in mPFC encode social exploration....

We thank the reviewer for appreciating the strengths and challenges of the study, and their constructive suggestions for improving it.

In response to the reviewer's question, expressed in the underlined sentence above, we did not find significant correlations between the rate of calcium transients and the percentage of time spent visiting the familiar mouse (M1), and so did not include this data in the original manuscript. We now show these data in the **new Supplementary Fig. 4d**.

...On the other hand, I have concerns about the statistics, about missing controls for some experiments, and about the interpretation of results. If the authors can address the issues I raise below, I would be enthusiastic about publication.

1. *Regarding the significance, the finding of hypoactivity in exc neurons in RTT mice is not novel (and they acknowledge this). One point the authors emphasize is that the activity of exc neurons is not just generally reduced, but that the problem is specifically that they are not modulated by specific object/mouse explorations. I'm not convinced about this interpretation. In Fig. 3a it looks like activity is*

depressed throughout the task.

The reviewer is correct that Fig. 3a (now **Fig. 2c**) shows that the activity of excitatory neurons in the RTT mPFC is always somewhat lower than in wild-type, but the difference between genotypes was not statistically significant after correction for multiple comparisons except when the animal interacts with a stimulus (see also "social ON" neuronal activity in the left column of the new **Supplementary Fig. 5b**).

Moreover, the optogenetic rescue strategy was only successful when light was applied throughout the entire session. An important control would be to show mPFC activity during open field exploration to demonstrate that mPFC hypoactivity is specific to social interaction.

Both Reviewers 1 and 2 made this important suggestion. We performed this experiment and found that the hypoactivity of the mPFC depends on the stimulus and the location of the mouse in the open field chamber (**new Supplementary Fig. 4e**). As we mentioned above, in WT mice, the average transient rate and amplitude were similar in the center and corners of the arena, but in the RTT mice, the transient rate was significantly lower—in comparison to WT as well as in comparison to RTT being in a corner—when the mouse was in the center area (but not corners). Thus, the hypoactivity is most acute just when the animal is being stimulated, either by an interaction or by something anxiety-provoking. (Perhaps interactions are subtly anxiety provoking, i.e., maybe curiosity and anxiety exist along a continuum. At the very least, we can say that both involve heightened alertness.)

A second important control would be to record from a neutral brain region that has little to do with social interaction (eg. visual cortex) during the same social assay, to rule out that hypoactivity is not just a global phenomenon in RTT mice.

As noted in response to the previous question, the open field experiment demonstrated that hypoactivity is not a global phenomenon in RTT mice, but is specific to stimulation. Furthermore, mechanisms of E/I imbalance in *Mecp2*-deficient mice vary by region: the visual cortex shows inhibitory hyperconnectivity (Durand et al., *Neuron* 2012); the frontal-motor cortex shows reduced excitation without significant alterations in inhibitory input (Wood et al., *J Neurosci* 2009); both the hindbrain and hippocampus show hyperexcitability (Shepherd and Katz, *Curr. Opin. Neurobiol.* 2011; Calfa et al., *J. Neurophys.* 2011; Kron et al., *J Neurosci* 2012).

In general, we think hypoactivity induced by *MECP2* loss is stimulation-dependent, as would be consistent with MeCP2's role in experience-dependent plasticity (Krishnan et al., *PNAS* 2015; Lau et al., *J Neurosci* 2020; Robinson et al. *Learn Mem* 2019; Krishnan et al., *Nat Commun* 2017). We recently found that excitatory neurons in the motor cortex show lower-than-WT activity during running but not at rest (Yue et al. *Science Advances*, in press). Kee et al. (*eLife* 2018) also found a decrease in hippocampal activity that was specific to a familiar environmental context.

2. We also don't know why exc neurons are hypoactive. Some mechanistic investigation of the cause of this network phenotype would have gone a long way as far as raising the significance. Is it because of hyperactive inhib interneurons?

Yes: we recorded the neural activity of PV neurons in the mPFC of WT and RTT mice during the social interaction test, and found that the amplitudes around the onset of stimulus interactions were increased in both genotypes, which is consistent with previous reports using fiber photometry (Selimbeyoglu et al., *Sci Transl Med* 2017) and electrophysiological recording (Liu L et al., *Science Advances* 2020). Compared to WT, the discharge of PV neurons reflected by the area under the curve (AUC) per second of the calcium traces was significantly increased throughout the stimulus interaction, although the increased amplitude ($\Delta F/F$) upon the onset of social interaction was diminished. This indicates that PV inhibitory neurons are indeed hyperactive in the local mPFC circuit in RTT mice (**new**

Supplementary Fig. 10).

These results are consistent with previous reports of decreased excitatory synapse strength (Sceniak et al. *Cereb. Cortex*, 2016) in *Mecp2*-null mPFC without a loss of inhibitory synapse function, and excessive inhibitory activity and arborization in PV interneurons in the visual cortex of *Mecp2*-deficient mice (Patrizi et al. *Cerebral Cortex* 2020).

3. The authors conduct further analyses on the calcium data to demonstrate pattern decorrelation when mice are interacting with a more 'attractive' social stimulus. This is interesting, but to show causality, the authors would need to interfere with decorrelation in WT mice and show that it is sufficient to replicate deficits seen in RTT mice.

We have now inhibited pyramidal neuron activity in the mPFC using halorhodopsin (NpHR, with EYFP as a control) to interfere with the neural decorrelation in WT mice. This effectively abolished social preference in WT mice (**Fig. 6**), mimicking the social deficits in RTT mice. This finding supports a causal relationship between pattern decorrelation and social preference.

4. Ultimately the bulk of their conclusions rests on a single measure of activity, miniscope calcium dynamics, which suffers from serious limitations. Confirming some of the main findings with a different technique (eg. silicon probes) would help a lot.

The limitations were ours more than the technique's, and we trust that our new experiments, per the reviewers' constructive suggestions, eliminate this concern.

There is no better way, currently, to simultaneously visualize the Ca²⁺ transients in large populations of cells (e.g., Grienberger et al., *Neuron* 2012; Mohammed et al., *Scientific Reports* 2016). Electrophysiological recording is temporally precise but less likely to accurately record neurons with low activity, which would be a concern for MeCP2 mutant mice. Moreover, the relationship between spike trains and fluorescence signals is not straightforward: Wei et al. (*PLoS Computational Biology* 2020) simultaneously recorded from individual frontal cortical neurons and found clear differences between the methods that will take quite a lot of effort to resolve (they even set up an online resource to help neuroscientists compare ephys and imaging—<http://im-phys.org/>).

5. I have concerns about the statistics. In Fig. 2f and 2g (and in Fig. 4f/4h), they appropriately use an ANOVA test for the multiple comparisons, but in Fig. 4c-d they use a student t-test for a similar comparison (they should use ANOVAs throughout). Also, what statistical test was used in Fig. 3e (again, it should be an ANOVA). In the case of Fig. 3c, it does not seem appropriate to use a t-test, since these are multiple comparisons across WT and RTT mice. It's also not clear whether the data are normally distributed. If not, they must use non-parametric test instead of ANOVA. In Suppl. Fig. 6, it seems they are using n = # neurons (rather than # of mice) as the sample size for statistics which does not seem appropriate. Elsewhere in the paper it seems they appropriately compared individual mice to each other and did not lump all the neurons. Or did they? Can the authors compare that stats used n = # mice for comparisons in Figs. 2f, 3a, 3c, 3e, 3h, 4c-d, 4f, 5d, 5i, and 5k-l.

We apologize for not being clearer about our statistical methods—we did indeed use ANOVA throughout, including Fig. 3c and e (now **Fig. 2e** and **2g**). We have clarified our methods in the revised manuscript and created a supplementary table with full statistical methods and values for each figure panel.

The data were normally distributed, as determined by SPSS software (this information is included in the Methods).

For **Supplementary Fig. 6**, we used *n* for the number of mice, not neurons. We did not lump all neurons together in any experiment.

6. There have been recent concerns about thermal impact of light stimulation for optogenetic manipulations (Owen et al., Nat Neurosci 2019). This would be particularly worrisome for prolonged light exposure as was done in this study. To diminish this concern, it would be important to show before-and-after transient rates for EGFP and NpHR mice separately. Control experiments in brain slices with ephys of PV neurons after 1 month of NpHR expression and exposed to 10 min long light stimulation seems necessary.

Thank you for raising this important point. Owen et al. reported that 'commonly used' optogenetic stimulation protocols at 10-30 mW of power led to increases in temperature of 0.2-2 °C within a few minutes, and they test and model lower powers of 3-15 mW. Note that our LED power was set to only 0.3-1.0 mW, which is quite a bit lower—we were, after all, only trying to stimulate neurons 20% to raise their activity to WT levels. This is different from many optogenetic experiments, whose goal is a more dramatic change.

Nonetheless, to explore the influence of prolonged light exposure, we investigated mPFC activity properties of 3-min periods (without manipulation), both before and after the 10-minute social interaction test, combined with optogenetic stimulation in both EYFP control and NpHR-expressing mice. We found no significant difference in transient rate or amplitude between EYFP and NpHR mice, either before or after recordings (**new Supplementary Fig. 11j**). This indicates that the neural activity of NpHR mice recovers to normal levels after ten minutes of optogenetic stimulation, without functional damage. Moreover, since we compared the NpHR mice to control EYFP mice, which both received the same amount light stimulation, any behavioral or neural differences cannot be due to light exposure.

7. Here are other concerns related to individual figures:

For Fig. 1, we need to know the number of seconds that mice spent in each zone. It is possible that RTT spend more time in the middle, or alternatively that they move from side to side very quickly. It is possible that the authors presented this data in Suppl. Fig. 2 but unfortunately, they reproduced Suppl. Fig 1 twice instead. It is not sufficient to only represent the data as a percentage of time. It would be very helpful to show videos of examples of how wild type and RTT mice behave in this task. Aren't MeCP2het mice known to have reduced locomotion?

Yes, MeCP2 hets and KO mice do become hypoactive. For RTT mice at 5 months of age, locomotion is slightly reduced (**Supplementary Fig. 2a**), but the reduction is not statistically significant compared to WT, as this is an early symptomatic stage.

We calculated the percentage of time mice spent in transition zones (middle area) and transition number through the middle area, which was shown in **Supplementary Fig. 1a** (now **Supplementary Fig. 2a**). The results showed that RTT and WT mice spent similar amounts of time in the transition zone and made similar numbers of crosses through the three sessions. Social behavior changes in RTT were therefore not induced by transitions in the middle area.

We have included videos with the revised manuscript.

For Fig. 2, what was the average number of neurons that they successfully recorded from in mice (they say 60-120 neurons per animal, but what was the exact range, and the mean)?

We detected 62-129 neurons per animal, with a mean of 116. We now state this in the methods.

Also, for many of the selected ROIs labeled as "other" were they active or inactive, and on average what % of ROIs were in this 'other' category?

On average, 30-40% of neurons detected were classified as "Other," meaning not selective in their response to stimulus (**Supplementary Fig. 5a**).

In Fig. 2b the traces need a scale bar for time. In Fig. 2c I could not see vertical bars to indicate the epochs of social interaction. It was not possible for me to interpret this figure.

Unfortunately, conversion to pdf removed the vertical bars, which we didn't realize. We have fixed Fig. 2c (which is now **Fig. 3a**) and added the scalebar to **Fig. 2b**.

When they say "...by S3, RTT mice showed a greater % of social-ON cells..." this was not significant, so they should make that clear.

The RTT mice showed significantly more social-ON cells (more active when interacting with M1; 34.56 ± 1.40 %) in S3 than WT mice (28.75 ± 1.78 %) in **Fig. 3e** and **Supplementary Fig. 5a** ($P = 0.043$, t-test). We added this statistical information in the text.

I did not understand panel e. Why is activity so high before the onset of interaction in the social OFF panel?

The $\Delta F/F$ of neurons around the onset of interaction (-2 sec before to 8 sec after onset) were normalized to the mean value of $\Delta F/F$ in the 2s before this 10s period, in order to provide a $\Delta F/F$ baseline before onset. Then, we calculated the average activity trace of this 10s period for all the stimulus interactions for each mouse in each session. Finally, we calculated the mean and SEM value of all mice and plotted these (**Fig. 3d** in the revision). The OFF neurons are defined by their activity after interaction onset being lower than before. Therefore, since the baseline is around 0, the normalized $\Delta F/F$ after the onset of interaction is negative.

Are shaded bars in panel f the s.e.m. of all neurons or all mice?

The shaded bars in Fig 2F (**Fig. 3d** in the revision) are for all mice; we have clarified this in the legend.

In Fig. 5 it is surprising that complete silencing of PV neurons leads to only a very modest increase (20%) in the frequency of calcium transients for exc neurons and no change in amplitude of transients. It's also surprising that the effect is not global, and only occurs with certain interactions (M1, M2). In Fig. 5h, it sounds like blue LEDs were turned on for the entire 10 min and this should be explicitly stated in the Results section too (not just Methods). [In general, the Methods section was missing important details.] It seems critical that the authors record from excitatory neurons in vivo (silicon probes) to confirm that activity was modulated by such prolonged optogenetic manipulation (is it possible that miniscopes underestimate the effect?) Were there any adverse effects of silencing PV neurons in mPFC, seizures perhaps, hyper-locomotion?

As noted above, we did not "silence all PV neurons" but only increased the frequency of excitatory neuronal activity by about 20% to reach WT levels. We did not observe any adverse effects of modestly increasing excitatory neuronal activity in the mPFC—in fact, no effects at all except increased social interaction (certainly no seizures or changes in locomotor activity).

Suppl. Fig. 3a does not make sense to me. Is there really an extremely brief increase in DF/F in all 1600 neurons? One would have expected to find a sequence of cell activations before, during and after the social interaction, but some cells would have fired for longer than others. Also, why is the hypoactivity of mPFC neurons in RTT mice not obvious from these pseudocolor raster plots? It would be important to show similar raster plots for social interaction vs. object interactions to see the differences

We think the reviewer may be misreading **Supplementary Fig. 3a**. This temporal alignment was

performed using data from each neuron over 10 minutes (600 seconds). The maximal peak activity (usually from the final 10-20 seconds) was sorted and aligned. For each neuron, many peaks were scattered over different time points, which may not look as obvious as the aligned peaks. The hypoactivity of mPFC neurons in RTT mice is reflected by a ~20% lower event rate than in WT mice. The 20% reduction may not be quite visible to the eye. Similar raster plots for social interaction vs. object interaction were shown in **Fig. 4a**.

In Suppl. Fig. 4a it seems several of the differences the authors point to are not significant, like social ON in WT (34.83) vs. RTT (29.01) or S2 vs. S3 within RTT mice (29.01 vs. 34.56). This should be stated in the text.

We have clarified this in the text.

Typos and other small things:

- *Abstract: I think some people may object to the notion that their results relate to social avoidance. They are studying social preference and discrimination.*

Point taken; we have changed to "social preference" in the abstract.

- *Why did they select 5 months of age for the mice (as opposed to 2-3 months)? Is it because the deficits in social behavior on their task are not present earlier? In the original 3-chamber assay, older mice (~6 months) tended not to participate as much. Also, the Methods section says they were tested at 3.5-4 months.*

Yes, 5 months is when the social deficit sets in. We clarify this in the revised manuscript and demonstrated the procedure in **Supplementary Fig. 1a**: we perform viral injection and GRIN lens implantation at 3.5-4 months of age, to allow 3-5 weeks for the expression of GCaMP6 in the mPFC pyramidal neurons, plus one week for baseplate mounting and habituation with the experimental room and dummy scope, before performing the social task at 5 months.

- *Page 5, the sentence "Whereas WT mice had significant changes in from one session..." needs to be rewritten.*

- *They refer to "firing rates" (e.g., in page 6), but this is not appropriate since they only recorded frequency rates of calcium transients.*

- *Page 8: "dimension-deduction" should be 'reduction'*

- *Page 9, "PV-neuron suppression increased the transient rate markedly" – not sure about the word 'markedly'; Perhaps just 'significantly' is more appropriate (it's a 30% increase at most in Fig. 5i)*

- *Page 10, "weakened, homogenized responses"; do they mean that the rate of transients is reduced throughout the various interactions (object and mouse; well, pretty much throughout the entire assay really)? If so, homogenized is not the right word. Maybe change to 'homogenously weakened', or 'weakened, non-modulated' responses?*

- *Page 14: The dimensions of the "square box" and the removable lateral chambers don't make sense. 45x10x20 cm is not a square and the dimensions of the side boxes don't fit with the main box. I suggest they draw the box to scale in Fig. 1a and Fig. 5c (and label the dimensions of each side)*

Thank you for catching these mistakes, which we have corrected, along with others we found.

- *Page 15: The paragraph entitled "Social preference paradigm" should be rewritten; it is not very clear. How was LED light delivered only when mice entered the stim zone, was it done manually under direct supervision, or was it automated somehow?*

It was automated: the LED light stimulation from the nVoke miniscope system was triggered by the behavior tracking software, which could detect the mouse's entry to the stim zone in real-time. We added this information to the Methods.

- *I was surprised that data in Fig. 5 was all from the same cohort of mice, which received the two patterns of stimulation in different sessions 1 month apart. It would be helpful if the authors labeled which mice underwent full-session stimulation first vs. last in different colors in Fig. 5.*

The mice expressing EYFP or NpHR in the mPFC were randomly assigned to two groups to receive whole-arena manipulation (pattern 1 group) or location-specific manipulation. One month later, we swapped the manipulation patterns that the two groups received and conducted the social test again. The pattern 1 and pattern 2 groups are now labeled with different colors.

- *Perhaps they should cite and comment on a different paper using miniscopes to record activity in hippocampal CA1 in social behavior in the mouse model of MeCP2dup syndrome (Sun et al., Science Bulletin 2020, 65:1192-1202), if only to mention a similar study using miniscopes in PFC, in a RTT model, is a social task*

Thank you for this suggestion. We now mention this paper in the revised manuscript.

- *In the Discussion they should comment on the fact that, in several other models of autism, there is hypoactivity of PV interneurons, not exc neurons (PMID: 30679017, 30250263, 26830140), which is the opposite of what they find. Does their data help us understand symptoms of RTT but not autism?*

All of these studies showed that E/I imbalance in the mPFC is associated with abnormal social behavior, whether in the context of autism or Rett. Previous work (Lu et al., *Neuron* 2016) showed that both too much and too little MeCP2 disrupt the same hippocampal circuit in the same way, causing hypersynchrony and impairing hippocampal-dependent memory. This could be why so many different developmental/intellectual disabilities have similar features (e.g., language delay or autistic features) despite very different molecular mechanisms—the circuits that underlie these functions can be disrupted by either too much or too little activity. We discuss this explicitly now in the revised manuscript.

- *They should probably cite Sacai et al. Nat Comm (2020) PMID: 33046712*

Thank you, we now cite this paper.

- *Figure 5k has mislabeled RTT and WT when it should be NpHR and EYFP.*

Thank you; the figure panel (which is now **Fig. 5j**) has been corrected.

Reviewer #3

This manuscript describes excitatory neuron ensembles in the mouse PL-mPFC (identified by in vivo Ca²⁺ imaging) that were either active or inactive during interactions with either mice or objects in a modified 3-chamber arena, and their altered features in female Mecp2 heterozygous mice. In addition, the authors show how optogenetic inhibition of PV interneurons in the PL-mPFC increases social interaction time in female Mecp2 heterozygous mice. The manuscript is well written (although it lacks precision about cited work and the meaning of the observed variables), and the data seem to be of sufficient quality to support the authors' interpretation. However, several issues should be addressed experimentally to better support the authors' conclusions (which are sometimes a bit exaggerated in their

extent and meaning).

We thank the reviewer for their thoughtful suggestions for sharpening the manuscript.

Major questions (require additional experiments)

1. The brain of female Mecp2 heterozygous mice shows a mosaic expression pattern of MeCP2 in all cells based on their random inactivation of the X chromosome, but the AAVs used to deliver GCaMP6m and eNpHR will transduce all cells equally. Therefore, the grouping of cells into ON ensembles using Ca²⁺ imaging traces in female Mecp2 heterozygous mice lacks information about the cellular genotype of the imaged neurons, i.e. are they MeCP2-expressing WT neurons or MeCP2-lacking mutant neurons? This is critical information because the recruitment of MeCP2-expressing WT neurons and MeCP2-lacking mutant neurons into different ensembles could be different or not, and cannot not be assumed to be one way or the other. In addition, the intrinsic (passive and active) and synaptic properties of MeCP2-expressing WT neurons and MeCP2-lacking mutant neurons cannot be assumed to be the same (because they do differ, in different ways, in different brain regions), and all those features have a significant role in the recruitment of neurons into ON ensembles and their drop-off in OFF ensembles. There are several studies using female Mecp2 heterozygous mice that revealed unexpected cell autonomous and non-cell autonomous consequences of Mecp2 loss in either neurons or glial cells. The authors need to find a way to know the cellular genotype of the imaged neurons, otherwise, the data from their “RTT mice” doesn’t have much information.

We completely agree that the synaptic properties of MeCP2⁺ and MeCP2⁻ neurons will differ, and that some effects of MeCP2 have been shown to be cell-autonomous while others are cell-non-autonomous (e.g. Maezawa et al, *J. Neurosci.*, 2009, Asgarihafshejani et al, *Neuroscience*, 2019; Rakela et al, *eLife*, 2018).

We sought to determine which cells express MeCP2 for another line of investigation in the motor cortex using an X-linked reporter line (HprtLSL-td in BL6 background, JAX stock# 021428) that labels the paternal X chromosome with tdTomato (tdT) (Wu et al., *Neuron*, 2014) to distinguish MeCP2⁺ and MeCP2⁻ neurons in female RTT mice. To make a long story short, we found that expression of reporter tdT for five months dampened neuronal activity.

We therefore took an alternative approach to answer the reviewer's question. First, we thoroughly examined all of our data looking for a bimodal distribution of effects across cells, as would be expected if MeCP2-lacking and MeCP2-expressing cells show differences with respect to firing rates, decorrelation magnitude, or other measures. We found no evidence of bimodality.

Second, we looked into the literature. Achilly et al. (*Nature* 2021) recently looked at the effects of exercise in female RTT mice and examined hippocampal neurons. Both MeCP2⁺ and MeCP2⁻ hippocampal cells were recruited during the Morris Water Maze task; as long as the MeCP2⁻ cells were activated on that specific task, they showed enhanced dendritic arborization, as did MeCP2⁺ cells.

Moreover, in previous work (Lu et al., *Neuron*, 2016) we showed that three different MeCP2-related models (male *Mecp2*-null, female *Mecp2*-het, and *Mecp2*-overexpressing mice) all showed the very same phenotype at the circuit level, namely, hypersynchrony, impaired response to perturbations of E/I homeostasis, and impaired excitatory synaptic response in inhibitory neurons. Differences at the molecular or cellular level therefore do not necessarily manifest as differences at the circuit (or behavioral) level—otherwise we would not have so many vastly different genetic mutations associated with the same behavior, such as autism or cerebellar ataxia or parkinsonism.

2. The authors need to provide specific information about how many action potentials are reflected in each GCaMP6m transient (“spike” in their words), taking into account the frame rate of image acquisition and the response kinetics of the Ca²⁺ sensor. Without this information from WT mice and the two cellular genotypes present in Mecp2 heterozygous mice, any reference to firing/spiking frequency does not carry much meaning, because it is unknown if all those parameters remain constant and similar

in neurons from WT mice, as well as in MeCP2-expressing WT neurons and MeCP2-lacking mutant neurons in Mecp2 heterozygous mice.

Unfortunately, as described in answer to the previous question, the tdT experiment failed to work. Nevertheless, we offer two reasons to believe that our findings do not depend on cell-level differences between MeCP2+ and MeCP2- neurons:

Firstly, several studies have shown that loss of MeCP2 does not affect the intrinsic properties of neurons: Sceniak et al. (*Cereb. Cortex*, 2016) for pyramidal mPFC neurons; Dani et al. (*PNAS*, 2005) and Wood et al. (*J. Neurosci.*, 2009) for cortical neurons; Calfa et al. (*Hippocampus*, 2015) for hippocampal CA3 pyramidal neurons. Loss of MeCP2 seems to affect the ability of neurons to maintain synapses rather than their intrinsic properties.

Secondly, several studies have analyzed neuronal calcium responses in MeCP2-deficient mouse neurons both *in vivo* and *ex vivo*, e.g., Runyan et al. (*PNAS* 2016) and Lu et al. (*Neuron* 2016), and found no meaningful differences in calcium kinetics in mutant neurons.

Especially now that we have established a causal relationship between our circuit observations and social behavior, knowing the cellular genotypes would be interesting but would not alter our circuit-level findings, which is the focus of our paper.

3. The authors need to provide direct evidence of what underlies the frequency of GCaMP6m transients, which they assume is different from what underlies the amplitude of each GCaMP6m transient (they seem to interpret that those two variables provide different information).

We apologize for not being clear on this point. The frequency and amplitude of the GCaMP6m transients are related: combined two-photon imaging of GCaMP6m and electrophysiology in L2/3 neurons has shown that events occur every time the neuron fires >1 action potential; the amplitude of the event ($\Delta F/F$, the change in fluorescence) correlates with the number of action potentials recorded in a burst (Chen et al., *Nature* 2013; doi:10.1038/nature12354). This information has been clarified in the Results section.

4. The authors need to perform optogenetic inhibition of PV cells in both WT and female Mecp2 heterozygous mice, not only in Mecp2 heterozygous mice.

We optogenetically inhibited PV neurons in female WT mice with NpHR (using EYFP as a control), and found that this manipulation impaired the social preference of WT mice for interacting with other mice, (**Supplementary Fig. 12a, d**). The pattern decorrelation during social interaction was also absent in NpHR-expressing WT mice (**Supplementary Fig. 12b-c, 12e-f**).

5. Optogenetic inhibition of PV cells should be confirmed by optogenetically exciting pyramidal neurons (same end result, as they interpret their data and based on their model of mPFC hypoactivity). In addition, it should be complemented with optogenetic inhibition of pyramidal neurons, as well as optogenetic excitation of PV cells. This should be done in both genotypes (i.e. “improve the RTT mice, mimic RTT in WT mice”). Only then their model and conclusions would be experimentally supported.

In addition to inhibiting PV neurons in both genotypes, we optogenetically inhibited mPFC pyramidal neurons in WT mice, replicating the hypoactive mPFC we see in RTT. The results in **Fig. 6** show that in WT mice expressing NpHR, pattern decorrelation was impaired and social preference was abolished, replicating the social deficits of RTT mice. This finding strongly supports a causal relationship between pattern decorrelation and social performance. These findings have been added to the results and are summarized in the new **Fig. 7**.

We are not sure, however, what directly exciting the WT mPFC (as opposed to exciting it by inhibiting PV neurons) might tell us that would be useful to this story. We would expect it to cause a

social abnormality, given that MeCP2 duplication mice also have a social recognition deficit due to mPFC abnormalities, which can be rescued by normalizing MeCP2 levels only in the mPFC (Yu et al., *Neurosci. Bull.* 2020). This would be interesting but not directly relevant to the situation in RTT. Exciting PV cells in RTT mice would be expected to make the mice even worse, perhaps in unexpected ways, as the mPFC has multiple functions; then again, there might be a floor effect for the social preference deficit (how much less social preference can one have?). These experiments could be very interesting, certainly, but to do them properly and pursue further questions that arise would be a whole other project, which does not seem to be of direct value for understanding mechanism or causality.

6. The authors' interpretation of the difference observed when inhibiting PV cells either in the interaction zone or the entire arena ignores the complex dynamics that result from hyperpolarizing GABAergic interneurons that are widely connected by electrical synapses (gap junctions). The authors need to consider other potential consequences of inhibiting PV cells, including their rebound excitation, and the spread of the hyperpolarization itself and of the rebound excitation through gap junctions.

We monitored the activity of mPFC pyramidal neurons during optogenetic inhibition of PV cells and titrated the increase of mPFC activity to meet the level of WT activity. The Ca²⁺ transient rate of the mPFC pyramidal neurons did not fall significantly compared to the contralateral group, after the LED light was turned off when the mouse was tested in the home cage (**Supplementary Fig. 11b**) or went to the object zone during the social test (**Supplementary Fig. 5d**). Therefore, we do not think the rebound excitation has a significant effect on the mPFC pyramidal neurons.

A recent study using *in vivo* tetrode recording showed various activity patterns of PV interneurons in response to social interaction (see Figs. 2G-H in Liu et al., *Sci. Adv.* 2020). This suggests that polarization spread via gap junctions, which should lead to synchronized activity, actually had little impact on the activity of PV interneurons. We think it unlikely that suppressing PV neurons to increase the activity of pyramidal neurons by just 20% elicited rebound excitation or gap junction spread.

We actually observed the activity changes of both pyramidal and PV neurons in the mPFC during the social interaction test. Compared to WT, the area under the curve (AUC) of PV activity traces per second in RTT mice significantly increased, while pyramidal neural activity decreased, in response to stimulus interaction. This indicates that the change in PV activity played a role in the disturbed E/I balance. Either inhibiting PV neurons or exciting pyramidal neurons directly will improve the RTT mice's social behavior, but only to the extent that they restore E/I balance and pattern decorrelation ability.

7. What was the statistical Power yielded by the sample numbers used (post-hoc)?

We calculated sample sizes initially to detect significance with an α set to 0.05 in a standard-powered experiment with a statistical power of 80% or better. Our post-hoc analysis shows that our statistical power is 92.9. We report these numbers in the revised manuscript.

Also, the authors should explicitly state if they were blind to genotypes and AAV injection conditions (for eNpHR), if mice were randomized to AAV injections (for eNpHR) and behavioral testings, and if they used specific criteria for data inclusion and exclusion. Sample numbers should include all mice used, not just the number of successful experiments that reached the analysis stage (which can be reported separately).

Thanks for catching these inadvertent omissions. Yes, experimenters were blind to both genotypes and AAV injection conditions, the mice were randomized for AAV and behavioral tests, and mice were excluded if the GRIN lens was placed outside of the prelimbic cortex (PL), or the imaging plane was unclear and occluded by blood or debris. For combined imaging and optogenetic manipulation experiments, mice were excluded if they did not exhibit NpHR or EYFP expression in PL regions, or the majority of expression was detected outside of the PL region. We have now added all this information to

the Methods.

Minor questions (may not require additional experiments)

8. *The genetic background of the Mecp2 mutant mice (S129) is different than the one of CaMKII-Cre and PV-Cre mice (C57/Bl6). The authors need to address the potential contribution of the genetic background to the observed differences in ensemble dynamics and mouse behaviors.*

The genetic background for all the experimental mice is the same: F1 hybrid 129SvEv; C57BL6 (Female heterozygous Mecp2^{+/-} on a 129SvEv genetic background crossed with male homozygous CaMKII-Cre or PV-Cre on a C57/Bl6 background).

9. *All bar graphs should include all data points, or be replaced by scatter plots.*

We have converted the graphs to scatter plots or added data points where bars still seem useful.

10. *The authors need to be explicit about the genotype of the mice in the studies cited, because the majority of the cited work used male Mecp2 KO mice and this is a major difference with the current study.*

Yes, we have revised accordingly.

11. *The text should be explicit in that the ensembles studied are made up of CaMKII-expressing cells, not all neurons in the mPFC (as implied in several parts of the manuscript).*

We have revised accordingly.

12. *The authors should be explicit about the limitations of extracting “neuronal activity” and “firing rates” from Ca²⁺ imaging data, which only provides a rough estimate by reflecting an envelope of activity due to the combination of Ca sensor kinetics and imaging frame rate. They also need to be explicit about the meaning of reductions in GCaMP6 $\Delta F/F$ (e.g. Fig. 2f): does intracellular Ca²⁺ go below resting/baseline levels during that prolonged time?*

Very good points and a very good question, as transient decreases in $\Delta F/F$ are not often observed in L2/3 neurons which fire sparsely. The $\Delta F/F$ of neurons at the onset of social interaction (-10s before to 10s after onset) were normalized to the mean value of the $\Delta F/F$ during the 2-s period before this 20-s period. Then, we calculated the average activity trace of this 20-s period for all the stimulus interactions of each mice in each session. The mean and SEM value of all mice through the 20 s around onset was calculated and plotted as Fig. 2F (which is **Fig. 3d** in the revision). The reduced $\Delta F/F$ during social interaction is due to the activity after the onset being lower than before the onset. We have updated the information in the method to explain this point.

13. *The occurrence of GCaMP6 transients should be expressed as inter-event intervals (not Hz) and plotted as cumulative distribution probabilities because the mathematical average (i.e. mean) does not accurately reflect the intrinsic variability of the data set.*

We agree that the mean does not fully illustrate the intrinsic variability of the data set. In the revised manuscript we also present the data both as a cumulative distribution of inter-event intervals, and the conclusion remains the same. The events in the RTT mice showed prolonged inter-event-intervals, a sign of hypoactivity.

14. *The reference to Phillips et al. 2019 is inaccurate: its title itself says exactly the opposite of what the*

authors wrote in the Introduction (page 5). The authors are encouraged to read carefully and completely all the papers they cite. In this case, they misrepresented the title itself; what other mistakes could be in other citations?

We apologize for this error and have made sure to be precise with our other citations.

15. The authors need to explain why did they use Camkii-Cre mice if they injected AAV1.GCaMP6m, which does not require Cre recombinase for its expression (like the DIO construct they used for eNpHR expression in PV cells from PV-Cre mice).

Thank you for catching this mistake. We updated the Methods to say we used AAV-Efla-Flex-GCaMP6m to express GCaMP6 in mPFC the pyramidal neurons and thus needed to use Camk2-Cre mice. When eNpHR is expressed in PV cells, we used AAV-CaMKII-GCamp6m virus to express GCaMP6m in the mPFC pyramidal neurons.

16. It is unclear by there are eNpHR-expressing neurons in the mPFC that is contralateral to the side where the GRIN lens was implanted, as shown in Suppl. Fig. 11. The authors state that all their injections were on a single hemisphere and right under the GRIN lens.

Briefly, we injected halorhodopsin AAV-Efla-DIO-eNpHR 3.0-EYFP (AAV-EF1a-DIO-EYFP as control virus) *bilaterally* into the prelimbic of the PV-Cre, Mecp2^{+/-} mice. Then we injected AAV-CaMKII-GCamp6m *unilaterally* into the left PL region. Thus, eNpHR was expressed bilaterally in the mPFC, but all of the imaging was performed only on the left PL. We have clarified this in the Methods.

Literature cited

- Achilly, N.P., Wang, W. & Zoghbi, H.Y. Presymptomatic training mitigates functional deficits in a mouse model of Rett syndrome. *Nature* 592, 596-600 (2021).
- Asgarihafshejani, A., Nashmi, R. & Delaney, K.R. Cell-Genotype Specific Effects of Mecp2 Mutation on Spontaneous and Nicotinic Acetylcholine Receptor-Evoked Currents in Medial Prefrontal Cortical Pyramidal Neurons in Female Rett Model Mice. *Neuroscience* 414, 141-153 (2019).
- Banerjee, A., Rikhye, R.V., Breton-Provencher, V., Tang, X., ... Jaenisch, R. & Sur, M. Jointly reduced inhibition and excitation underlies circuit-wide changes in cortical processing in Rett syndrome. *Proc Natl Acad Sci U S A* 113, E7287-E7296 (2016).
- Calfa, G., Hablitz, J.J. & Pozzo-Miller, L. Network hyperexcitability in hippocampal slices from Mecp2 mutant mice revealed by voltage-sensitive dye imaging. *J. Neurophysiol.* 105, 1768-1784 (2011).
- Calfa, G., Li, W., Rutherford, J.M. & Pozzo-Miller, L. Excitation/inhibition imbalance and impaired synaptic inhibition in hippocampal area CA3 of Mecp2 knockout mice. *Hippocampus* 25, 159-168 (2015).
- Chahrouh, M., Jung, S.Y., Shaw, C., ... Zoghbi, H.Y. MeCP2, a Key Contributor to Neurological Disease, Activates and Represses Transcription. *Science* 320, 1224 (2008).
- Chen, T.-W., Wardill, T.J., Sun, Y., Pulver, S.R., Renninger, S.L., Baohan, A., Schreiter, E.R., Kerr, R.A., Orger, M.B. & Jayaraman, V. Ultrasensitive fluorescent proteins for imaging neuronal activity. *Nature* 499, 295-300 (2013).
- Dani, V.S., Chang, Q., Maffei, A., Turrigiano, G.G., Jaenisch, R. & Nelson, S.B. Reduced cortical activity due to a shift in the balance between excitation and inhibition in a mouse model of Rett syndrome. *Proc. Natl. Acad. Sci. U. S. A.* 102, 12560-12565 (2005).
- Durand, S., Patrizi, A., Quast, Kathleen B., Hachigian, L., ... Takao K. & Fagiolini, M. NMDA Receptor Regulation Prevents Regression of Visual Cortical Function in the Absence of Mecp2. *Neuron* 76, 1078-1090 (2012).

- Grienberger, C. & Konnerth, A. Imaging calcium in neurons. *Neuron* 73, 862-885 (2012).
- Kee, S.E., Mou, X., Zoghbi, H.Y. & Ji, D. Impaired spatial memory codes in a mouse model of Rett syndrome. *Elife* 7 (2018).
- Krishnan, K., Lau, B.Y.B., Ewall, G., Huang, Z.J. & Shea, S.D. MECP2 regulates cortical plasticity underlying a learned behaviour in adult female mice. *Nature Communications* 8, 14077 (2017).
- Krishnan, K., Wang, B.S., Lu, J., ...Cang, J. & Huang, Z.J. MeCP2 regulates the timing of critical period plasticity that shapes functional connectivity in primary visual cortex. *Proc. Natl. Acad. Sci. U. S. A.* 112, E4782-4791 (2015).
- Kron, M., Howell, C.J., Adams, I.T., ... Ogier, M. & Katz, D.M. Brain activity mapping in mecp2 mutant mice reveals functional deficits in forebrain circuits, including key nodes in the default mode network, that are reversed with ketamine Treatment. *J. Neurosci.* 32, 13860-13872 (2012).
- Lau, B.Y.B., Krishnan, K., Huang, Z.J. & Shea, S.D. Maternal Experience-Dependent Cortical Plasticity in Mice Is Circuit- and Stimulus-Specific and Requires MECP2. *The Journal of Neuroscience* 40, 1514 (2020).
- Liu, L., Xu, H., Wang, J., Li, J., Tian, Y., ... Duan, S.-M. & Xu, H. Cell type-differential modulation of prefrontal cortical GABAergic interneurons on low gamma rhythm and social interaction. *Science Advances* 6, eaay4073 (2020).
- Lu, H., Ash, R.T., He, L., ...Arenkiel, B.R., Smirnakis, S.M. & Zoghbi, H.Y. Loss and gain of MeCP2 cause similar hippocampal circuit dysfunction that is rescued by deep brain stimulation in a Rett Syndrome mouse model. *Neuron* 91, 739-747 (2016).
- Maezawa, I., Swanberg, S., Harvey, D., LaSalle, J.M. & Jin, L.-W. Rett syndrome astrocytes are abnormal and spread MeCP2 deficiency through gap junctions. *J. Neurosci.* 29, 5051-5061 (2009).
- Mohammed, A.I., Gritton, H.J., Tseng, H., Bucklin, M.E., Yao, Z. & Han, X. An integrative approach for analyzing hundreds of neurons in task performing mice using wide-field calcium imaging. *Scientific Reports* 6, 20986 (2016).
- Owen, S.F., Liu, M.H. & Kreitzer, A.C. Thermal constraints on in vivo optogenetic manipulations. *Nat Neurosci* 22, 1061-1065 (2019).
- Patrizi, A., Awad, P.N., Chattopadhyaya, B., Li, C., Di Cristo, G. & Fagiolini, M. Accelerated Hyper-Maturation of Parvalbumin Circuits in the Absence of MeCP2. *Cereb. Cortex* 30, 256-268 (2020).
- Rakela, B., Brehm, P. & Mandel, G. Astrocytic modulation of excitatory synaptic signaling in a mouse model of Rett syndrome. *Elife* 7 (2018).
- Robinson, H.A. & Pozzo-Miller, L. The role of MeCP2 in learning and memory. *Learn. Mem.* 26, 343-350 (2019).
- Sceniak, M.P., Lang, M., Enomoto, A.C., ... Katz, D.M. Mechanisms of functional hypoconnectivity in the medial prefrontal cortex of Mecp2 Null mice. *Cereb. Cortex* 26, 1938-1956 (2016).
- Selimbeyoglu, A., Kim, C.K., Inoue, M., Lee, S.Y., ... Wright, M. & Deisseroth, K. Modulation of prefrontal cortex excitation/inhibition balance rescues social behavior in CNTNAP2-deficient mice. *Sci. Transl. Med.* 9, eaah6733 (2017).
- Shepherd, G.M. & Katz, D.M. Synaptic microcircuit dysfunction in genetic models of neurodevelopmental disorders: focus on Mecp2 and Met. *Curr. Opin. Neurobiol.* 21, 827-833 (2011).
- Wei, Z., Lin, B.J., Chen, T.W., Daie, K., Svoboda, K. & Druckmann, S. A comparison of neuronal population dynamics measured with calcium imaging and electrophysiology. *PLoS Comput. Biol.* 16, e1008198 (2020).
- Wood, L., Gray, N.W., Zhou, Z., Greenberg, M.E. & Shepherd, G.M. Synaptic Circuit Abnormalities of Motor-Frontal Layer 2/3 Pyramidal Neurons in an RNA Interference Model of Methyl-CpG-Binding Protein 2 Deficiency. *The Journal of Neuroscience* 29, 12440 (2009).
- Wu, H., Luo, J., Yu, H., Rattner, A., Mo, A., Wang, Y., Smallwood, P. M., Erlanger, B., Wheelan, S. J., Nathans J. Cellular Resolution Maps of X Chromosome Inactivation: Implications for Neural Development, Function, and Disease. *Neuron* 81, 103-119 (2014).
- Yue, Y., Xu, P., Liu, Z., Sun, X, Su, J., ... Lu, H. Motor training improves coordination and anxiety in symptomatic Mecp2-null mice despite impaired functional connectivity within the motor circuit.

Science Advances, In press (2021).
Yu, B., Yuan, B., Dai, J.K., Cheng, T.L., ... Liang, Z.F. & Qiu, Z.L. Reversal of Social Recognition Deficit in Adult Mice with MECP2 Duplication via Normalization of MeCP2 in the Medial Prefrontal Cortex. *Neurosci Bull* 36, 570-584 (2020).

Reviewers' Comments:

Reviewer #1:

Remarks to the Author:

Original point not being addressed:

In Figure 2c, It is still not clear how exactly did the authors calculate the "normalized transient rate". In the Method session, the authors only described how to normalized DF/F values. But in order to calculate the "transient rate" which is the spike frequency, one needs to use the number of calcium spikes being divided by the time duration. How did the author determine the duration when the mouse was running along the trial? How did the author determine the duration when the mouse stopped in front of the end chamber and started to interact with the target? Did the author use only the duration when the mouse actually interacted with the target or did the author also include the duration when mouse was just sitting there with no interacting? After calculated the frequency, how did the authors further normalize the frequency? Was the normalization done for each cell? Or was it done for each mouse?

There are many additional concerns for this panel. For instance, was this panel the averaged result from S1, S2, and S3? Or S1 and S2? Or only one session? Also, what is the point of only show WT heat map under neath? Shouldn't the authors show both WT and RTT mice? And the bottom heat map lacks the color (or gray) indicator.

Additional points:

1. The authors calculated and presented so many different parameters and different correlations. Many parameters are not in parallel with either the behavior deficits in RTT or the behavioral rescue and therefore are very distractive. For example, the authors presented amplitude difference but this was not changed in rescue experiments despite the rescue efficacy. The authors presented transient rate differences between WT and RTT and claimed that this was responsible for the social deficits. However, in the first rescue experiment, although the transient rates were increased (Figure 5d) as that of the second rescue experiment (Figure 5i), the behavior was not restored in the first experiment but worked in the second rescue experiment. All these information were very distractive. The author should focus on the story line of the most important data/conclusion which is presented in Figure 7 and clean off those unnecessary distractions.

2. In abstract, line 9-10 "Optogenetic stimulation of PV neurons in wild-type mice reproduced both the social deficit and the lack of pattern decorrelation".

There is no such data in the revised manuscript. The new experiment is to silencing the PV neurons instead of stimulating them. Did the authors mean "optogenetic silencing"?

3. Page 5, line 1-4 "Whereas the mPFC ensembles in WT mice became roughly twice as active upon the mouse's approach to either M1 or O, in RTT mice there was a much smaller difference between normalized transient rate during interactions and in the middle of the chamber".

1) Please define "mPFC ensemble", is this a subset of pyramidal neurons from the recorded mPFC neuronal population? Or is this the entire recorded population? 2) Please specify the parameter of measurement here, was it the frequency or amplitude or AUC (area underneath curve)? 3) Also, please specify if this is the averaged result from S1 and S2 sessions. 4) When you talk about "active upon approach to", have the authors included the moments after the mouse started interacting with the social target/object, or was this only restrict to the moments before mouse started interacting with the social subject or object?

4. In Figure 2e, the authors used two-way ANOVA. However, data set for some groups do not look like normal distributions, and therefore using ANOVA is problematic.

5. Page 5, line 10-11 "Notably, mPFC pyramidal neurons in both WT and RTT mice showed similar transient rates during interactions with the social and object stimuli," On Page 5, line 18-19, "These data suggest that amplitude is related to the representation of stimulus salience and novelty, whereas the Ca²⁺ transient rate underlies the relative lack of social interest of RTT mice." Both WT and RTT mice show similar transient rates for interacting with social target and object. Yet, WT mice display greater interest (longer interaction time) with social target than object, while RTT mice don't have this preference. Therefore the authors cannot use the transient rate to

explain the lack of sociability for RTT mice.

6. Page 5, line15-17, "Within-genotype comparisons showed that correlations were stronger for interaction with M1, and that the difference between M1 and O interactions was muted for the RTT mice (Supplementary Fig. 4d)". The authors claimed the stronger correlation for interaction with M1, but in Supplementary Fig. 4d, all p values are bigger than 0.05, which basically means no correlation.

7. Figure 4e: The authors put together WT and RTT to plot the correlation of discrimination index and deltaFWHM, this is to assuming WT and RTT display same relationship. This is problematic, since RTT mice display lower or even no discrimination between social target and object, they might display very different pattern or even no correlation. The authors should separately plot the correlations for WT and RTT mice.

8. Figure 5a does not provide transient rate comparison (quantification) before and after PV neuron silencing.

9. Figure 5g and 5l: I have trouble to understand these two panels. In both cases, there is a strong correlation between deltaFWHM and discrimination index. However, in 5g, the behavior was not rescued at all (i.e., no discrimination between social target and object); in 5l, the behavior was rescued. It seems to indicate that the correlation is not important (or not able) to reflect behavior deficits or behavior normality. I could not understand the point the authors were trying to present here.

Reviewer #2:

Remarks to the Author:

The revised manuscript by Xu et al is greatly improved. The authors painstakingly addressed all of my comments, and their responses are very detailed. As far as the more minor comments (related to typos or problems with figures), I have no further concerns. Regarding the more major comments, the majority were also adequately addressed, and I am satisfied. In the revised version, they have now included a missing supplemental figure, explained their statistics, inserted new references, addressed concerns about potential toxicity of prolonged LED stimulation, and expanded the methods. This helps a lot.

They also did three important additional experiments that I requested (other Reviewers did too):

1. Miniscope recordings of mPFC pyr cells during open field (I'll come back to this below); 2. Record from PV interneurons in WT and RTT mice during social interaction; and 3. Optogenetics to inhibit pyramidal neurons in WT mice. The latter two sets of experiments support their conclusions, even though the PV recordings in Suppl. Fig. 10 are not very convincing.

Still, I have some lingering concerns. First, I had asked them to record from a neutral brain region, one that would in theory have little to do with a social interaction per se. The authors respond by saying that many brain areas are affected in RTT mice. Sure, that may be the case, but presumably their activity will not be modulated by a social interaction the way RTT neurons are. If they were, then it would mean that RTT mice have global deficits in pyramidal cell activity that emerge in situations of stress or social interaction. Because the title of the paper emphasizes the "prelimbic circuit" the authors should have ruled out the possibility that other brain areas respond the same way. I am not asking for additional experiments, but they should mention this caveat in the Discussion, namely that excitatory cell hypoactivity is potentially present throughout the brain of RTT mice.

Second, related to Fig. 5, I'm still not clear on why the optogenetic silencing of inhibitory interneurons (continuous activation for 10 min) only caused modest changes in excitatory neurons. They argue that their manipulation was not so potent, because they used low LED intensities. That may be the case, but the authors should show, in a suppl. figure, how different light intensities recruit PV neurons to different degrees. If these experiments have not been done, and cannot be done, then they should state in the Methods that they did not confirm whether optogenetics

worked as intended by lowering the activity of PV neurons. In page 10 line 28 they say they increased the activity of mPFC pyramidal neurons in RTT mice by ~20% to roughly WT levels, but does Suppl. Fig. 11b show optogenetic effect presumably in RTT mice or in WT mice?

I had also requested that they confirm their result with silicon probes, as miniscope calcium imaging provides a gross underestimate of neuronal activity. I felt this would strengthen their conclusions, by independently confirming the data not just for NpHR in inhibitory neurons, but also in general for the hypoactivity of excitatory cells. They argue that Ephys is less accurate for recording neurons with low activity, and while that may be true when compared to two-photon calcium imaging, I don't think it would be the case for miniscopes. Anyway, the authors should simply acknowledge that, because of the limitations of miniscopes their data will eventually need to be confirmed with in vivo ephys.

I wanted to end by coming back to the miniscope recording experiment in the open field (Suppl. Fig. 4e), which I think is critical to the interpretation of the whole paper. I am glad they did this because it clearly shows that the hypoactivity in PFC is not solely driven by social interaction (as a reader of the original manuscript would have concluded), but more generally by anxiety-provoking situations. The difference in transient rate between RTT and WT mice in the center is striking and fits nicely with Suppl. Fig. 2a showing RTT mice avoid the center more than WT mice. In fact, the authors may want to correlate the behavior (% of time spent in center) with activity of PFC neurons (transient rate) for all the mice. Of course, this changes the main conclusion of the paper. I think it is important for the authors to really present a balanced interpretation of this finding in the Discussion. One does wonder whether the title remains accurate (should it say "social anxiety" instead of "social discrimination"?).

In conclusion, this was an ambitious and very challenging in vivo study that provides new and interesting data related to how differences in PFC circuits in RTT mice might explain their atypical behaviors in stressful situations, including social interaction. In my opinion, the authors should still consider alternative interpretations of the results of the open field test.

Small things:

- I asked that they cite Sun et al., Science Bulletin 2020, 65:1192-1202. They said they did, but I don't see it.
- Same with Sacai et al., Nat Comm 2020 (PMID: 33046712)
- 'during' is misspelled in p. 9, line 6

Reviewer #3:

Remarks to the Author:

NCOMMS-20-46394A-Z

Xu et al.

This manuscript describes excitatory neuron ensembles in the mouse PL-mPFC (identified by in vivo Ca²⁺ imaging) that were either active or inactive during interactions with either mice or objects in a modified 3-chamber arena, and their altered features in female Mecp2 heterozygous mice. In addition, the authors show how optogenetic inhibition of PV interneurons in the PL-mPFC increases social interaction time in female Mecp2 heterozygous mice. The manuscript is well written, the data seem to be of sufficient quality to support the authors' interpretation, and the responses to the prior reviews are somewhat (but not completely) satisfactory. However, several issues remain unclear and should be addressed experimentally to better support the authors' conclusions.

Major questions (require additional experiments)

1. The issue of the potential differences in the recruitment and drop-off of MeCP2-expressing and MeCP2-lacking neurons in the mosaic female Mecp2 Het mice brain is not resolved by re-analysis of the data looking for 2 different populations of cells in the dataset. For example, if MeCP2-lacking neurons were not recruited into a social ensemble, then they would not be even present in the authors' dataset. How could the authors find them in a re-analysis of their existing data, if they

(essentially) are not there?

Also, the example of dendritic morphology in hippocampal neurons after the Morris water maze is so much different at the technical and biological levels to the authors' conditions that this reviewer see no support whatsoever to the authors' interpretation of their in vivo GCaMP6 imaging data. Again, this reviewer sees no benefit in trying to find convoluted ways to avoid an experiment that it is critical to make the point that the authors are trying to make.

Regarding their failed attempt, using a tdTomato-reporter line is not the only way to identify MeCP2-expressing and MeCP2-lacking neurons in the mosaic female Mecp2 Het mice brain.

This reviewer doesn't understand how the authors' work on synchrony, E/I, and synaptic inputs to INs in Mecp2 KO, Hets, and overexpressing mice relates to this specific point: none of those parameters are in play here. If the authors want to assign mPFC neurons in the mosaic Mecp2 Het brain to social ensembles, they need to know their genetic phenotype, because MeCP2-expressing and MeCP2-lacking neurons do show some differences in some (not all) their intrinsic, synaptic, and structural properties, and ignoring them is not proper scientifically, and raises serious issues about the rigor of the authors' approach.

The revised manuscript continues to ignore this serious limitation of the authors' approach to study a mosaic brain with 2 very different types of neurons with respect to the very same gene they are precisely studying. The biology underlying this technical limitation should be made immediately clear for all readers of the manuscript to allow their full understanding of the report. Anything else still reads as disingenuous towards the readers.

2. Not all neurons lacking Mecp2 show unaltered intrinsic properties: Gantz et al. J Neurosci 2011 showed differences in Cm and Ri in MeCP2-lacking neurons compared to MeCP2-expressing in female Mecp2 Hets, and those in WT mice. Again, the authors are encouraged to reach deep into the literature to avoid ignoring critical work that is necessary for the interpretation of their own data. And they still need to provide their own experimental data of the underlying neuronal activity that is reflected in their own GCaMP6 data: what is the relationship between spikes and individual GCaMP6 transients, imaged at 15fps with their head-mounted miniscope?

3. The authors need to provide direct evidence of the relationship between spikes and individual GCaMP6 transients in their own dataset, not from the literature. Imaging GCaMP6 in a 2-photon microscope is quite different than with a head-mounted miniscope. The authors need to consider the kinetics of the sensor *and* the time resolution of their own system. Comparing their dataset from a miniscope to that from a 2-photon microscope (likely used in line-scan mode for the observations cited by the authors) is disingenuous, and again raises serious issues about the rigor of the authors' approach.

4. There are no details on how long is the GRIN lens (only its diameter) or how it was implanted into the mPFC: did they aspirated the overlying cortex like done for deeper brain regions?

5. There are still several typographical errors thought out the manuscript. The authors are encouraged to carefully revise all their files before the next submission.

The reviewers all provided very constructive comments, which have helped us further clarify and streamline the manuscript. For the sake of clarity, we present the reviewers' comments in *italics* and our responses in regular font. In-text citations are provided as a list of references at the end of this document.

Reviewer #1

We want to start by thanking Rev. 1 overall for helping us see that less is more. We think that fewer figures and analyses make the revised paper much clearer, and we appreciate the Reviewer's questioning, which helped us see we were making things unnecessarily complicated.

Original point not being addressed:

In Figure 2c, It is still not clear how exactly did the authors calculate the “normalized transient rate”. In the Method session, the authors only described how to normalized DF/F values. But in order to calculate the “transient rate” which is the spike frequency, one needs to use the number of calcium spikes being divided by the time duration. How did the author determine the duration when the mouse was running along the trial?

For duration of 'running', we measured how long the center of the mouse's body (tracked by software Topscan suite; Cleversys Inc., VA) moved in one direction without stopping. We did not mean to imply a certain speed, and so have avoided the term 'running' in the revised manuscript for this measurement.

...How did the author determine the duration when the mouse stopped in front of the end chamber and started to interact with the target? Did the author use only the duration when the mouse actually interacted with the target or did the author also include the duration when mouse was just sitting there with no interacting?

"Interaction" here was defined by sniffing behavior, which had two components: 1) the mouse had to be within the "sniffing zone," a 3 cm-wide zone near the wire that separates an end chamber from the middle chamber (see **Fig. 1a**), and 2) the mouse's nose had to be oriented toward the stimulus in the end chamber. We clarified this in the revised manuscript, in the Results section when describing the set-up of this experiment.

... After calculated the frequency, how did the authors further normalize the frequency? Was the normalization done for each cell? Or was it done for each mouse?

To determine the spike frequency we aligned the frames of the image with the behavioral data so that we could count the number of spikes that took place when the mouse was in each of the 5 zones (social/object sniffing zones, social / object zones, and the middle zone—see diagram in Fig 1a). For each recorded neuron, we divided the number of spikes counted in each zone by the time the mouse spent in that zone; this yielded 5 averaged data points for each recorded neuron. We then averaged the transient rates for *all* recorded neurons of each subject mouse in a given genotype by zone, so that we had five average transient rates for the WT and RTT groups. We then normalized these five averaged transient rates to the maximal value of the average transient rates in the WT group. We then produced a smoothed curve from these five averaged rates using BSpline and `make_interp_spline` in the `spicy` package of python (this is now **Fig. 2e** in the revised manuscript).

We have now clarified all this in the Results and Methods.

There are many additional concerns for this panel. For instance, was this panel the averaged result from S1, S2, and S3? Or S1 and S2? Or only one session? Also, what is the point of only show WT heat map under neath? Shouldn't the authors show both WT and RTT mice? And the bottom heat map lacks the

color (or gray) indicator.

Fig. 2e is the averaged result from S1 only, which we clarify in the figure legend. We showed the heat maps for both WT and RTT, as black/grey is always WT and red/coral always RTT; we included the color key in this panel in the revision. We also now include brackets to the right to make more obvious the enormous difference in dynamic range between WT and RTT mPFC.

Additional points:

1. The authors calculated and presented so many different parameters and different correlations. Many parameters are not in parallel with either the behavior deficits in RTT or the behavioral rescue and therefore are very distractive. For example, the authors presented amplitude difference but this was not changed in rescue experiments despite the rescue efficacy. The authors presented transient rate differences between WT and RTT and claimed that this was responsible for the social deficits. However, in the first rescue experiment, although the transient rates were increased (Figure 5d) as that of the second rescue experiment (Figure 5i), the behavior was not restored in the first experiment but worked in the second rescue experiment. All these information were very distractive. The author should focus on the story line of the most important data/conclusion which is presented in Figure 7 and clean off those unnecessary distractions.

The reviewer is absolutely right: we presented so many analyses in an attempt to demonstrate our point from multiple perspectives that we completely obscured that point. So what is the main point? That in order for a social preference to exist, there must be at least two meaningful patterns to disambiguate. The MeCP2-deficient circuit is so limited in its dynamic range that it cannot encode distinct patterns for object and mouse, or familiar mouse and new mouse—and without two patterns to choose from, there can be no preference. This is what Fig. 5d and i (which we have now removed) were about: Location-specific stimulation is not enough for the mPFC circuit to form two distinct patterns (e.g. M and O), so it doesn't restore social preference even though it raises the transient rate. But whole-area stimulation increases the dynamic range of the circuit so that it can form one pattern for M and another for O, and the mouse can distinguish between the two.

We have streamlined the results to make the story line clearer. We have removed, for example, the amplitude analysis as well as the section on ON and OFF neurons, which does not directly relate to our main point, except to say that from sessions 1 to 3, the WT neurons become less excitable as they become familiar with the stimuli but the RTT neurons don't seem to learn (**Fig. 3a,b**).

2. In abstract, line 9-10 “Optogenetic stimulation of PV neurons in wild-type mice reproduced both the social deficit and the lack of pattern decorrelation”. There is no such data in the revised manuscript. The new experiment is to silencing the PV neurons instead of stimulating them. Did the authors mean “optogenetic silencing”?

Yes, we meant suppressing the PV neurons—thank you for catching this mistake. We have corrected it in the abstract.

3. Page 5, line 1-4 “Whereas the mPFC ensembles in WT mice became roughly twice as active upon the mouse's approach to either M1 or O, in RTT mice there was a much smaller difference between normalized transient rate during interactions and in the middle of the chamber”.

1) Please define “mPFC ensemble”, is this a subset of pyramidal neurons from the recorded mPFC neuronal population? Or is this the entire recorded population?

It is the entire population of recorded mPFC pyramidal neurons.

2) Please specify the parameter of measurement here, was it the frequency or amplitude or AUC (area

underneath curve)?

The transient rate.

3) Also, please specify if this is the averaged result from S1 and S2 sessions.

Only the result from S1 was described.

4) When you talk about “active upon approach to”, have the authors included the moments after the mouse started interacting with the social target/object, or was this only restrict to the moments before mouse started interacting with the social subject or object?

It means the whole period over which the mouse interacted with either M1 and O in S1 (mouse nose was in the sniffing zone, oriented toward M or O).

4. In Figure 2e, the authors used two-way ANOVA. However, data set for some groups do not look like normal distributions, and therefore using ANOVA is problematic.

We performed the Shapiro-Wilk test for the normality of "transient rates" on the data of mice that are classified based on two independent variables, "genotype" and "stimuli." The results listed below showed that the subsets of individual mice are normally distributed. Therefore, we used two-way ANOVA. (Note that this is now **Fig. 2d.**)

S1

Tests of Normality^a

	Kolmogorov-Smirnov ^b			Shapiro-Wilk		
	Statistic	df	Sig.	Statistic	df	Sig.
Diff	.160	8	.200*	.966	8	.863

*. This is a lower bound of the true significance.

a. Mice = RTT, Sess = 1.0000

b. Lilliefors Significance Correction

S2

Tests of Normality^a

	Kolmogorov-Smirnov ^b			Shapiro-Wilk		
	Statistic	df	Sig.	Statistic	df	Sig.
Diff	.228	8	.200*	.902	8	.302

*. This is a lower bound of the true significance.

a. Mice = RTT, Sess = 1.0000

b. Lilliefors Significance Correction

S3

Tests of Normality^a

	Kolmogorov-Smirnov ^b			Shapiro-Wilk		
	Statistic	df	Sig.	Statistic	df	Sig.
Diff	.236	8	.200*	.908	8	.339

*. This is a lower bound of the true significance.

a. Mice = RTT, Sess = 1.0000

b. Lilliefors Significance Correction

5. Page 5, line 10-11 “Notably, mPFC pyramidal neurons in both WT and RTT mice showed similar transient rates during interactions with the social and object stimuli,” On Page 5, line 18-19, “These data suggest that amplitude is related to the representation of stimulus salience and novelty, whereas the Ca²⁺ transient rate underlies the relative lack of social interest of RTT mice.” Both WT and RTT mice show similar transient rates for interacting with social target and object. Yet, WT mice display greater interest (longer interaction time) with social target than object, while RTT mice don’t have this preference. Therefore the authors cannot use the transient rate to explain the lack of sociability for RTT mice.

We meant transient amplitudes, not rates, but we removed these panels because they didn't add anything meaningful to the story.

6. Page 5, line 15-17, “Within-genotype comparisons showed that correlations were stronger for interaction with M1, and that the difference between M1 and O interactions was muted for the RTT mice (Supplementary Fig. 4d)”. The authors claimed the stronger correlation for interaction with M1, but in Supplementary Fig. 4d, all p values are bigger than 0.05, which basically means no correlation.

We removed this panel as well.

7. Figure 4e: The authors put together WT and RTT to plot the correlation of discrimination index and deltaFWHM, this is to assuming WT and RTT display same relationship. This is problematic, since RTT mice display lower or even no discrimination between social target and object, they might display very different pattern or even no correlation. The authors should separately plot the correlations for WT and RTT mice.

We now show within-genotype correlations in **Fig. 4d** (which used to be 4e).

8. Figure 5a does not provide transient rate comparison (quantification) before and after PV neuron silencing.

The result was originally displayed in Supplementary Fig. 11b. It is now added to **Fig. 5b** in the revision.

9. Figure 5g and 5l: I have trouble to understand these two panels. In both cases, there is a strong correlation between deltaFWHM and discrimination index. However, in 5g, the behavior was not rescued at all (i.e., no discrimination between social target and object); in 5l, the behavior was rescued. It seems to indicate that the correlation is not important (or not able) to reflect behavior deficits or behavior normality. I could not understand the point the authors were trying to present here.

We have removed these panels because we realized they're not necessary and require more

explanation than they're worth. In brief, the old Fig. 5g had shown that when the mice could not discriminate between fellow mouse and object, both deltaFWHM and the discrimination index were evenly distributed around zero on both axes; Fig. 5l showed that when the mice were able to discriminate between animal and object, there was a synchronized increase in both parameters.

Reviewer #2

The revised manuscript by Xu et al is greatly improved. The authors painstakingly addressed all of my comments, and their responses are very detailed. As far as the more minor comments (related to typos or problems with figures), I have no further concerns. Regarding the more major comments, the majority were also adequately addressed, and I am satisfied. In the revised version, they have now included a missing supplemental figure, explained their statistics, inserted new references, addressed concerns about potential toxicity of prolonged LED stimulation, and expanded the methods. This helps a lot.

They also did three important additional experiments that I requested (other Reviewers did too): 1. Miniscope recordings of mPFC pyr cells during open field (I'll come back to this below); 2. Record from PV interneurons in WT and RTT mice during social interaction; and 3. Optogenetics to inhibit pyramidal neurons in WT mice. The latter two sets of experiments support their conclusions, even though the PV recordings in Suppl. Fig. 10 are not very convincing.

We thank the reviewer for recognizing our effort in addressing each comment. We believe that the revised manuscript is now even better, thanks to additional constructive comments.

Still, I have some lingering concerns. First, I had asked them to record from a neutral brain region, one that would in theory have little to do with a social interaction per se. The authors respond by saying that many brain areas are affected in RTT mice. Sure, that may be the case, but presumably their activity will not be modulated by a social interaction the way RTT neurons are. If they were, then it would mean that RTT mice have global deficits in pyramidal cell activity that emerge in situations of stress or social interaction.

We have now performed an additional experiment to address this question. We monitored layer 2/3 pyramidal neurons in the primary motor cortex (a neutral brain region) of both WT and RTT mice during the social interaction test. We found no significant difference in the transient rates of the motor cortical neurons of RTT mice at any point in the three-chamber test (see new **Suppl. Fig. 4b**). Therefore, we concluded that social-induced hypoactivity is not a global deficit, but specific to the mPFC. PV neurons have the inverse problem, being too active in response to interaction with stimuli but quite variable in the middle of the chamber (**Suppl. Fig. 6**). Nonetheless, both mPFC and PV neurons in RTT mice have a more limited dynamic range than WT.

We discuss this important point in the revision.

Because the title of the paper emphasizes the "prelimbic circuit" the authors should have ruled out the possibility that other brain areas respond the same way. I am not asking for additional experiments, but they should mention this caveat in the Discussion, namely that excitatory cell hypoactivity is potentially present throughout the brain of RTT mice.

As noted above, we now explicitly mention that the abnormal responsiveness of *MECP2*-deficient neurons becomes evident only in the context of stimulation.

Second, related to Fig. 5, I'm still not clear on why the optogenetic silencing of inhibitory interneurons (continuous activation for 10 min) only caused modest changes in excitatory neurons. They argue that

their manipulation was not so potent, because they used low LED intensities. That may be the case, but the authors should show, in a suppl. figure, how different light intensities recruit PV neurons to different degrees. If these experiments have not been done, and cannot be done, then they should state in the Methods that they did not confirm whether optogenetics worked as intended by lowering the activity of PV neurons.

Although we did not record the activity of the NpHR-expressing PV neurons during the light stimulation, we did record the activity of the excitatory pyramidal neurons in the RTT mice with a subpopulation of PV neurons expressing NpHR and the NpHR-expressing pyramidal neurons in the WT mice during the light stimulation. The results in **Fig. 5b** and **Fig. 6b** in the revision confirmed that the optogenetics worked as intended. We also explicitly mention in the manuscript that we did not explore how different light intensities recruit PV neurons to different degrees.

In page 10 line 28 they say they increased the activity of mPFC pyramidal neurons in RTT mice by ~20% to roughly WT levels, but does Suppl. Fig. 11b show optogenetic effect presumably in RTT mice or in WT mice?

Suppl. Fig. 11b (**Fig. 5b** in the revision) shows optogenetic effect in the RTT mice, as noted in the revised figure legend.

I had also requested that they confirm their result with silicon probes, as miniscope calcium imaging provides a gross underestimate of neuronal activity. I felt this would strengthen their conclusions, by independently confirming the data not just for NpHR in inhibitory neurons, but also in general for the hypoactivity of excitatory cells. They argue that Ephys is less accurate for recording neurons with low activity, and while that may be true when compared to two-photon calcium imaging, I don't think it would be the case for miniscopes. Anyway, the authors should simply acknowledge that, because of the limitations of miniscopes their data will eventually need to be confirmed with in vivo ephys.

This would be great to do, but unfortunately, we do not currently have the capacity to do silicon probe experiments in our lab or department.

We will note, however, that another group did use electrophysiology on mPFC slices: Sceniak et al. (*Cerebral Cortex*, 2016) showed that pyramidal neurons in the *Mecp2* null mPFC exhibit significant reductions in excitatory postsynaptic currents, the duration of excitatory UP-states, evoked population activity, and the ratio of NMDA:AMPA currents. We cite this paper in the manuscript and mention the concern that miniscope calcium imaging can underestimate neuronal activity.

I wanted to end by coming back to the miniscope recording experiment in the open field (Suppl. Fig. 4e), which I think is critical to the interpretation of the whole paper. I am glad they did this because it clearly shows that the hypoactivity in PFC is not solely driven by social interaction (as a reader of the original manuscript would have concluded), but more generally by anxiety-provoking situations. The difference in transient rate between RTT and WT mice in the center is striking and fits nicely with Suppl. Fig. 2a showing RTT mice avoid the center more than WT mice. In fact, the authors may want to correlate the behavior (% of time spent in center) with activity of PFC neurons (transient rate) for all the mice. Of course, this changes the main conclusion of the paper. I think it is important for the authors to really present a balanced interpretation of this finding in the Discussion. One does wonder whether the title remains accurate (should it say "social anxiety" instead of "social discrimination"?).

We had acknowledged the possibility of an influence of anxiety in our previous version of the manuscript, but the reviewer's comments here prompted us to perform an additional experiment: We now show that optogenetic stimulation rescues anxiety in the anxiogenic center area. More specifically, we optogenetically suppressed PV neurons in the anxiogenic center area of the open field chamber, elevating

the transient rate of mPFC excitatory neurons, and this enabled the mice to spend more time in the center, i.e., it relieved anxiety (new **Suppl. Fig. 9**). Because this stimulation is precisely the same as what we used in the location-specific social test, yet the mice still displayed a lack of social preference, this shows that anxiety is not the primary cause of the social deficit. If it were, then rescuing the circuit hypoactivity in the social zone would have prolonged the time that the RTT mice spent in the social zone. Therefore, we conclude that the problem in RTT is actually a lack of pattern decorrelation, which is now a much stronger focus in the revised manuscript.

In conclusion, this was an ambitious and very challenging in vivo study that provides new and interesting data related to how differences in PFC circuits in RTT mice might explain their atypical behaviors in stressful situations, including social interaction. In my opinion, the authors should still consider alternative interpretations of the results of the open field test.

We thank the reviewer for their appreciation of our work and for the very helpful comments. We believe that our revised manuscript resolves the question of whether anxiety contributes to the lack of social preference in RTT.

Small things:

- *I asked that they cite Sun et al., Science Bulletin 2020, 65:1192-1202. They said they did, but I don't see it.*
- *Same with Sacai et al., Nat Comm 2020 (PMID: 33046712)*
- *'during' is misspelled in p. 9, line 6*

Sun et al. is ref. 42; Sacai et al. is ref. 24 in the revised manuscript.

Reviewer #3

This manuscript describes excitatory neuron ensembles in the mouse PL-mPFC (identified by in vivo Ca2+ imaging) that were either active or inactive during interactions with either mice or objects in a modified 3-chamber arena, and their altered features in female Mecp2 heterozygous mice. In addition, the authors show how optogenetic inhibition of PV interneurons in the PL-mPFC increases social interaction time in female Mecp2 heterozygous mice. The manuscript is well written, the data seem to be of sufficient quality to support the authors' interpretation, and the responses to the prior reviews are somewhat (but not completely) satisfactory. However, several issues remain unclear and should be addressed experimentally to better support the authors' conclusions.

Thanks to the reviewers' comments, we realized we had inadvertently made it more difficult for the reader to discern the main point. The main thrust of the work is that the ability to form distinct patterns encoding different experiences (here, encountering an object or another mouse) is necessary for an animal to be able to exercise a preference. In RTT, the mPFC circuit has a dampened response to stimulation; its dynamic range is much narrower than that of WT neurons, and so it is impaired in pattern decorrelation. The major implication of our work is that pattern decorrelation in the mPFC is necessary for social preference, and that the social deficit in RTT is not a matter of "social avoidance" due to anxiety or aversion to fellow creatures, but rather an inability to disambiguate fellow creature and inanimate object.

Major questions (require additional experiments)

1. The issue of the potential differences in the recruitment and drop-off of MeCP2-expressing and MeCP2-lacking neurons in the mosaic female Mecp2 Het mice brain is not resolved by re-analysis of the

data looking for 2 different populations of cells in the dataset. For example, if MeCP2-lacking neurons were not recruited into a social ensemble, then they would not be even present in the authors' dataset. How could the authors find them in a re-analysis of their existing data, if they (essentially) are not there?

The penultimate paragraph in the Discussion now explicitly refers to the limitation of our study in not being able to discern the MeCP2 genotype (more on this below). Here, we will just say that previous studies have shown that MeCP2- and MeCP2+ neurons are both recruited to neural ensembles (Braunschweig et al., Hum. Mol. Genet., 2004); this was the best we could do in this circumstance.

Also, the example of dendritic morphology in hippocampal neurons after the Morris water maze is so much different at the technical and biological levels to the authors' conditions that this reviewer see no support whatsoever to the authors' interpretation of their in vivo GCaMP6 imaging data. Again, this reviewer sees no benefit in trying to find convoluted ways to avoid an experiment that it is critical to make the point that the authors are trying to make.

We were not trying to "find ways to avoid an experiment"—we simply could not perform the requested experiment. Again, more on this below, in response to a later comment.

Regarding their failed attempt, using a tdTomato-reporter line is not the only way to identify MeCP2-expressing and MeCP2-lacking neurons in the mosaic female Mecp2 Het mice brain.

This reviewer doesn't understand how the authors' work on synchrony, E/I, and synaptic inputs to INs in Mecp2 KO, Hets, and overexpressing mice relates to this specific point: none of those parameters are in play here.

We apologize for not being clear. It's not the parameters, it's the level of function. The circuits of Mecp2 KO, hets, and duplication mice are utterly different at the level of individual neurons, but the circuits behaved the same way across genotypes.

If the authors want to assign mPFC neurons in the mosaic Mecp2 Het brain to social ensembles, they need to know their genetic phenotype, because MeCP2-expressing and MeCP2-lacking neurons do show some differences in some (not all) their intrinsic, synaptic, and structural properties, and ignoring them is not proper scientifically, and raises serious issues about the rigor of the authors' approach.

The reviewer's interest is in how the genotype of individual neurons in the mPFC affects their recruitment to the circuit (female "Rett" mice are heterozygous for *Mecp2* expression). After realizing that the section on socially-tuned (ON and OFF) neurons did not add anything useful to the story, we have removed this data.

In fact, we did find one paper (Asgarihafshejani et al., *Neuroscience* 2019) that showed that MeCP2 genotype does exert both cell-autonomous and nonautonomous effects, and that both MeCP2- and MeCP2+ neurons are recruited to the mPFC circuit. Moreover, the behavior of the cells depended on the age of the mice. We now mention this paper in our revised manuscript, and include a paragraph in the Discussion explicitly pointing out that it is a limitation of our study that we were not able to discern the genotypes of the individual neurons.

Lest the reviewer continue to doubt our rigor, however, we want to explain the difficulty posed by their experimental request. In order for us to know how MeCP2-negative neurons are recruited into the mPFC circuit during social behavior, we need to be able to visualize the neurons while the mice are freely engaging (or not) with other mice or objects, so we have to use a miniscope that can be mounted onto the mouse's head so it can move around freely. It also means that we need a red reporter to distinguish the MeCP2-positive and negative neurons (since GCaMP is green). In the current work we used an nVoke miniscope, which combines imaging and optogenetics. When we got the first set of reviewers' comments

back in January 2021, there was no miniscope on the market that could visualize red. Therefore, our only option was to use the desktop 2-photon system in our lab that can image neuronal activity in mice that are head-fixed beneath the microscope's objective (our laser has two beams to enable two-color imaging). Head-fixed animals can perform only very restricted behaviors, such as motor skill learning (Yue et al., *Science Advances*, 2021). We therefore decided to image motor cortical neurons first with our mice expressing tdTomato (tdT) in the MeCP2+ neurons to validate performing Ca²⁺ imaging in these tdT-expressing mice. As we explained in our rebuttal, we found that tdT expression for six months (the age of the mice when we performed the social interaction experiments) suppresses the activity of wild-type neurons, rendering any comparison with MeCP2-negative neurons moot. We then decided to image 2-month-old mice and found that at this earlier stage, tdT expression had not yet exerted toxic effects in wild-type neurons. But 2-month-old RTT mice do not show the same behavioral deficit as 6-month-old RTT mice, so this still wouldn't give us a *useful* answer to the question of neuronal genotype.

However, the result we got from 2-month old RTT mice is still valuable. We found that the MeCP2-negative cells in the RTT mice have, as expected, a lower event rate than MeCP2-positive cells. Despite this difference between MeCP2-positive and -negative neurons, the correlation coefficients (on which many of our analyses are based) do *not* differ between the MeCP2-positive and MeCP2-negative cell populations. (Please pardon us for not showing these results in a figure because figures are not allowed in the rebuttal if they are not part of the paper, per editorial instructions)

Ultimately, we have to consider the costs of getting an answer to a question that does not seem germane to the study. The costs would indeed be high, while the benefits would be uncertain. There are only two approaches we can conceive of that might provide an answer to the genotype question:

- 1) Upgrade our nVoke one-photon head-mounted miniscope, which can only visualize green fluorescence, to a model (nVue), just released in August, that can image both green and red fluorescence (at a cost of \$80k) and then measure the activity from both tdT-labeled red MeCP2+ neurons and MeCP2- neurons in 2-month-old mice (i.e., before the tdT has a chance to suppress neuronal activity). This approach would still not provide an answer, however, despite the expense, because we would have to repeat all our studies with 2-month-old mice, who do not yet show deficits in the social approach test. We might learn the genotype of individual neurons, but before the circuit becomes clearly dysfunctional—so it would add nothing to our understanding of circuit dysfunction after the onset of RTT.
- 2) Use the CreER line to breed the mice and induce the expression of the reporter in a time-controlled manner. This is also far from ideal, as it would take at least one year of breeding and aging to get the experimental mice—if we are lucky and this strategy works at the first attempt. (We asked Jeremy Nathans, who gave us the tdT reporter line, if his lab has tried this strategy, and they have not.)

In sum: We are not opposed to the Reviewer's request in principle, but we do not think it can be properly accommodated. Because I don't consider myself an expert in mouse genetics, I asked both Jeremy Nathans and my former mentor, Huda Zoghbi, if they could think of any other approach that would work, and they could not.

Therefore, we are not dismissing this request, but simply unable to meet it. Because of this, we reasoned from the existing literature—it's the best we can do, and we do feel there is merit in our arguments, although we understand this is not very satisfying to the reviewer's curiosity.

The revised manuscript continues to ignore this serious limitation of the authors' approach to study a mosaic brain with 2 very different types of neurons with respect to the very same gene they are precisely studying. The biology underlying this technical limitation should be made immediately clear for all readers of the manuscript to allow their full understanding of the report. Anything else still reads as disingenuous towards the readers.

We want to be clear: we are not studying MeCP2 *per se*. We are trying to understand what enables or disables social preference in mice, focusing on the mPFC.

2. Not all neurons lacking Mecp2 show unaltered intrinsic properties: Gantz et al. J Neurosci 2011 showed differences in Cm and Ri in MeCP2-lacking neurons compared to MeCP2-expressing in female Mecp2 Hets, and those in WT mice. Again, the authors are encouraged to reach deep into the literature to avoid ignoring critical work that is necessary for the interpretation of their own data. And they still need to provide their own experimental data of the underlying neuronal activity that is reflected in their own GCaMP6 data: what is the relationship between spikes and individual GCaMP6 transients, imaged at 15fps with their head-mounted miniscope?

Unfortunately, not only are we not able to do such an experiment, but we are unaware of any lab that could perform such verification, given that it requires doing electrophysiological recording on a mouse with a miniscope mounted on the head and a GRIN lens implanted in a deep brain region (mPFC). We would need to record Ca²⁺ activity in vivo with the head-mounted miniscope and then do patch-clamp recordings from the same neuron. This is not possible: the GRIN lens allows visualization of neurons but cannot be used to guide the patch-clamp to specific neurons. Once again, to make sure I wasn't missing something, I wrote to Inscopix to ask what they knew about the relationship between spikes and individual GCaMP6 transients, and was told that they do not have a method to register cells across the two methods, either. They said the only option would be to do a slice study, but that this would have the obvious drawback of comparing in vivo with in vitro data (not to mention the fact that we couldn't be sure to patch-clamp the same cells that we visualized in vivo).

Given this, we did the best we could. We compared the Ca²⁺ transients/events (not individual neuronal spikes) from the *same* group of neurons within each mouse throughout the whole behavioral assay and across different tests, over a period of approximately one month. A Ca²⁺ transient/event was calculated as a way to make comparisons between sessions and groups with a consistent standard, as have many papers published in reputable journals, including *Nature Communications* (e.g., Jimenez et al., *Nature Communications*, 2020; Kingsbury et al., *Cell*, 2019; Liang et al., *Neuron*, 2018).

*3. The authors need to provide direct evidence of the relationship between spikes and individual GCaMP6 transients in their own dataset, not from the literature. Imaging GCaMP6 in a 2-photon microscope is quite different than with a head-mounted miniscope. The authors need to consider the kinetics of the sensor *and* the time resolution of their own system. Comparing their dataset from a miniscope to that from a 2-photon microscope (likely used in line-scan mode for the observations cited by the authors) is disingenuous, and again raises serious issues about the rigor of the authors' approach.*

As noted above, we compared the Ca²⁺ transients/events (not individual neuronal spikes) from the same group of neurons within each mouse throughout the whole behavioral assay and across different tests, over a period of approximately one month. A Ca²⁺ transient/event was calculated as a way to make comparisons between sessions and groups with a consistent standard, as have many papers published in reputable journals, including *Nature Communications* (e.g., Jimenez et al., *Nature Communications*, 2020; Kingsbury et al., *Cell*, 2019; Liang et al., *Neuron*, 2018—see full citations below).

4. There are no details on how long is the GRIN lens (only its diameter) or how it was implanted into the mPFC: did they aspirated the overlying cortex like done for deeper brain regions?

The length of the GRIN lens used in the current study is 4 mm. It was inserted into the brain at a very low speed along with the virus injection track, without aspiration of any brain tissue. This information has been added to the Methods.

5. There are still several typographical errors thought out the manuscript. The authors are encouraged to carefully revise all their files before the next submission.

Thank you; we have done our best in the revision. We appreciate the reviewer's close reading and patience with our sometimes lengthy answers. We hope that Reviewer 3 better understands now the technical limitations that we face, and that we are not being "disingenuous." The central question we set out to answer is: what enables or disables social preference mPFC circuit. We hope the reviewers will judge the paper on whether our data answer that question.

Reviewers' Comments:

Reviewer #1:

Remarks to the Author:

All my concerns have been addressed in the revision. I have no further questions.

Reviewer #2:

Remarks to the Author:

The authors have addressed my remaining concerns after submitting this re-revised manuscript. They have even done additional experiments (e.g., recording in motor cortex). They have also streamlined the main figures, removing non-essential analyses, which will help the reader. Although I'm still skeptical of certain results and interpretations (e.g., I feel the authors conclusion that PV interneurons are hypoactive in PFC of RTT mice is based on tenuous miniscope evidence), I think the overall contribution is substantial and worthy of publication. It is certainly greatly improved from the original submission. I have no further comments.

Reviewer #3:

Remarks to the Author:

NCOMMS-20-46394B

Xu et al.

This manuscript describes excitatory neuron ensembles in the mouse PL-mPFC (identified by in vivo Ca²⁺ imaging) that were either active or inactive during interactions with either mice or objects in a modified 3-chamber arena, and their altered features in female *Mecp2* heterozygous mice. In addition, the authors show how optogenetic inhibition of PV interneurons in the PL-mPFC increases social interaction time in female *Mecp2* heterozygous mice. The manuscript is well written, the data seem to be of sufficient quality to support the authors' interpretation, and the responses to the prior reviews are somewhat (but not completely) satisfactory. However, a critical issue for this reviewer remains unsupported, which in this reviewer's opinion should be addressed experimentally to fully support the authors' conclusions regarding the mPFC network deficits in female *Mecp2* Het mice.

In short, either address the weaknesses experimentally or tone down the interpretations and conclusions so the future readers are fully aware, early on in the manuscript (in Results when first describing miniscope data, rather than deep into the Discussion), of what the actual raw data of GCaMP6-expressing neurons imaged with a miniscope actually reflects, especially in the mosaic brain of female *Mecp2* Het mice with unknown levels of cell-autonomous and non-cell-autonomous *functional* consequences at the cellular and network levels, as well as at the behavioral level.

Major issue

The authors response to the issue of the mixed population of neurons in the female *Mecp2* Het mice makes this reviewer wonder what is the benefit of characterizing ensembles of active neurons if there are 2 cell populations with major genetic differences, MeCP2 itself and the ~1,100 genes modulated by it. In short, if the authors do not see the benefit of distinguishing between them (focusing instead at the systems level), why then use an imaging approach, which yields cellular resolution?

About their reference to Braunschweig et al. (2004) to support the statement "MeCP2- and MeCP2+ neurons are both recruited to neural ensembles", that article does not include any experiments showing recruitment of neurons into ensembles. In short, it seems that they did not read that paper before citing it as proving support for their statements.

About their reference of Asgarihafshejani et al. (2019), that paper shows synaptic currents and ACh-evoked currents from individual mPFC pyramidal neurons, which cannot infer their "recruitment into ensembles" because every recording is from a single neuron at a time. In short, it seems that they did not read that paper either before citing it as proving support for their

statements.

Minor issue

The authors need to choose a definition for what they mean by “ensemble” and explicitly share it with the future readers, because it seems that it refers to any random group of neurons that showed GCaMP6 transient signals during a behavioral task. The problem is that neurons that did not get transduced by AAV to express GCaMP6 may still be part of the ensemble of socially-engaged neurons but will be missed, while GCaMP6-expressing neurons that are not part of such ensemble will be incorrectly included. This limitation of this imaging approach should also be explicitly stated for future readers.

As with previous responses to the reviewers, we present the reviewer comments in *italics* and our responses in regular font.

Reviewer #1

All my concerns have been addressed in the revision. I have no further questions.

This reviewer's detailed, constructive comments helped us greatly improve the paper.

Reviewer #2

The authors have addressed my remaining concerns after submitting this re-revised manuscript. They have even done additional experiments (e.g., recording in motor cortex). They have also streamlined the main figures, removing non-essential analyses, which will help the reader. Although I'm still skeptical of certain results and interpretations (e.g., I feel the authors conclusion that PV interneurons are hypoactive in PFC of RTT mice is based on tenuous miniscope evidence), I think the overall contribution is substantial and worthy of publication. It is certainly greatly improved from the original submission. I have no further comments.

We appreciate that the reviewer finds this version greatly improved. We thoroughly agree, and thank the reviewer for their experimental suggestions.

We have qualified the conclusion about PV interneurons (although consistent with Asgarihafshejani et al. 2019, which we cite, as their electrophysiological studies found conserved or increased inhibitory charge in the context of diminished excitatory activity in the mPFC of MeCP2-deficient mice).

Reviewer 3

This manuscript describes excitatory neuron ensembles in the mouse PL-mPFC (identified by in vivo Ca2+ imaging) that were either active or inactive during interactions with either mice or objects in a modified 3-chamber arena, and their altered features in female Mecp2 heterozygous mice. In addition, the authors show how optogenetic inhibition of PV interneurons in the PL-mPFC increases social interaction time in female Mecp2 heterozygous mice. The manuscript is well written, the data seem to be of sufficient quality to support the authors' interpretation, and the responses to the prior reviews are somewhat (but not completely) satisfactory...

We appreciate this reviewer's patience and persistence. We also appreciate that they think the manuscript well-written, but that reflects a very slow and painstaking process along with a lot of editing help from a native speaker.

The summary above really refers to an earlier version of the manuscript, when we had defined Social-ON and Social-OFF neurons as ensembles in a stricter sense of being coactive under the same circumstances. For various reasons we removed this particular data but the word "ensemble" was left behind in a couple of places in the manuscript, and we didn't even realize it, in part because we're not native English speakers and think of ensemble as 'group' as well as the more strict definition. (We still do examine coactivity when we look at the most highly correlated neurons and find that they still show pattern decorrelation in WT and lack of it in RTT, but we do not talk about ensembles there. We also still discuss socially responsive neurons in Figure 3b and describe our calculations in the Methods; we used the method of Liang et al., comparing the activity level with the neuron's own chance level of activity.).

Thus, whereas Reviewers 1 and 2 had early on picked up our interest in pattern decorrelation, Reviewer 3 maintained a focus on ON and OFF neurons from the very first round of review. We should note that Reviewer 1, in round 2, also asked us to clarify what we meant by “ensemble”, which we did, but apparently Reviewer 3 did not catch that little exchange in the last rebuttal. In short, we and Rev. 3 have been 'talking past' each other for the last two rounds.

Luis Carrillo-Reid and Rafael Yuste recently put forward a definition of "ensemble" (in "What is a Neuronal Ensemble?" *Oxford Research Encyclopedia: Neuroscience* 2020) as "a group of neurons repeatedly firing together." This implies not only coactivity but the sequential activation of the group. The social-responsive and social-nonresponsive neurons we discussed in the paper were active under the same social/nonsocial circumstances, but the experimental design did not allow us to test consistent reactivation of the group. This would entail a lengthy series of experiments such as in the study performed by Liang et al. ("Distinct and dynamic ON and OFF neural ensembles in the prefrontal cortex code social exploration," *Neuron* 2018), when they explored distinct ON and OFF ensembles in the mPFC during social exploration. Interestingly, that paper also slips between looser and stricter definitions of ensemble: they begin by identifying ON and OFF ensembles, but as they scrutinize the behavior of these groups in more detail, they find that there are sub-ensembles (our term, not theirs!), that are reactivated under the same experimental conditions. This leads them to propose a model of "dynamic population coding" because "neural ensembles tuned to the same behavior identified at different testing stages were not identical, but they shared common neurons across stages." (In fact, only 30-50% of the ON or OFF neurons were coactive across any two epochs.) So by the strict Carrillo-Reid/Yuste definition, the original ON/OFF groups would not be considered ensembles, but Liang et al. were clear about the meaning they were using and we would not quibble with their terminology.

Regrettably, we were not so clear. Through the evolution of the manuscript, we slipped between stricter and looser definitions. (Carrillo-Reid and Yuste discuss the history of thinking about ensembles and say, rightly, that the literature has "not been very disciplined" in using a variety of terms for ensemble, assembly, etc.) We went back through the manuscript to make sure we didn't refer to “ensemble” (there were two instances of the word in the results, and they are now gone).

...However, a critical issues for this reviewer remains unsupported, which in this reviewer's opinion should be addressed experimentally to fully support the authors' conclusions regarding the mPFC network deficits in female Mecp2 Het mice.

*In short, either address the weaknesses experimentally or tone down the interpretations and conclusions so the future readers are fully aware, early on in the manuscript (in Results when first describing miniscope data, rather than deep into the Discussion), of what the actual raw data of GCaMP6-expressing neurons imaged with a miniscope actually reflects, especially in the mosaic brain of female Mecp2 Het mice with unknown levels of cell-autonomous and non-cell-autonomous *functional* consequences at the cellular and network levels, as well as at the behavioral level.*

We removed the word "ensemble" from the manuscript and have pointed out the limitations of the experiments. In the revision, we explained how to quantify GCaMP6m signal and what it reflects early in the Results, saying:

To determine whether population coding differs in response to different stimuli, we first estimated the Ca²⁺ transient rate and amplitude (here defined as $\Delta F/F$, the change in GCaMP fluorescence intensity relative to baseline; it reflects number of spikes per transient²⁶).

The reference is as follows:

26. Chen T-W, *et al.* Ultrasensitive fluorescent proteins for imaging neuronal activity. *Nature* **499**, 295-300 (2013).

We end an early section of the Results with a comment about the genotype:

...we recorded these socially-neutral L2/3 motor cortical neurons in RTT mice during the three-chamber test and found they were not hypoactive in the context of social behavior (**Supplementary Fig. 4b**). Therefore, the loss of MeCP2 seems to dampen the responsiveness of excitatory circuits, in both the mPFC and the motor cortex. Because we were unable to determine the genotypes of individual recorded neurons, we cannot say whether this is due to cell-autonomous or non-cell-autonomous effects of MeCP2 loss, but both are possible²¹.

As in the last version of the manuscript, we mention the genotype issue also in the penultimate paragraph of the discussion.

Major issue

The authors response to the issue of the mixed population of neurons in the female Mecp2 Het mice makes this reviewer wonder what is the benefit of characterizing ensembles of active neurons if there are 2 cell populations with major genetic differences, MeCP2 itself and the ~1,100 genes modulated by it. In short, if the authors do not see the benefit of distinguishing between them (focusing instead at the systems level), why then use an imaging approach, which yields cellular resolution?

If we are correct, and the Reviewer has been thinking that we were defining an ensemble as all coactive neurons rather than just studying all the recorded neurons, we can understand now why they ask this question. To define an ensemble (in the strict sense) requires identifying those individual neurons that are coactive and re-engaged in sequence.

In this manuscript, however, we focus on pattern decorrelation among the population of neurons we were able to record (62-129 neurons per mouse). We explicitly stated this in the last version, in both the Results and the Methods. We found that pattern decorrelation is necessary for social preference, which is the main message of the current work. Other imaging techniques such as photometry, which measures the bulk response from a group of neurons without cellular resolution, would not have allowed us to analyze the temporal correlation of neuronal activities.

About their reference to Braunschweig et al. (2004) to support the statement “MeCP2- and MeCP2+ neurons are both recruited to neural ensembles”, that article does not include any experiments showing recruitment of neurons into ensembles. In short, it seems that they did not read that paper before citing it as proving support for their statements.

The reviewer is quoting from the rebuttal. We read Braunschweig et al., but we were not precise in what we wrote, and used 'ensemble' in the rebuttal to mean the population of recorded neurons.

About their reference of Asgarihafshejani et al. (2019), that paper shows synaptic currents and ACh-evoked currents from individual mPFC pyramidal neurons, which cannot infer their “recruitment into ensembles” because every recording is from a single neuron at a time. In short, it seems that they did not read that paper either before citing it as proving support for their statements.

Again, we read the paper, and in the manuscript we cited it correctly. We cited it first in the introduction, to say "a more recent study in heterozygous *Mecp2* female mice used *ex vivo* electrophysiology to find reduced excitatory postsynaptic currents and altered excitatory/inhibitory (E/I) balance²¹", and later in the Discussion to say that "one study using *ex vivo* physiology showed that MeCP2-deficient neurons in the female RTT mouse mPFC have increased inhibitory input, while the MeCP2-expressing neurons are similar to those in WT animals²¹." Both of these statements are correct, and we made a point of bringing it to the Discussion to support bringing up the issue of MeCP2 heterozygosity, at this reviewer's request.

We believe that if the reviewer reads our revised manuscript, rather than just the rebuttal, s/he will see this is the case.

Minor issue

The authors need to choose a definition for what they mean by "ensemble" and explicitly share it with the future readers, because it seems that it refers to any random group of neurons that showed GCaMP6 transient signals during a behavioral task.

This comment helped us realize the source of all this misunderstanding—thank you. We have specified in the manuscript when we are looking at the whole population of recorded neurons (e.g., Fig 2, 4) and when we are looking only at socially-responsive neurons (Fig 3b, Suppl Fig 5).

The problem is that neurons that did not get transduced by AAV to express GCaMP6 may still be part of the ensemble of socially-engaged neurons but will be missed, while GCaMP6-expressing neurons that are not part of such ensemble will be incorrectly included. This limitation of this imaging approach should also be explicitly stated for future readers.

The reviewer is correct, and this would indeed be a concern if we were trying to identify ensembles in the strict sense of coactive neurons whose sequential activity can be repeated. We are not. We are analyzing the whole population of neurons that express GCaMP6. Small groups of neurons that are repeatedly coactivated by the same stimulus are, by definition, differentiated. What we show is that, at the level of the recorded population, this sort of differentiation must be able to take place in order to enable the expression of social preference.